# A Theoretical Analysis on Feature Learning in Neural Networks: Emergence from Inputs and Advantage over Fixed Features

**Zhenmei Shi[*], Junyi Wei[*], Yingyu Liang**
University of Wisconsin-Madison
`zhmeishi@cs.wisc.edu,jwei53@wisc.edu,yliang@cs.wisc.edu`

## Abstract

An important characteristic of neural networks is their ability to learn representations of the input data with effective features for prediction, which is believed to be a key factor to their superior empirical performance. To better understand the source and benefit of feature learning in neural networks, we consider learning problems motivated by practical data, where the labels are determined by a set of class relevant patterns and the inputs are generated from these along with some background patterns. We prove that neural networks trained by gradient descent can succeed on these problems. The success relies on the emergence and improvement of effective features, which are learned among exponentially many candidates efficiently by exploiting the data (in particular, the structure of the input distribution). In contrast, no linear models on data-independent features of polynomial sizes can learn to as good errors. Furthermore, if the specific input structure is removed, then no polynomial algorithm in the Statistical Query model can learn even weakly. These results provide theoretical evidence showing that feature learning in neural networks depends strongly on the input structure and leads to the superior performance. Our preliminary experimental results on synthetic and real data also provide positive support.

## 1 Introduction

Various empirical studies have shown that an important characteristic of neural networks is their feature learning ability, i.e., to learn a feature mapping for the inputs which allow accurate prediction (e.g., Zeiler & Fergus (2014); Girshick et al. (2014); Zhang et al. (2019); Manning et al. (2020)). This is widely believed to be a key factor to their remarkable success in many applications, in particular, an advantage over traditional machine learning methods. To understand their success, it is then crucial to understand the source and benefit of feature learning in neural networks. Empirical observations show that networks can learn neurons that correspond to different semantic patterns in the inputs (e.g., eyes, bird shapes, tires, etc. in images (Zeiler & Fergus, 2014; Girshick et al., 2014)). Moreover, recent progress (e.g., Caron et al. (2018); Chen et al. (2020b); He et al. (2020); Jing & Tian (2020)) shows that one can even learn a feature mapping using only unlabeled inputs and then learn an accurate predictor (usually a linear function) on it using labeled data. This further demonstrates the feature learning ability of neural networks and that these input distributions contain important information for learning useful features. These empirical observations strongly suggest that the structure of the input distribution is crucial for feature learning and feature learning is crucial for the strong performance. However, it is largely unclear how practical training methods (gradient descent or its variants) learn important patterns from the inputs and whether this is necessary for obtaining the superior performance, since the empirical studies do not exclude the possibility that some other training methods can achieve similar performance without feature learning or with feature learning that does not exploit the input structure. Rigorous theoretical investigations are thus needed for answering these fundamental questions: *How can effective features emerge from inputs in the training dynamics of gradient descent? Is learning features from inputs necessary for the superior performance?*

---

[*]Equal contribution

Compared to the abundant empirical evidence, the theoretical understanding still remains largely open. One line of work (e.g. Jacot et al. (2018); Li & Liang (2018); Du et al. (2019); Allen-Zhu et al. (2019); Zou et al. (2020); Chizat et al. (2019) and many others) shows in certain regime, sufficiently overparameterized networks are approximately linear models, i.e., a linear function on the Neural Tangent Kernel (NTK). This falls into the traditional approach of linear models on fixed features, which also includes random features (Rahimi & Recht, 2008) and other kernel methods (Kamath et al., 2020). The kernel viewpoint thus does not explain feature learning in networks nor the advantage over fixed features. A recent line of work (e.g. Daniely & Malach (2020); Bai & Lee (2019); Ghorbani et al. (2020); Yehudai & Shamir (2019); Allen-Zhu & Li (2019; 2020a); Li et al. (2020); Malach et al. (2021) and others) shows examples where networks provably enjoy advantages over fixed features, under different settings and assumptions. While providing insightful results separating the two approaches, most studies have not investigated if the input structure is crucial for feature learning and thus the advantage. Also, most studies have not analyzed how gradient descent can learn important input patterns as effective features, or rely on strong assumptions like models or data atypical in practice (e.g., special networks, Gaussian data, etc).

Towards a more thorough understanding, we propose to analyze learning problems motivated by practical data, where the labels are determined by a set of class relevant patterns and the inputs are generated from these along with some background patterns. We use comparison for our study: (1) by comparing network learning approaches with fixed feature approaches on these problems, we analyze the emergence of effective features and demonstrate feature learning leads to the advantage over fixed features; (2) by comparing these problems to those with the input structure removed, we demonstrate that the input structure is crucial for feature learning and prediction performance.

More precisely, we obtain the following results. We first prove that two-layer networks trained by gradient descent can efficiently learn to small errors on these problems, and then prove that no linear models on fixed features of polynomial sizes can learn to as good errors. These two results thus establish the provable advantage of networks and implies that feature learning leads to this advantage. More importantly, our analysis reveals the dynamics of feature learning: the network first learns a rough approximation of the effective features, then improves them to get a set of good features, and finally learns an accurate classifier on these features. Notably, the improvement of the effective features in the second stage is needed for obtaining the provable advantage. The analysis also reveals the emergence and improvement of the effective features are by exploiting the data, and in particular, they rely on the input structure. To formalize this, we further prove the third result: if the specific input structure is removed and replaced by a uniform distribution, then no polynomial algorithm can even weakly learn in the Statistical Query (SQ) learning model, not to mention the advantage over fixed features. Since SQ learning includes essentially all known algorithms (in particular, mini-batch stochastic gradient descent used in practice), this implies that feature learning depends strongly on the input structure. Finally, we perform simulations on synthetic data to verify our results. We also perform experiments on real data and observe similar phenomena, which show that our analysis provides useful insights for the practical network learning.

Our analysis then provides theoretical support for the following principle: *feature learning in neural networks depends strongly on the input structure and leads to the superior performance.* In particular, our results make it explicit that learning features from the input structure is crucial for the superior performance. This suggests that input-distribution-free analysis (e.g., traditional PAC learning) may not be able to explain the practical success, and advocates an emphasis of the input structure in the analysis. While these results are for our proposed problem setting and network learning in practice can be more complicated, the insights obtained match existing empirical observations and are supported by our experiments. The compelling evidence hopefully can attract more attention to further studies on modeling the input structure and analyzing feature learning.

## 2 RELATED WORK

This section provides an overview while more technical discussions can be found in Appendix A.

**Neural Tangent Kernel (NTK) and Linearization of Neural Networks.** One line of work (e.g. Jacot et al. (2018); Li & Liang (2018); Matthews et al. (2018); Lee et al. (2019); Novak et al. (2019); Yang (2019); Du et al. (2019); Allen-Zhu et al. (2019); Zou et al. (2020); Ji & Telgarsky (2019); Cao et al. (2020); Geiger et al. (2020); Chizat et al. (2019) and more) explains the success of sufficiently

over-parameterized neural network by connecting them to linear methods like NTK. Though their approaches are different, they all base on the observation that when the network is sufficiently large, the weights stay close to the initialization during the training, and training is similar to solving a kernel method problem. This is typically referred to as the NTK regime, or lazy training, or linearization. However, networks used in practice are usually not large enough to enter this regime, and the weights are frequently observed to traverse away from the initialization. Furthermore, in this regime, network learning is essentially the traditional approach of linear methods over fixed features, which cannot establish or explain feature learning and the advantage of network learning.

**Advantage of Neural Networks over Linear Models on Fixed Features.** Since the superior network learning results via gradient descent are not well explained by the NTK view, a recent line of work has turned to learning settings where neural networks provably have advantage over linear models on fixed features (e.g. Daniely & Malach (2020); Refinetti et al. (2021); Malach et al. (2021); Dou & Liang (2020); Bai & Lee (2019); Ghorbani et al. (2020); Allen-Zhu & Li (2019); see the great summary in Malach et al. (2021)). While formally establishing the advantage, they have not thoroughly answered the two fundamental questions this work focuses on; in particular, most existing work has not studied whether the input structure is a crucial factor for feature learning and thus the advantage, and/or has not considered how the features are learned in more practical training scenarios. For example, Ghorbani et al. (2020) show the advantage of networks in approximation power and Dou & Liang (2020) show their statistical advantage, but they do not consider the learning dynamics (i.e., how the training method obtains the good network). Allen-Zhu & Li (2019) prove the advantage of the networks for PAC learning with labels given by a depth-2 ResNet and Allen-Zhu & Li (2020a) prove for Gaussian inputs with labels given by a multiple-layer network, while neither considers the influence of the input structure on feature learning or the advantage. Daniely & Malach (2020) prove the advantage of the networks for learning sparse parities on specific input distributions that help gradient descent learn effective features for prediction, and Malach et al. (2021) consider similar learning problems but with specifically designed differentiable models, while our work analyzes data distributions and models closer to those in practice and also explicitly focuses on whether the input structure is needed for the learning. There are also other theoretical studies on feature learning in networks (e.g. Yehudai & Ohad (2020); Zhou et al. (2021); Diakonikolas et al. (2020); Frei et al. (2020)), which however do not directly relate feature learning to the input structure or the advantage of network learning.

## 3 PROBLEM SETUP

To motivate our setup, consider images with various kinds of patterns like lines and rectangles. Some patterns are relevant for the labels (e.g., rectangles for distinguishing indoor or outdoor images), while the others are not. If the image contains a sufficient number of the former, then we are confident that the image belongs to a certain class. Dictionary learning or sparse coding is a classic model of such data (e.g., Olshausen & Field (1997); Vinje & Gallant (2000); Blei et al. (2003)). We thus model the patterns as a dictionary, generate a hidden vector indicating the presence of the patterns, and generate the input and label from this vector.

Let $\mathcal{X} = \mathbb{R}^d$ be the input space, and $\mathcal{Y} = \{\pm 1\}$ be the label space. Suppose $M \in \mathbb{R}^{d \times D}$ is an unknown dictionary with $D$ columns that can be regarded as patterns. For simplicity, assume $M$ is orthonormal. Let $\tilde{\phi} \in \{0, 1\}^D$ be a hidden vector that indicates the presence of each pattern. Let $A \subseteq [D]$ be a subset of size $k$ corresponding to the class relevant patterns. Then the input is generated by $M\tilde{\phi}$, and the label can be any binary function on the number of class relevant patterns. More precisely, let $P \subseteq [k]$. Given $A$ and $P$, we first sample $\tilde{\phi}$ from a distribution $\mathcal{D}_{\tilde{\phi}}$, and then generate the input $\tilde{x}$ and the class label $y$ from $\tilde{\phi}$:

$$\tilde{\phi} \sim \mathcal{D}_{\tilde{\phi}}, \quad \tilde{x} = M\tilde{\phi}, \quad y = \begin{cases} +1, & \text{if } \sum_{i \in A} \tilde{\phi}_i \in P, \\ -1, & \text{otherwise.} \end{cases} \tag{1}$$

**Learning with Input Structure.** We allow quite general $\mathcal{D}_{\tilde{\phi}}$ with the following assumptions:

- **(A0)** The class probabilities are balanced: $\Pr[\sum_{i \in A} \tilde{\phi}_i \in P] = 1/2$.
- **(A1)** The patterns in $A$ are correlated with the labels with the same correlation: for any $i \in A$, $\gamma = \mathbb{E}[y\tilde{\phi}_i] - \mathbb{E}[y]\mathbb{E}[\tilde{\phi}_i] > 0$.

**(A2)** Each pattern outside $A$ is identically distributed and independent of all other patterns. Let $p_o := \Pr[\tilde{\phi}_i = 1]$ and without loss of generality assume $p_o \le 1/2$.

Let $\mathcal{D}(A, P, \mathcal{D}_{\tilde{\phi}})$ denote the distribution on $(\tilde{x}, y)$ for some $A, P$, and $\mathcal{D}_{\tilde{\phi}}$. Given parameters $\Xi = (d, D, k, \gamma, p_o)$, the family $\mathcal{F}_\Xi$ of distributions include all $\mathcal{D}(A, P, \mathcal{D}_{\tilde{\phi}})$ with $A \subseteq [D]$, $P \subseteq [k]$, and $\mathcal{D}_{\tilde{\phi}}$ satisfying the above assumptions. The labeling function includes some interesting special cases:

*Example 1.* Suppose $P = \{i \in [k] : i > k/2\}$ for some threshold, i.e., we will set the label $y = +1$ when more than a half of the relevant patterns are presented in the input.

*Example 2.* Suppose $k$ is odd, and let $P = \{i \in [k] : i \text{ is odd}\}$, i.e., the labels are given by the parity function on $\tilde{\phi}_j (j \in A)$. This is useful to prove our lower bounds via the properties of parities.

Appendix F presents results for more general settings (e.g., incoherent dictionary, unbalanced classes, etc.). On the other hand, our problem setup does not include some important data models. In particular, one would like to model hierarchical representations often observed in practical data and believed to be important for deep learning. We leave such more general cases for future work.

**Learning Without Input Structure.** For comparison, we also consider learning problems without input structure. The data are generated as above but with different distributions $\mathcal{D}_{\tilde{\phi}}$:

**(A1')** The patterns are uniform over $\{0, 1\}^D$: for any $i \in [D], \Pr[\tilde{\phi}_i = 1] = 1/2$ independently.

Given parameters $\Xi_0 = (d, D, k)$, the family $\mathcal{F}_{\Xi_0}$ of distributions without input structure is the set of all the distributions with $A \subseteq [D]$, $P \subseteq [k]$ and $\mathcal{D}_{\tilde{\phi}}$ satisfying the above assumptions.

## 3.1 Neural Network Learning

**Networks.** We consider training a two-layer network via gradient descent on the data distribution:

$$g(x) = \sum_{i=1}^{2m} a_i \sigma(\langle w_i, x \rangle + b_i) \tag{2}$$

where $w_i \in \mathbb{R}^d, b_i, a_i \in \mathbb{R}$, and $\sigma(z) = \min(1, \max(z, 0))$ is the truncated rectified linear unit (ReLU) activation function. Let $\theta = \{w_i, b_i, a_i\}_{i=1}^{2m}$ denote all the parameters, and let superscript $(t)$ denote the time step, e.g., $g^{(t)}$ denote the network at time step $t$ with $\theta^{(t)} = \{w_i^{(t)}, b_i^{(t)}, a_i^{(t)}\}$.

**Loss Function.** Similar to typical practice, we will normalize the data for learning: first compute $x = (\tilde{x} - \mathbb{E}[\tilde{x}])/\tilde{\sigma}$ where $\tilde{\sigma}^2 = \mathbb{E} \sum_{i=1}^{d} (\tilde{x}_i - \mathbb{E}[\tilde{x}_i])^2$ is the variance of the data, and then train on $(x, y)$. This is equivalent to setting $\phi = (\tilde{\phi} - \mathbb{E}[\tilde{\phi}])/\tilde{\sigma}$ and generating $x = M\phi$. For $(\tilde{x}, y)$ from $\mathcal{D}$ and the normalized $(x, y)$, we will simply say $(x, y) \sim \mathcal{D}$.

For the training, we consider the hinge-loss $\ell(y, \hat{y}) = \max\{1 - y\hat{y}, 0\}$. We will inject some noise $\xi$ to the neurons for the convenience of the analysis. (This can be viewed as using a smoothed version of the activation $\tilde{\sigma}(z) = \mathbb{E}_\xi \sigma(z + \xi)$ similar to those in existing studies like Allen-Zhu & Li (2020b); Malach et al. (2021). See Section 5 for more explanations.) Formally, the loss is:

$$L_\mathcal{D}(g; \sigma_\xi) = \mathbb{E}_{(x,y)}[\ell(y, g(x; \xi))], \text{ where } g(x; \xi) = \sum_{i=1}^{2m} a_i \mathbb{E}_\xi[\sigma(\langle w_i, x \rangle + b_i + \xi_i)] \tag{3}$$

where $\xi \sim \mathcal{N}(0, \sigma_\xi^2 I_{m \times m})$ are independent Gaussian noise. Let $L_\mathcal{D}(g)$ denote the typical hinge-loss without noise. We also consider $\ell_2$ regularization: $R(g; \lambda_a, \lambda_w) = \sum_{i=1}^{2m} \lambda_a |a_i|^2 + \lambda_w \|w_i\|_2^2$ with regularization coefficients $\lambda_a, \lambda_w$.

**Training Process.** We first perform an unbiased initialization: for every $i \in [m]$, initialize $w_i^{(0)} \sim \mathcal{N}(0, \sigma_w^2 I_{d \times d})$ with $\sigma_w = 1/k$, $b_i^{(0)} \sim \mathcal{N}(0, \sigma_b^2)$ with $\sigma_b = 1/k^2$, $a_i^{(0)} \sim \mathcal{N}(0, \sigma_a^2)$ with $\sigma_a = \tilde{\sigma}^2/(\gamma k^2)$, and then set $w_{m+i}^{(0)} = w_i^{(0)}, b_{m+i}^{(0)} = b_i^{(0)}, a_{m+i}^{(0)} = -a_i^{(0)}$. We then do gradient updates:

$$\theta^{(t)} = \theta^{(t-1)} - \eta^{(t)} \nabla_\theta \left( L_\mathcal{D}(g^{(t-1)}; \sigma_\xi^{(t)}) + R(g^{(t-1)}; \lambda_a^{(t)}, \lambda_w^{(t)}) \right), \text{ for } t = 1, 2, \dots, T, \tag{4}$$

for some choice of the hyperparameters $\eta^{(t)}, \lambda_a^{(t)}, \lambda_w^{(t)}, \sigma_\xi^{(t)}$, and $T$.

## 4 MAIN RESULTS

**Provable Guarantee for Neural Networks.** The network learning has the following guarantee:

**Theorem 1.** *For any $\delta, \epsilon \in (0,1)$, if $k = \Omega\left(\log^2(D/(\delta\gamma))\right)$, $p_o = \Omega(k^2/D)$, and $\max\{\Omega(k^{12}/\epsilon^{3/2}), D\} \le m \le poly(D)$, then with properly set hyperparameters, for any $\mathcal{D} \in \mathcal{F}_\Xi$, with probability at least $1 - \delta$, there exists $t \in [T]$ such that $\Pr[\text{sign}(g^{(t)}(x)) \ne y] \le L_\mathcal{D}(g^{(t)}) \le \epsilon$.*

The theorem shows that for a wide range of the background pattern probability $p_o$ and the number of class relevant patterns $k$, the network trained by gradient descent can obtain a small classification error. More importantly, the analysis shows the success comes from feature learning. In the early stages, the network learns and improves the neuron weights such that on the features (i.e., the neurons' outputs) there is an accurate classifier; afterwards it learns such a classifier. The next section will provide a detailed discussion on the feature learning.

**Lower Bound for Fixed Features.** Empirical observations and Theorem 1 do not exclude the possibility that some methods without feature learning can achieve similar performance. We thus prove a lower bound for the fixed feature approach, i.e., linear models on data-independent features.

**Theorem 2.** *Suppose $\Psi$ is a data-independent feature mapping of dimension $N$ with bounded features, i.e., $\Psi : \mathcal{X} \to [-1,1]^N$. For $B > 0$, the family of linear models on $\Psi$ with bounded norm $B$ is $\mathcal{H}_B = \{h(\tilde{x}) : h(\tilde{x}) = \langle \Psi(\tilde{x}), w \rangle, \|w\|_2 \le B\}$. If $3 < k \le D/16$ and $k$ is odd, then there exists $\mathcal{D} \in \mathcal{F}_\Xi$ such that all $h \in \mathcal{H}_B$ have hinge-loss at least $p_o\left(1 - \frac{\sqrt{2N}B}{2^k}\right)$.*

So using *fixed* features independent of the data cannot get loss nontrivially smaller than $p_o$ unless with exponentially large models. In contrast, viewing the neurons $\sigma(\langle w_i, x \rangle + b_i)$ as *learned* features, network learning can achieve any loss $\epsilon \in (0,1)$ with models of polynomial sizes. We emphasize the lower bound is because the feature map $\Psi$ is independent of the data. Indeed, there exists a small linear model on a small dimensional feature map allowing 0 loss for each data distribution in our problem set $\mathcal{F}_\Xi$ (Lemma 5). However, this feature map $\Psi^*$ is different for different data distribution in $\mathcal{F}_\Xi$, i.e., depends on the data. On the other hand, the feature map $\Psi$ in the lower bound is data-independent, i.e., fixed before seeing the data. For $\Psi$ to work simultaneously for all distributions in $\mathcal{F}_\Xi$, it needs to have exponential dimensions. Intuitively, it needs a large number of features, so that there are some features to approximate each $\Psi_i^*$. There are exponentially many data distributions in $\mathcal{F}_\Xi$, and thus exponentially many data-dependent features $\Psi_i^*$, which requires $\Psi$ to have an exponentially large dimension. Network learning updates the hidden neurons using the data and can learn to move the features to the right positions to approximate the ground-truth data-dependent features $\Psi^*$, so it does not need an exponentially large dimension feature map.

The theorem directly applies to linear models on fixed finite-dimensional feature maps, e.g., linear models on the input or random feature approaches (Rahimi & Recht, 2008). It also implies lower bounds to infinite dimensional feature maps (e.g., some kernels) that can be approximated by feature maps of polynomial dimensions. For example, Claim 1 in Rahimi & Recht (2008) implies that a function $f$ using shift-invariant kernels (e.g., RBF) can be approximated by a model $\langle \Psi(\tilde{x}), w \rangle$ with the dimension $N$ and weight norm $B$ bounded by polynomials of the related parameters of $f$ like its RKHS norm and the input dimension. Then our theorem implies some related parameter of $f$ needs to be exponential in $k$ for $f$ to get nontrivial loss, formalized in Corollary 3. Kamath et al. (2020) has more discussions on approximating kernels with finite dimensional maps.

**Corollary 3.** *For any function $f$ using a shift-invariant kernel $K$ with RKHS norm bounded by $L$, or $f(x) = \sum_i \alpha_i K(z_i, x)$ for some data points $z_i$ and $\|\alpha\|_2 \le L$. If $3 < k \le D/16$ and $k$ is odd, then there exists $\mathcal{D} \in \mathcal{F}_\Xi$ such that $f$ has hinge-loss at least $p_o\left(1 - \frac{poly(d,L)}{2^k}\right) - \frac{1}{poly(d,L)}$.*

**Lower Bound for Without Input Structure.** Existing results do not exclude the possibility that some learning methods without exploiting the input structure can achieve strong performance. To show the necessity of the input structure, we consider learning $\mathcal{F}_{\Xi_0}$ with input structure removed. We obtain a lower bound for such learning problems in the classic Statistical Query (SQ) model (Kearns, 1998). In this model, the algorithm can only receive information about the data through statistical queries. A statistical query is specified by some polynomially-computable property predicate $Q$ of labeled instances and a tolerance parameter $\tau \in [0,1]$. For a query $(Q, \tau)$, the algorithm receives

a response $\hat{P}_Q \in [P_Q - \tau, P_Q + \tau]$, where $P_Q = \Pr[Q(x, y)$ is true]. Notice that a query can be simulated using the average of roughly $O(1/\tau^2)$ random data samples with high probability. The SQ model captures almost all common learning algorithms (except Gaussian elimination) including the commonly used mini-batch SGD, and thus is suitable for our purpose.

**Theorem 4.** *For any algorithm in the Statistical Query model that can learn over $\mathcal{F}_{\Xi_0}$ to classification error less than $\frac{1}{2} - \frac{1}{\left(\frac{D}{k}\right)^3}$, either the number of queries or $1/\tau$ must be at least $\frac{1}{2}\left(\frac{D}{k}\right)^{1/3}$.*

The theorem shows that without the input structure, polynomial algorithms in the SQ model cannot get a classification error nontrivially smaller than random guessing. The comparison to the result for with input structure then shows that the input structure is crucial for network learning, in particular, for achieving the advantage over fixed feature models.

## 5 PROOF SKETCHES

Here we provide the sketch of our analysis, focusing on the key intuition and discussing some interesting implications. The complete proofs are included in Appendix B-D.

### 5.1 PROVABLE GUARANTEES OF NEURAL NETWORKS

**Overall Intuition.** We first show that there is a two-layer network that can represent the target labeling function, whose neurons can be viewed as the "ground-truth" features to be learned. We then show that after the first gradient step, the hidden neurons of the trained network become close to the ground-truth: their weights contain large components along the class relevant patterns but small along the background patterns. We further show that in the second gradient step, these features get improved: the "signal-noise" ratio between the components for class relevant patterns and those for the background ones becomes larger, giving a set of good features. Finally, we show that the remaining steps learn an accurate classifier on these features.

**Existence of A Good Network.** We show that there is a two-layer network that can fit the labels.

**Lemma 5.** *For any $\mathcal{D} \in \mathcal{F}_\Xi$, there exists a network $g^*(x) = \sum_{i=1}^n a_i^* \sigma(\langle w_i^*, x \rangle + b_i^*)$ with $y = g^*(x)$ for any $(x, y) \sim \mathcal{D}$. Furthermore, the number of neurons $n = 3(k + 1)$, $|a_i^*| \leq 32k$, $1/(32k) \leq |b_i^*| \leq 1/2$, $w_i^* = \tilde{\sigma} \sum_{j \in A} M_j/(4k)$, and $|\langle w_i^*, x \rangle + b_i^*| \leq 1$ for any $i \in [n]$ and $(x, y) \sim \mathcal{D}$.*

In particular, the weights of the neurons are proportional to $\sum_{j \in A} M_j$, the sum of the class relevant patterns. We thus focus on analyzing how the network learns such neuron weights.

**Feature Emergence in the First Gradient Step.** The gradient for $w_i$ (ignoring the noise) is:

$$\frac{\partial L_\mathcal{D}(g)}{\partial w_i} = -a_i \mathbb{E}_{(x,y)\sim\mathcal{D}} \{y\mathbb{I}[yg(x) \leq 1]\sigma'[\langle w_i, x \rangle + b_i]x\} = -a_i \mathbb{E}_{(x,y)\sim\mathcal{D}} \{yx\sigma'[\langle w_i, x \rangle + b_i]\}$$

where the last step is due to $g(x) = 0$ by the unbiased initialization. Let $q_j = \langle M_j, w_i \rangle$ denote the component along the direction of the pattern $M_j$. Then the component of the gradient on $M_j$ is:

$$\left\langle M_j, \frac{\partial}{\partial w_i} L_\mathcal{D}(g) \right\rangle = -a_i \mathbb{E}\{y\phi_j\sigma'[\langle w_i, x \rangle + b_i]\} = -a_i \mathbb{E}\left\{ y\phi_j\sigma'\left[\sum_{\ell \in [D]} \phi_\ell q_\ell + b_i\right]\right\}.$$

The key intuition is that with the randomness of $\phi_\ell$ (and potentially that of the injected noise $\xi$), the random variable under $\sigma'$ is not significantly affected by a small subset of $\phi_\ell q_\ell$. For example, for class relevant patterns $j \in A$, let $\mathbb{I}_{[D]} := \sigma'\left[\sum_{\ell \in [D]} \phi_\ell q_\ell + b_i\right]$ and $\mathbb{I}_{-A} := \sigma'\left[\sum_{\ell \notin A} \phi_\ell q_\ell + b_i\right]$. We have $\mathbb{I}_{[D]} \approx \mathbb{I}_{-A}$ and thus:

$$\left\langle M_j, \frac{\partial}{\partial w_i} L_\mathcal{D}(g) \right\rangle \propto \mathbb{E}\{y\phi_j\mathbb{I}_{[D]}\} \approx \mathbb{E}\{y\phi_j\mathbb{I}_{-A}\} = \mathbb{E}\{y\phi_j\}\mathbb{E}[\mathbb{I}_{-A}] = \frac{\gamma}{\tilde{\sigma}}\mathbb{E}[\mathbb{I}_{-A}]$$

since $y$ only depends on $\phi_j(j \in A)$. Then the gradient has a nontrivial component along the pattern. Similarly, for background patterns $j \notin A$, the component of the gradient along $M_j$ is close to 0.

**Lemma 6** (Informal). *Assume $p_o, k$ as in Theorem 1 and $\sigma_\xi^{(1)} < 1/k$, then with high probability $\frac{\partial}{\partial w_i} L_\mathcal{D}(g^{(0)}; \sigma_\xi^{(1)}) = -a_i^{(0)} \sum_{j=1}^D M_j T_j$ where for a small $\epsilon_e$:*

- *if $j \in A$, then $|T_j - \beta\gamma/\tilde\sigma| \leq O(\epsilon_e/\tilde\sigma)$ with $\beta \in [\Omega(1), 1]$;*

- *if $j \notin A$, then $|T_j| \leq O(\sigma_\phi^2 \epsilon_e \tilde\sigma)$.*

By setting $\lambda_w^{(1)} = 1/(2\eta^{(1)})$, we have $w_i^{(1)} = \eta^{(1)} a_i^{(0)} \sum_{j=1}^D M_j T_j \approx \eta^{(1)} a_i^{(0)} \frac{\beta\gamma}{\tilde\sigma} \sum_{j \in A} M_j$. For small $p_o$, e.g., $p_o = \tilde{O}(k^2/D)$, these neurons can already allow accurate prediction. However, for such small $p_o$, we cannot show a provable advantage of networks over fixed features. On the other hand, for larger $p_o$ meaning a significant number of background patterns in the input, the approximation error terms $T_j (j \notin A)$ together can overwhelm the signals $T_j (j \in A)$ and lead to bad prediction, even though each term is small. Fortunately, we will show that the second gradient step can improve the weights by decreasing the ratio between $T_j (j \notin A)$ and $T_j (j \in A)$.

**Feature Improvement in the Second Gradient Step.** We note that by setting a small $\eta^{(1)}$, after the update we still have $yg(x; \xi) < 1$ for most $(x, y) \sim \mathcal{D}$ and thus the gradient in the second step is:

$$\frac{\partial}{\partial w_i} L_\mathcal{D}(g; \sigma_\xi) \approx -a_i \mathbb{E}_{(x,y)\sim\mathcal{D}} \left\{ yx \mathbb{E}_\xi \sigma'[\langle w_i, x \rangle + b_i + \xi_i] \right\}.$$

We can then follow the intuition for the first step again. For $j \in A$, the component $\langle M_j, \frac{\partial}{\partial w_i} L_\mathcal{D}(g) \rangle$ is roughly proportional to $\frac{\gamma}{\tilde\sigma} \mathbb{E}[\mathbb{I}_{-A,\xi}]$ where $\mathbb{I}_{-A,\xi} := \sigma' \left[ \sum_{\ell \notin A} \phi_\ell q_\ell + b_i + \xi_i \right]$. While $\phi_\ell q_\ell$ may not have large enough variance, the injected noise $\xi_i$ makes sure that a nontrivial amount of data activate the neuron.[1] Then $\mathbb{I}_{-A,\xi} \neq 0$, leading to a nontrivial component along $M_j$, similar to the first step. On the other hand, for $j \notin A$, the approximation error term $T_j$ depends on how well $\sigma' \left[ \sum_{\ell \notin A, \ell \neq j} \phi_\ell q_\ell + b_i + \xi_i \right]$ approximates $\sigma' \left[ \sum_{\ell \in [D]} \phi_\ell q_\ell + b_i + \xi_i \right]$. Since the $q_\ell$'s (the weight's component along $M_\ell$) in the second step are small compared to those in the first step, we can then get a small error term $T_j$. So the ratio between $T_j (j \notin A)$ over $T_j (j \in A)$ improves after the second step, giving better features allowing accurate prediction.

**Classifier Learning Stage.** Given the learned features, we are then ready to show the remaining gradient steps can learn accurate classifiers. Intuitively, with small hyperparameter values ($\eta^{(t)} = \frac{k^2}{Tm^{1/3}}, \lambda_a^{(t)} = \lambda_w^{(t)} \leq \frac{k^3}{\tilde\sigma m^{1/3}}, \sigma_\xi^{(t)} = 0$ for $2 < t \leq T = m^{4/3}$), the first layer's weights do not change too much and thus the learning is similar to convex learning using the learned features. Formally, our proof uses the online convex optimization technique in Daniely & Malach (2020).

## 5.2 Lower Bounds

The lower bounds are based on the following observation: our problem setup is general enough to include learning sparse parity functions. Consider an odd $k$, and let $P = \{i \in [k] : i \text{ is odd}\}$. Then $y$ is given by $\Pi_A(z) := \prod_{j \in A} z_j$ for $z_j = 2\tilde\phi_j - 1$, i.e., the parity function on $z_j (j \in A)$. Then known results for learning parity functions can be applied to prove our lower bounds.

**Lower Bound for Fixed Features.** We show that $\mathcal{F}_\Xi$ contains learning problems that consist of a mixture of two distributions with weights $p_o$ and $1 - p_o$ respectively, where in the first distribution $\mathcal{D}_A^{(1)}, \tilde x$ is given by the uniform distribution over $\tilde\phi$ and the label $y$ is given by the parity function on $A$. On such $\mathcal{D}_A^{(1)}$, Daniely & Malach (2020) shows that exponentially large models over fixed features is needed to get nontrivial loss. Intuitively, there are exponentially many labeling functions $\Pi_A$ that are uncorrelated (i.e., "orthogonal" to each other): $\mathbb{E}[\Pi_{A_1}\Pi_{A_2}] = 0$ for any $A_1$ and $A_2$. Note that the best approximation of $\Pi_A$ by a fixed set of features $\Psi_i$'s is its projection on the linear span of the features. Then with polynomial-size models, there always exists some $\Pi_A$ far from the linear span.

---

[1]Equivalently, the network uses $\tilde\sigma(z) = \mathbb{E}_\xi \sigma(z + \xi)$, a Gaussian smoothed version of $\sigma$, and the smoothing allows $z$ slightly outside the activated region of $\sigma$ to generate gradient for the learning. Empirically it is not needed since typically sufficient data can activate the neurons. One potential reason is that the data have their own noise to achieve a similar effect (a remote analog being noisy gradients can help the optimization). Further analysis on such an effect is left for future work.

*Remark.* It is instructive to compare to network learning, which finds the effective weights $\sum_{j \in A} M_j$ among the exponentially many candidates corresponding to different $A$'s. This can be done efficiently by exploiting the data since the gradient is roughly proportional to $\mathbb{E}\{yx\} = \sum_{j \in A} M_j$. The network then learns *data-dependent* features on which polynomial size linear models can achieve small loss.

**Lower Bound for Learning without Input Structure.** Clearly, $\mathcal{F}_{\Xi_0}$ contains the distributions $\mathcal{D}_A^{(1)}$ described above. The lower bound then follows from classic SQ learning results (Blum et al., 1994).

*Remark.* The SQ lower bound analysis does not apply to $\mathcal{F}_\Xi$, because in $\mathcal{F}_\Xi$ the input distribution is related the labeling function. This allows networks to learn with polynomial time/sample. While both the labeling function and the input distribution affect the learning, few existing studies explicitly point out the importance of the input structure. We thus emphasize the input structure is crucial for networks to learn effective features and achieve superior performance.

## 6 Experiments

Our experiments mainly focus on feature learning and the effect of the input structure. We first perform simulations on our learning problems to (1) verify our main theorems on the benefit of feature learning and the effect of input structure; (2) verify our analysis of feature learning in networks. We then check if our insights carry over to real data: (3) whether similar feature learning is presented in real network/data; (4) whether damaging the input structure lowers the performance. The results are consistent with our analysis and provide positive support for the theory. Below we present part of the results and include the complete experimental details and results in Appendix E.

**Simulation: Verification of the Main Results.** We generate data according to our problem setup, with $d = 500, D = 100, k = 5, p_o = 1/2$, a randomly sampled $A$, and labels given by the parity function. We then train a two-layer network with $m = 300$ following our learning process, and for comparison, we also use two fixed feature methods (the NTK and random feature methods based on the same network). Finally, we also use these three methods on the data distribution with the input structure removed (i.e., $\mathcal{F}_{\Xi_0}$ in Theorem 4).

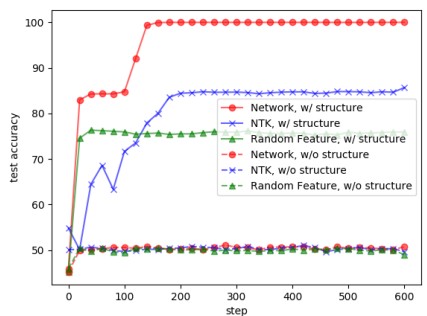

Figure 1: Test accuracy on simulated data with or without input structure.

Figure 1 shows that the results are consistent with our results. Network learning gets high test accuracy while the two fixed feature methods get significantly lower accuracy. Furthermore, when the input structure is removed, all three methods get test accuracy similar to random guessing.

**Simulation: Feature Learning in Networks.** We compute the cosine similarities between the weights $w_i$'s and visualize them by Multidimensional Scaling. (Recall that our analysis is on the *directions* of the weights without considering their *scaling*, and thus it is important to choose cosine similarity rather than say the typical Euclidean distance.) Figure 2 shows that the results are as predicted by our analysis. After the first gradient step, some weights begin to cluster around the ground-truth $\sum_{j \in A} M_j$ (or $-\sum_{j \in A} M_j$ due to the $a_i$ in the gradient update which can be positive or negative). After the second step, the weights get improved and well-aligned with the ground-truth (with cosine similarities $> 0.99$). Furthermore, if a classifier is trained on the features after the first step, the test accuracy is about $52\%$; if the same is done after the second step, the test accuracy is about $100\%$. This demonstrates while some effective features emerge in the first step, they need to be improved in the second step to get accurate prediction.

**Real Data: Feature Learning in Networks.** We perform experiments on MNIST (LeCun et al., 1998; Deng, 2012), CIFAR10 (Krizhevsky, 2012), and SVHN (Netzer et al., 2011). On MNIST, we train a two-layer network with $m = 50$ on the subset with labels 0/1 and visualize the neurons' weights as in the simulation. Figure 3 shows a similar feature learning phenomenon: effective features emerge after a few steps and then get improved to form two clusters. Similar results are observed on other datasets. These suggest the insights obtained in our analysis are also applicable to the real data.

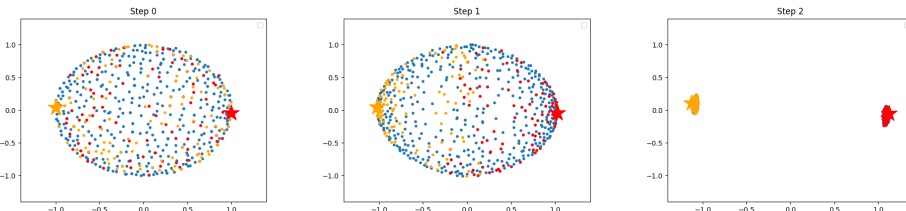

Figure 2: Visualization of the weights $w_i$'s after initialization/one gradient step/two steps in network learning on the synthetic data. The red star denotes the ground-truth $\sum_{j \in A} M_j$; the orange star is $-\sum_{j \in A} M_j$. The red/orange dots are the weights closest to the red/orange star, respectively.

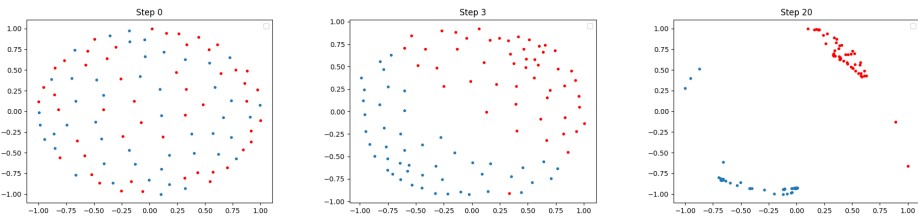

Figure 3: Visualization of the neurons' weights in a two-layer network trained on the subset of MNIST data with label 0/1. The weights gradually form two clusters.

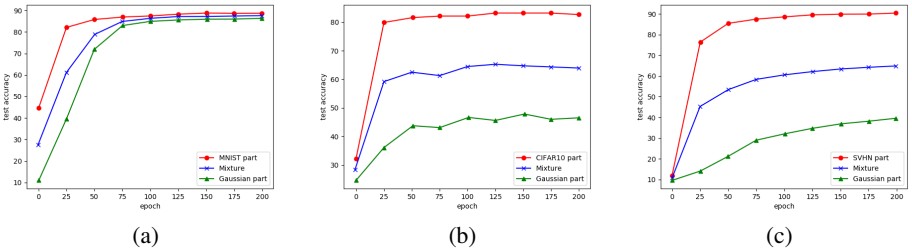

Figure 4: Test accuracy at different steps for an equal mixture of Gaussian inputs with data: (a) MNIST, (b) CIFAR10, (c) SVHN.

**Real Data: The Effect of Input Structure.** Since we cannot directly manipulate the input distribution of real data, we perform controlled experiments by injecting different inputs. For labeled dataset $\mathcal{L}$ and injected input $\mathcal{U}$, we first train a teacher network fitting $\mathcal{L}$, then use the teacher network to give labels on a mixture of inputs from $\mathcal{L}$ and $\mathcal{U}$, and finally train a student network on this new dataset $\mathcal{M}$ consisting of the mixed inputs and the teacher network's labels. Checking the student' performance on different parts of $\mathcal{M}$ and comparing to those by directly training the student on the original data $\mathcal{L}$ can reveal the impact of changing the input structure. We use MNIST, CIFAR10, or SVHN as $\mathcal{L}$, and use Gaussian or images in Tiny ImageNet (Le & Yang, 2015) as $\mathcal{U}$. The networks for MNIST are two-layer with $m = 9$, and those for CIFAR10/SVHN are ResNet-18 convolutional neural networks (He et al., 2016).

Figure 4 shows the results on an equal mixture of data and Gaussian. It presents the test accuracy of the student on the original data part, the Gaussian part, and the whole mixture. For example, on CIFAR10, the network learns well over the CIFAR10 part (with accuracy similar to directly training on the original data) but learns slower with worse accuracy on the Gaussian part. Furthermore, the accuracy on the whole mixture is lower than that of training on the original CIFAR10. This shows that the input structure indeed has a significant impact on the learning. While MNIST+Gaussian shows a less significant trend (possibly because the tasks are simpler), the other datasets show similar significant trends as CIFAR10+Gaussian (the results using Tiny ImageNet are in the appendix).

## 7 ETHICS STATEMENT

Our paper is mostly theoretical in nature and thus we foresee no immediate negative ethical impact. We are of the opinion that our theoretical framework may lead to better understanding and inspire development of improved network learning methods, which may have a positive impact in practice. In addition to the theoretical machine learning community, we perceive that our conceptual message that the input structure is crucial for the network learning's performance can be beneficial to engineering-inclined machine learning researchers.

## 8 REPRODUCIBILITY STATEMENT

For theoretical results in the Section 4, a complete proof is provided in the Appendix B-D. The theoretical results and complete proofs for a setting more general than that in the main text are provided in the Appendix F. For experiments in the Section 6, complete details and experimental results are provided in the Appendix Section E. The source code with explanations and comments is provided in the supplementary material.

## 9 ACKNOWLEDGEMENT

The work is partially supported by Air Force Grant FA9550-18-1-0166, the National Science Foundation (NSF) Grants 2008559-IIS and CCF-2046710.

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

# Appendix

Section A presents more technical discussion on related work. Section B-D provides the complete proofs for our results in the main text. Section E provides the complete details and experimental results for our experiments.

Finally, Section F provides the theoretical results and complete proofs for a setting more general than that in the main text, allowing incoherent dictionaries, unbalanced classes, and Gaussian noise in the data.

## CONTENTS

## A    MORE TECHNICAL DISCUSSION ON RELATED WORK

**Advantage of Neural Networks over Linear Models on Fixed Features.** A recent line of work has turned to show learning settings where network learning provably has advantage over linear models on fixed features; see the nice summary in Malach et al. (2021). Here we highlight the results and focuses of the existing related studies and discuss the differences from ours.

Yehudai & Shamir (2019) shows that the random feature method fails to learn even a single ReLU neuron on Gaussian inputs unless its size is exponentially large in dimension. This points out the limitation of the random feature method (belonging to the fixed feature approach) but does not consider feature learning in networks.

Some studies show that a single ReLU neuron can be learnt by gradient descent (Yehudai & Ohad, 2020; Diakonikolas et al., 2020; Frei et al., 2020). The analysis typically involves feature learning. However, their focus is different: they do not show the advantage over fixed feature methods and do not consider the effect of the input structures.

Zhou et al. (2021) shows that in a special teacher-student setting, the student network will do exact local convergence in a surprising way that all student neurons will converge to one of the teacher neurons. The work does not consider the effect of the input structure nor the advantage over fixed features.

Dou & Liang (2020) explains the advantage of network learning by constructing adaptive Reproducing Kernel Hilbert Space (RKHS) indexed by the training process of the neural network, and shows that adaptive RKHS benefits from a smaller function space containing the residue comparing to RKHS. The work shows the statistical advantage of networks over data-independent kernels, but does not consider the optimization for learning the network.

Ghorbani et al. (2020) considers data generated from a hidden vector with two subsets of variables, each uniformly distributed in a high-dimensional sphere (with a different radius), while the label is determined by only the first subset of variables. It shows the existence of good neural networks that can overcome the curse of dimensionality by representing the best low-dimensional hidden structure. However, it studies the approximation power of neural networks rather than the learning, i.e., it does not show how to learn the good network.

Fang et al. (2019) argues that in the infinite width limit, a two-layer neural network will learn a nearly optimal feature representation in the distribution sense, thanks to the convexity of the limit problem. It is unclear how this result helps to understand the feature learning procedure for practical networks, which is usually a non-convex process.

Chen et al. (2020a) considers a fixed, randomly initialized neural network as a representation function fed into another trainable network which is the quadratic Taylor model of a wide two-layer network. It shows that learning over the random representation can achieve improved sample complexities compared to learning over the raw data. However, the representation considered is not learned, which is different from our focus on feature learning.

Allen-Zhu & Li (2020a) considers Gaussian inputs with labels given by a multiple-layer network with quadratic activations and skip connections (with the assumption of information gap on the weights), and studies training a deep network with quadratic activation. It shows that the trained network can learn proper representations and obtain small errors while no polynomial fixed feature methods can. On the other hand, it does not focus on the influence of input structure on feature learning: note that its input distribution contains no information about the "ground-truth" features in the target network. It also points out that the learned features get improved during training: higher-level layers will help lower-level layers to improve by backpropagating correction signals. Our analysis also shows feature improvement which however is by signals from the input distribution.

Allen-Zhu & Li (2019) considers PAC learning with labels given by a depth-2 ResNet, and studies training an overparameterized depth-2 ResNet (using uniform inputs over Boolean hypercube as an example). It shows the trained network can obtain small errors while no polynomial kernel methods can obtain as good errors. Similar to Allen-Zhu & Li (2020a), it does not focus on the influence of input structure on feature learning or the advantage of networks.

Allen-Zhu & Li (2020c) studies how ensemble of deep learning models can improve test accuracy and how the ensemble can be distilled into a single model. It develops a theory which assumes the data has multi-view structure and shows that the ensemble of independently trained networks can provably improve test accuracy and the ensemble can also be provably distilled into a single model. The analysis also relies on showing that the data structure can help the ensemble and the distillation. On the other hand, their focus is on ensembles and is quite different from ours: the analysis is on showing the multi-view input structure allows the ensembles of networks to improve over single ones and ensembles of fixed feature mappings do not have improvement. While our focus is on supervisedly learning one single network that outperforms the fixed feature approaches.

Daniely & Malach (2020) considers the task of learning sparse parities with two-layer networks, and the analysis suggests that the ability to learn the label-correlated features also seems to be critical towards the success of neural networks, although the authors did not explore much in this direction. Malach et al. (2021) also considers similar learning problems but with specifically designed models for the problems. The learning problems considered in Daniely & Malach (2020); Malach et al. (2021) have input distributions that leak information about the target labeling function, which is similar to our setting, and their analysis also shows that the first gradient descent can learn a set of good features and later steps can learn an accurate classifier on top. Our work is inspired by their studies, while there are some important differences. First, their focuses are different from ours. Daniely & Malach (2020) focuses on showing neural networks can learn targets (i.e., $k$-parity functions) that are inherently non-linear. Our analysis generalizes to more general distributions, including practically motivated ones. Malach et al. (2021) focuses on strong separations between learning with gradient descent on differentiable models (including typical neural networks) and learning using the corresponding tangent kernels. The analysis is on specific differentiable models, while our work is on two-layer neural networks similar to practical ones. Second, our analysis relies on the feature improvement in the second gradient step. This is not an artifact of the analysis but comes from our problem setup. While in Daniely & Malach (2020) the data distribution allows some neurons to be sufficiently good after the first gradient step and needs no feature improvement, our setup is more general where the data distribution may not have a similar strong benign effect and thus needs feature improvement in the second gradient step.

Most related to our work is Daniely & Malach (2020). Therefore, we provide a detailed discussion to highlight the connections and differences.

1. Our problem setting is *more general* than that in Daniely & Malach (2020). To see this, let our dictionary be the identity matrix, the set $P$ to be the odd numbers (i.e., the labeling function is a sparse parity). Furthermore, let the distribution of the hidden representation be an equal mixture of the following two:

   (a) $\mathcal{D}_1$: Uniform distribution over the hypercube.

   (b) $\mathcal{D}_2$: Irrelevant patterns $\tilde{\phi}_j (j \notin A)$ have appearance probability $p_0 = 1/2$. And the distribution of relevant patterns $\tilde{\phi}_j (j \in A)$ is: all 0's with probability 1/2, and all 1's with probability 1/2.

   Then our problem setting reduces to their setting (up to scaling/translation of $\tilde{\phi}_j$'s). On the other hand, in general our setting allows for more choices for the labeling, the dictionary, and the distributions over $\tilde{\phi}$.

2. Upper bound: Because of the more general setting, our upper bound proof requires *technical novelty*. Recall that in their work, the input distribution is essentially a mixture of $\mathcal{D}_1$ and $\mathcal{D}_2$ above. In $\mathcal{D}_2$, the relevant patterns $\tilde{\phi}_j (j \in A)$ have the specific structure of all 0's or all 1's with probability 1/2. This allows to show that neurons with weight $w$ satisfying $\sum_{j \in A} w_j = 0$ will have good gradients: small components from irrelevant patterns (their Lemma 7) and large components from relevant patterns (their Lemma 8). However, in our setting, the relevant patterns do not have this specific structure, and thus their proof technique is not applicable (or can be applied only when we have an exponentially large number of hidden neurons so that some hit the good positions at random initialization). What we showed is that the gradient has some correlation with the good feature direction. So after the first gradient step, the neuron weights are not good yet but are in a better position for further improvement (in particular, their setting corresponds to $p_0 = 1/2$ which means large noise in the weights after the first step; see discussion after our Lemma 6 in Section 5). Then

the latter gradient steps are able to improve the weights to better "signal-to-noise-ratio". In summary, our proof does not rely on their specific input structure or an exponentially large number of hidden neurons for hitting some good positions. The key is that the good feature will emerge with the help of the input structure, and once in a better position, the neurons' weights can be improved to the desired quality.

3. Lower bound: On the other hand, our lower bound is proved by a reduction to the lower bound results in Daniely & Malach (2020). They have shown that $\mathcal{D}_1$ above can lead to large errors for fixed feature models of polynomial size. Our proof is essentially constructing a mixture of $\mathcal{D}_1$ and $\mathcal{D}_2$ with mixture weights $p_0$ and $(1 - p_0)$, and applying their lower bound for $\mathcal{D}_1$. See our proof in Appendix C.

4. Conceptually, our work belongs to the same line of research as Daniely & Malach (2020), to analyze how feature learning leads to the superior performance of networks. While their analysis also relies on feature learning from good gradients induced by input structure, their focus is more on separating network learning and fixed feature models and has not explicitly explored the impact of input structures (while we agree that such an explicit study will not be difficult in their setting). More importantly, their input distribution is specific and atypical in practice, which allows a specific type of feature learning (as explained in the above discussion on upper bounds). Our work thus considers a more general setting that is motivated by practical problems. Our results then bring theoretical insights closer for explaining the feature learning in practice and provide some positive evidence for the importance of analysis under proper models of the input distributions.

**Sparse Coding and Subspace Data Models.** To analyze neural networks' performance, various data models have been considered. A practical way to model the underlying structure of data is by assuming that a set of hidden variables exists and the input data is a high dimensional projection of the hidden vector (possibly with noise). Along this line, the classic sparse coding model has been used in existing works for analyzing networks. Koehler & Risteski (2018) considers such a data distribution where the label is given by a linear function on the hidden sparse vector, but studies the approximation power of networks and classic polynomial methods rather than the learning. Allen-Zhu & Li (2020b) considers similar data distributions, but studies the performance of networks under adversarial perturbations. Another type of related data models assumes that the label is determined by a subset of hidden variables. Ghorbani et al. (2020) considers a hidden vector with two subsets of variables, each uniformly distributed in a high-dimensional sphere (with a different radius), while the label is determined by only the first subset of variables. However, Ghorbani et al. (2020) studies the approximation power of neural networks rather than the learning. Compared to these studies, our work assumes the input is given by a dictionary multiplied with a hidden vector (not necessarily sparse) while the label is determined by a subset of the hidden vector, as motivated by pattern recognition applications in practice. Furthermore, we focus on the learning ability of networks instead of approximation.

## B    COMPLETE PROOFS FOR PROVABLE GUARANTEES OF NEURAL NETWORKS

We first make a few remarks about the proof.

*Remark.* The analysis can be carried out for more gradient steps following similar intuition, while we analyze two steps for simplicity.

*Remark.* Readers may notice that the network can be overparameterized. With sufficient overparameterization and proper initialization and step sizes, network learning becomes approximately NTK. However, here our learning scheme allows going beyond this kernel regime: we use aggressive gradient updates $\lambda_w^{(t)} = 1/(2\eta^{(t)})$ in the first two steps, completely forgetting the old weights to learn effective features. Using proper initialization and aggressive updates early to escape the kernel regime has been studied in existing work (e.g., Woodworth et al. (2020); Li et al. (2019)). Our result thus adds another concrete example.

**Notations.**    For a vector $v$ and an index set $I$, let $v_I$ denote the vector containing the entries of $v$ indexed by $I$, and $v_{-I}$ denote the vector containing the entries of $v$ with indices outside $I$.

By initialization, $w_i^{(0)}$ for $i \in [m]$ are i.i.d. copies of the same random variable $w^{(0)} \sim \mathcal{N}(0, \sigma_w^2 I_{d\times d})$; similar for $a^{(0)}$ and $b^{(0)}$. Let $q_\ell := \langle w^{(0)}, M_\ell \rangle$, then $\langle w^{(0)}, x \rangle = \langle \phi, q \rangle$. Similarly, define $q_{i,\ell}^{(t)} := \langle w_i^{(t)}, M_\ell \rangle$. Let $\sigma_\phi^2 := p_o(1 - p_o)/\tilde{\sigma}^2$ denote the variance of $\phi_\ell$ for $\ell \notin A$.

We also define the following sets to denote typical initialization. For a fixed $\delta \in (0, 1)$, define

$$\mathcal{G}_w(\delta) := \left\{ w \in \mathbb{R}^d : q_\ell = \langle w, M_\ell \rangle, \frac{\sigma_w^2(D - k)}{2} \leq \sum_{\ell \notin A} q_\ell^2 \leq \frac{3\sigma_w^2(D - k)}{2}, \right.$$

$$\left. \max_\ell |q_\ell| \leq \sigma_w \sqrt{2\log(Dm/\delta)} \right\}, \tag{5}$$

$$\mathcal{G}_a(\delta) := \{a \in \mathbb{R} : |a| \leq \sigma_a \sqrt{2\log(m/\delta)}\}. \tag{6}$$

$$\mathcal{G}_b(\delta) := \{b \in \mathbb{R} : |b| \leq \sigma_b \sqrt{2\log(m/\delta)}\}. \tag{7}$$

### B.1    EXISTENCE OF A GOOD NETWORK

we first show that there exists a network that can fit the data distribution.

**Lemma 7.** *For some $s, a, b \in \mathbb{R}$ with $a, b \geq 0$, define a function $\delta_{s,a,b} : \mathbb{R} \to \mathbb{R}$ as*

$$\delta_{s,a,b}(z) = a\sigma_r(z - s + b) - 2a\sigma_r(z - s) + a\sigma_r(z - s - b). \tag{8}$$

*where $\sigma_r(z) = \max\{z, 0\}$ is the ReLU activation function. Then*

$$\delta_{s,a,b}(z) = \begin{cases} 0 & \text{when } z \leq s - b, \\ a(z - s) + ab & \text{when } s - b \leq z \leq s, \\ -a(z - s) + ab & \text{when } s \leq z \leq s + b, \\ 0 & \text{when } s + b \leq z. \end{cases} \tag{9}$$

*That is, $\delta_{s,a,b}(z)$ linearly interpolates between $(s - b, 0), (s, ab), (s + b, 0)$ when $z \in [s - b, s + b]$, and is 0 elsewhere.*

*Proof of Lemma 7.* This can be simply verified for the four cases of the value of $z$. $\qquad\square$

**Lemma 8** (Restatement of Lemma 5). *For any $\mathcal{D} \in \mathcal{F}_\Xi$, there exists a network $g^*(x) = \sum_{i=1}^n a_i^* \sigma(\langle w_i^*, x \rangle + b_i^*)$ with $y = g^*(x)$ for any $(x, y) \sim \mathcal{D}$. Furthermore, the number of neurons $n = 3(k + 1)$, $|a_i^*| \leq 32k, 1/(32k) \leq |b_i^*| \leq 1/2$, $w_i^* = \tilde{\sigma} \sum_{j \in A} M_j/(4k)$, and $|\langle w_i^*, x \rangle + b_i^*| \leq 1$ for any $i \in [n]$ and $(x, y) \sim \mathcal{D}$.*

*Proof of Lemma 5.* Let $w = \tilde{\sigma} \sum_{j \in A} M_j$ and let $\mu = \sum_{j \in A} \mathbb{E}[\tilde{\phi}_j]$. We have

$$\langle w, x \rangle = \tilde{\sigma} \sum_{j \in A} \langle M_j, M\phi \rangle = \tilde{\sigma} \sum_{j \in A} \phi_j = \sum_{j \in A} \tilde{\phi}_j - \mu. \tag{10}$$

Then by Lemma 7,

$$g_1^*(x) := \sum_{p \in P} \delta_{p-\mu,2,1/2}(\langle w, x \rangle) - \sum_{p \notin P, 0 \le p \le k} \delta_{p-\mu,2,1/2}(\langle w, x \rangle) \tag{11}$$

$$= \sum_{p \in P} \delta_{p,2,1/2}(\langle w, x \rangle + \mu) - \sum_{p \notin P, 0 \le p \le k} \delta_{p,2,1/2}(\langle w, x \rangle + \mu) \tag{12}$$

$$= \sum_{p \in P} \delta_{p,2,1/2}\left(\sum_{j \in A} \tilde{\phi}_j\right) - \sum_{p \notin P, 0 \le p \le k} \delta_{p,2,1/2}\left(\sum_{j \in A} \tilde{\phi}_j\right) \tag{13}$$

$$= y \tag{14}$$

for any $(x, y) \sim \mathcal{D}$. Similarly,

$$g_2^*(x) := \sum_{p \in P} \delta_{p-\mu+1/4,4,1/2}(\langle w, x \rangle) - \sum_{p \notin P, 0 \le p \le k} \delta_{p-\mu+1/4,4,1/2}(\langle w, x \rangle) \tag{15}$$

$$= \sum_{p \in P} \delta_{p+1/4,4,1/2}(\langle w, x \rangle + \mu) - \sum_{p \notin P, 0 \le p \le k} \delta_{p+1/4,4,1/2}(\langle w, x \rangle + \mu) \tag{16}$$

$$= \sum_{p \in P} \delta_{p+1/4,4,1/2}\left(\sum_{j \in A} \tilde{\phi}_j\right) - \sum_{p \notin P, 0 \le p \le k} \delta_{p+1/4,4,1/2}\left(\sum_{j \in A} \tilde{\phi}_j\right) \tag{17}$$

$$= y \tag{18}$$

for any $(x, y) \sim \mathcal{D}$. Note that the bias terms in $g_1^*$ and $g_2^*$ have distance at least $1/4$, then at least one of them satisfies that all its bias terms have absolute value $\ge 1/8$. Pick that one and denote it as $g(x) = \sum_{i=1}^n a_i \sigma_r(\langle w_i, x \rangle + b_i)$. By the positive homogeneity of $\sigma_r$, we have

$$g(x) = \sum_{i=1}^n 4k a_i \sigma_r(\langle w_i, x \rangle/(4k) + b_i/(4k)). \tag{19}$$

Since for any $(x, y) \sim \mathcal{D}$, $|\langle w_i, x \rangle/(4k) + b_i/(4k)| \le 1$, then

$$g(x) = \sum_{i=1}^n 4k a_i \sigma(\langle w_i, x \rangle/(4k) + b_i/(4k)) \tag{20}$$

where $\sigma$ is the truncated ReLU. Now we can set $a_i^* = 4k a_i, w_i^* = w_i/(4k), b_i^* = b_i/(4k)$, to get our final $g^*$. $\qquad\square$

## B.2 INITIALIZATION

We first show that with high probability, the initial weights are in typical positions.

**Lemma 9.** *For any $\delta \in (0, 1)$, with probability at least $1 - \delta - 2 \exp(-\Theta(D - k))$ over $w^{(0)}$,*

$$\sigma_w^2(D - k)/2 \le \sum_{\ell \notin A} q_\ell^2 \le 3\sigma_w^2(D - k)/2,$$

$$\max_\ell |q_\ell| \le \sigma_w \sqrt{2 \log(D/\delta)}.$$

*With probability at least $1 - \delta$ over $b^{(0)}$,*

$$|b^{(0)}| \le \sigma_b \sqrt{2 \log(1/\delta)}.$$

*With probability at least $1 - \delta$ over $a^{(0)}$,*

$$|a^{(0)}| \le \sigma_a \sqrt{2 \log(1/\delta)}.$$

*Proof of Lemma 9.* From $q \sim \mathcal{N}(0, \sigma_w^2 I_{d \times d})$, we have:

- With probability $\geq 1 - \delta/2$, $\max_\ell |q_\ell| \leq \sqrt{2\sigma_w^2 \log \frac{D}{\delta}}$, and

- For any subset $S \subseteq [D]$, with probability $\geq 1 - 2\exp(-\Theta(|S|))$, $\|q_S\|_2^2 \in \left(\frac{|S|\sigma_w^2}{2}, \frac{3|S|\sigma_w^2}{2}\right)$.

Similar for $b^{(0)}$ and $a^{(0)}$. The lemma then follows. $\qquad\square$

**Lemma 10.** *We have:*

- *With probability $\geq 1 - \delta - 2m\exp(-\Theta(D-k))$ over $w_i^{(0)}$'s, for all $i \in [2m]$, $w_i^{(0)} \in \mathcal{G}_w(\delta)$.*

- *With probability $\geq 1 - \delta$ over $b_i^{(0)}$'s, for all $i \in [2m]$, $b_i^{(0)} \in \mathcal{G}_b(\delta)$.*

- *With probability $\geq 1 - \delta$ over $a_i^{(0)}$'s, for all $i \in [2m]$, $a_i^{(0)} \in \mathcal{G}_a(\delta)$.*

*Proof of Lemma 10.* This follows from Lemma 9 by union bound. $\qquad\square$

The following lemma about the typical $w_i^{(0)}$'s will be useful for later analysis.

**Lemma 11.** *Fix $\delta \in (0, 1)$. For any $w_i^{(0)} \in \mathcal{G}_w(\delta)$, we have*

$$\Pr_\phi \left[ \sum_{\ell \notin A} \phi_\ell q_{i,\ell}^{(0)} \geq \Theta\left(\sqrt{(D-k)\sigma_\phi^2 \sigma_w^2}\right) \right] = \Theta(1) - \frac{O(\log^{3/2}(Dm/\delta))}{\sqrt{(D-k)\sigma_\phi^2 \tilde{\sigma}^2}}. \tag{21}$$

*Consequently, when $p_o = \Omega(k^2/D)$ and $k = \Omega(\log^2(Dm/\delta))$,*

$$\Pr_\phi \left[ \sum_{\ell \notin A} \phi_\ell q_{i,\ell}^{(0)} \geq \Theta(\sigma_w) \right] = \Theta(1) - \frac{O(1)}{k^{1/4}}. \tag{22}$$

*Proof of Lemma 11.* Note that for $\ell \notin A$, $\mathbb{E}[\phi_\ell] = 0$, $\mathbb{E}[\phi_\ell^2] = \sigma_\phi^2$, and $\mathbb{E}[|\phi_\ell|^3] = \Theta(\sigma_\phi^2/\tilde{\sigma})$. Then the statement follows from Berry-Esseen Theorem. $\qquad\square$

## B.3 Some Auxiliary Lemmas

The expression of the gradients will be used frequently.

**Lemma 12.**

$$\frac{\partial}{\partial w_i} L_\mathcal{D}(g; \sigma_\xi) = -a_i \mathbb{E}_{(x,y) \sim \mathcal{D}} \left\{ y\mathbb{I}[yg(x;\xi) \leq 1] \mathbb{E}_{\xi_i} \mathbb{I}[\langle w_i, x\rangle + b_i + \xi_i \in (0,1)]x \right\}, \tag{23}$$

$$\frac{\partial}{\partial b_i} L_\mathcal{D}(g; \sigma_\xi) = -a_i \mathbb{E}_{(x,y) \sim \mathcal{D}} \left\{ y\mathbb{I}[yg(x;\xi) \leq 1] \mathbb{E}_{\xi_i} \mathbb{I}[\langle w_i, x\rangle + b_i \in (0,1)] \right\}, \tag{24}$$

$$\frac{\partial}{\partial a_i} L_\mathcal{D}(g; \sigma_\xi) = -\mathbb{E}_{(x,y) \sim \mathcal{D}} \left\{ y\mathbb{I}[yg(x;\xi) \leq 1] \mathbb{E}_{\xi_i} \sigma(\langle w_i, x\rangle + b_i + \xi_i) \right\}. \tag{25}$$

*Proof of Lemma 12.* It follows from straightforward calculation. $\qquad\square$

We now show that a small subset of the entries in $\phi, q$ does not affect the probability distribution of $\langle \phi, q\rangle$ much.

**Lemma 13.** *Suppose $\nu \sim \mathcal{N}(0, \sigma^2)$. For any $B \supseteq A$ and any $b$:*

$$\left| \Pr_{\phi_{-B}, \nu} \{\langle \phi, q\rangle + \nu \geq b\} - \Pr_{\phi_{-B}, \nu} \{\langle \phi_{-B}, q_{-B}\rangle + \nu \geq b\} \right| \tag{26}$$

$$\leq O\left( \frac{|\langle \phi_B, q_B\rangle|}{(\sigma_\phi^2\|q_{-B}\|_2^2 + \sigma^2)^{1/2}} + \frac{\sigma^3 + \sigma_\phi^2\|q_{-B}\|_3^3/\tilde{\sigma}}{(\sigma^2 + \sigma_\phi^2\|q_{-B}\|_2^2)^{3/2}} \right). \tag{27}$$

*Similarly,*

$$\left| \Pr_{\phi_{-B}} \left\{ \langle \phi, q \rangle \geq b \right\} - \Pr_{\phi_{-B}} \left\{ \langle \phi_{-B}, q_{-B} \rangle \geq b \right\} \right| \tag{28}$$

$$\leq O\left( \frac{|\langle \phi_B, q_B \rangle|}{\sigma_\phi \|q_{-B}\|_2} + \frac{\|q_{-B}\|_3^3}{\tilde{\sigma} \sigma_\phi \|q_{-B}\|_2^3} \right). \tag{29}$$

*Proof of Lemma 13.* Note that for $\ell \notin A$, $\mathbb{E}[\phi_\ell] = 0$, $\mathbb{E}[\phi_\ell^2] = \sigma_\phi^2$, and $\mathbb{E}[|\phi_\ell|^3] = \Theta(\sigma_\phi^2/\tilde{\sigma})$. Let $t = |\langle \phi_B, q_B \rangle|$. Then by the Berry-Esseen Theorem,

$$\left| \Pr_{\phi_{-B}} \left\{ \langle \phi, q \rangle + \nu \geq b \right\} - \Pr_{\phi_{-B}} \left\{ \langle \phi_{-B}, q_{-B} \rangle + \nu \geq b \right\} \right| \tag{30}$$

$$\leq \Pr_{\phi_{-B}} \left\{ \langle \phi_{-B}, q_{-B} \rangle + \nu \in [-t + b, t + b] \right\} \tag{31}$$

$$\leq \frac{2t}{(\sigma_\phi^2 \|q_{-B}\|_2^2 + \sigma^2)^{1/2}} + \frac{O(\sigma^3 + \sigma_\phi^2 \|q_{-B}\|_3^3/\tilde{\sigma})}{(\sigma^2 + \sigma_\phi^2 \|q_{-B}\|_2^2)^{3/2}}. \tag{32}$$

The second statement follows from a similar argument. $\qquad\square$

We also have the following auxiliary lemma for later calculations.

**Lemma 14.**

$$\mathbb{E}_{\phi_A} \{y\} = 0, \tag{33}$$

$$\mathbb{E}_{\phi_A} \{|y|\} = 1, \tag{34}$$

$$\mathbb{E}_{\phi_j} \{|\phi_j|\} = 2\sigma_\phi^2 \tilde{\sigma}, \text{ for } j \notin A, \tag{35}$$

$$\mathbb{E}_{\phi_A} \{y\phi_j\} = \frac{\gamma}{\tilde{\sigma}}, \tag{36}$$

$$\mathbb{E}_{\phi_A} \{|y\phi_j|\} \leq \frac{1}{\tilde{\sigma}}, \text{ for all } j \in [D]. \tag{37}$$

*Proof of Lemma 14.*

$$\mathbb{E}_{\phi_A} \{y\} = \sum_{v \in \{\pm 1\}} \mathbb{E}_{\phi_A} \{y | y = v\} \Pr[y = v] \tag{38}$$

$$= \frac{1}{2} \sum_{v \in \{\pm 1\}} \mathbb{E}_{\phi_A} \{y | y = v\} \tag{39}$$

$$= 0. \tag{40}$$

$$\mathbb{E}_{\phi_A} \{|y|\} = \sum_{v \in \{\pm 1\}} \mathbb{E}_{\phi_A} \{|y| \, | y = v\} \Pr[y = v] \tag{41}$$

$$= \frac{1}{2} \sum_{v \in \{\pm 1\}} \mathbb{E}_{\phi_A} \{|y| \, | y = v\} \tag{42}$$

$$= 1. \tag{43}$$

$$\mathbb{E}_{\phi_j} \{|\phi_j|\} = \frac{|-p_o|(1 - p_o) + |1 - p_o|p_o}{\tilde{\sigma}} = 2\sigma_\phi^2 \tilde{\sigma}. \tag{44}$$

$$\mathbb{E}_{\phi_A} \{y\phi_j\} = \mathbb{E}_{\phi_A} \left\{ y \frac{\tilde{\phi}_j - \mathbb{E}[\tilde{\phi}_j]}{\tilde{\sigma}} \right\} \tag{45}$$

$$= \frac{1}{\tilde{\sigma}} \mathbb{E}_{\phi_A} \left\{ y\tilde{\phi}_j - y\mathbb{E}[\tilde{\phi}_j] \right\} \tag{46}$$

$$= \frac{\gamma}{\tilde{\sigma}}. \tag{47}$$

$$\mathbb{E}_{\phi_A} \{|y\phi_j|\} = \mathbb{E}_{\phi_A} \{|\phi_j|\} \tag{48}$$

$$\leq \frac{1}{\tilde{\sigma}}. \tag{49}$$

$\square$

## B.4 FEATURE EMERGENCE: FIRST GRADIENT STEP

We will show that w.h.p. over the initialization, after the first gradient step, there are neurons that represent good features.

We begin with analyzing the gradients.

**Lemma 15** (Full version of Lemma 6). *Fix $\delta \in (0,1)$ and suppose $w_i^{(0)} \in \mathcal{G}_w(\delta), b_i^{(0)} \in \mathcal{G}_b(\delta)$ for all $i \in [2m]$. Let*

$$\epsilon_e := \frac{k \log^{1/2}(Dm/\delta) + \log^{3/2}(Dm/\delta)}{\sqrt{\sigma_\phi^2 \tilde{\sigma}^2 (D-k)}}.$$

*If $p_o = \Omega(k^2/D)$, $k = \Omega(\log^2(Dm/\delta))$, and $\sigma_\xi^{(1)} < 1/k$, then*

$$\frac{\partial}{\partial w_i} L_\mathcal{D}(g^{(0)}; \sigma_\xi^{(1)}) = -a_i^{(0)} \sum_{j=1}^{D} M_j T_j \tag{50}$$

*where $T_j$ satisfies:*

- *if $j \in A$, then $|T_j - \beta\gamma/\tilde{\sigma}| \leq O(\epsilon_e/\tilde{\sigma})$, where $\beta \in [\Omega(1), 1]$ and depends only on $w_i^{(0)}, b_i^{(0)}$;*

- *if $j \notin A$, then $|T_j| \leq O(\sigma_\phi^2 \epsilon_e \tilde{\sigma})$.*

*Proof of Lemma 15.* Consider one neuron index $i$ and omit the subscript $i$ in the parameters. Since the unbiased initialization leads to $g^{(0)}(x; \xi^{(1)}) = 0$, we have

$$\frac{\partial}{\partial w} L_\mathcal{D}(g^{(0)}; \sigma_\xi^{(1)}) \tag{51}$$

$$= -a^{(0)} \mathbb{E}_{(x,y)\sim\mathcal{D}} \left\{ y \mathbb{I}[yg^{(0)}(x; \xi^{(1)}) \leq 1] \mathbb{E}_{\xi^{(1)}} \mathbb{I}[\langle w^{(0)}, x \rangle + b^{(0)} + \xi^{(1)} \in (0,1)]x \right\} \tag{52}$$

$$= -a^{(0)} \mathbb{E}_{(x,y)\sim\mathcal{D},\xi^{(1)}} \left\{ y \mathbb{I}[\langle w^{(0)}, x \rangle + b^{(0)} + \xi^{(1)} \in (0,1)]x \right\} \tag{53}$$

$$= -a^{(0)} \sum_{j=1}^{D} M_j \underbrace{\mathbb{E}_{(x,y)\sim\mathcal{D},\xi^{(1)}} \left\{ y \mathbb{I}[\langle w^{(0)}, x \rangle + b^{(0)} + \xi^{(1)} \in (0,1)]\phi_j \right\}}_{:=T_j}. \tag{54}$$

First, consider $j \in A$.

$$T_j = \mathbb{E}_{(x,y)\sim\mathcal{D},\xi^{(1)}} \left\{ y \mathbb{I}[\langle w^{(0)}, x \rangle + b^{(0)} + \xi^{(1)} \in (0,1)]\phi_j \right\} \tag{55}$$

$$= \mathbb{E}_{\phi_A,\xi^{(1)}} \left\{ y\phi_j \Pr_{\phi_{-A}} \left[ \langle \phi, q \rangle + b^{(0)} + \xi^{(1)} \in (0,1) \right] \right\}. \tag{56}$$

Let

$$I_a := \Pr_{\phi_{-A}} \left[ \langle \phi, q \rangle + b^{(0)} + \xi^{(1)} \in (0,1) \right], \tag{57}$$

$$I_a' := \Pr_{\phi_{-A}} \left[ \langle \phi_{-A}, q_{-A} \rangle + b^{(0)} + \xi^{(1)} \in (0,1) \right]. \tag{58}$$

We have

$$|\mathbb{E}_{\xi^{(1)}}(I_a - I_a')| \tag{59}$$

$$\leq \mathbb{E}_{\xi^{(1)}} \left| \Pr_{\phi_{-A}} \left[ \langle \phi, q \rangle + b^{(0)} + \xi^{(1)} \geq 0 \right] - \Pr_{\phi_{-A}} \left[ \langle \phi_{-A}, q_{-A} \rangle + b^{(0)} + \xi^{(1)} \geq 0 \right] \right| \tag{60}$$

$$+ \Pr_{\phi_{-A},\xi^{(1)}} \left[ \langle \phi, q \rangle + b^{(0)} + \xi^{(1)} \geq 1 \right] + \Pr_{\phi_{-A},\xi^{(1)}} \left[ \langle \phi_{-A}, q_{-A} \rangle + b^{(0)} + \xi^{(1)} \geq 1 \right]. \tag{61}$$

Then by Lemma 13,

$$\left| \Pr_{\phi_{-A}}\left[ \langle \phi, q \rangle + b^{(0)} + \xi^{(1)} \geq 0 \right] - \Pr_{\phi_{-A}}\left[ \langle \phi_{-A}, q_{-A} \rangle + b^{(0)} + \xi^{(1)} \geq 0 \right] \right| = O(\epsilon_{\mathrm{e}}). \quad (62)$$

Note that $\sum_{\ell \notin A} \mathrm{Var}(\phi_\ell q_\ell) = \Theta(\sigma_\phi^2 \sigma_w^2 (D - k)) = \Theta(\sigma_w^2)$, and $|\phi_\ell| \leq \frac{1}{\tilde{\sigma}}$, $\max_\ell |q_\ell| \leq \sigma_w \sqrt{2 \log(Dm/\delta)}$. Applying Bernstein's inequality for bounded distributions, we have:

$$\Pr_{\phi_{-A}}\left[ \langle \phi_{-A}, q_{-A} \rangle \geq 1/4 \right] = \exp(-\Omega(k)) = O(\epsilon_{\mathrm{e}}). \quad (63)$$

We also have:

$$\Pr_{\xi^{(1)}}\left[ b^{(0)} + \xi^{(1)} \geq 1/4 \right] = \exp(-\Omega(k)) = O(\epsilon_{\mathrm{e}}). \quad (64)$$

Therefore,

$$\Pr_{\phi_{-A}, \xi^{(1)}}\left[ \langle \phi, q \rangle + b^{(0)} + \xi^{(1)} \geq 1 \right] = \exp(-\Omega(k)) = O(\epsilon_{\mathrm{e}}) \quad (65)$$

where the last step follows from the assumption on $\sigma_w$ and $k$. A similar argument gives:

$$\Pr_{\phi_{-A}, \xi^{(1)}}\left[ \langle \phi_{-A}, q_{-A} \rangle + b^{(0)} + \xi^{(1)} \geq 1 \right] = \exp(-\Omega(k)) = O(\epsilon_{\mathrm{e}}). \quad (66)$$

Then we have

$$\left| T_j - \mathbb{E}_{\phi_A, \xi^{(1)}} \left\{ y\phi_j I'_a \right\} \right| \quad (67)$$

$$\leq \mathbb{E}_{\phi_A} \left\{ |y\phi_j| \left| \mathbb{E}_{\xi^{(1)}}(I_a - I'_a) \right| \right\} \quad (68)$$

$$\leq O(\epsilon_{\mathrm{e}}) \mathbb{E}_{\phi_A} \left\{ |y\phi_j| \right\} \quad (69)$$

$$\leq O(\epsilon_{\mathrm{e}}/\tilde{\sigma}) \quad (70)$$

where the last step is from Lemma 14. Furthermore,

$$\mathbb{E}_{\phi_A, \xi^{(1)}} \left\{ y\phi_j I'_a \right\} \quad (71)$$

$$= \mathbb{E}_{\phi_A} \left\{ y\phi_j \right\} \mathbb{E}_{\xi^{(1)}}[I'_a] \quad (72)$$

$$= \mathbb{E}_{\phi_A} \left\{ y\phi_j \right\} \Pr_{\phi_{-A}, \xi^{(1)}} \left[ \langle \phi_{-A}, q_{-A} \rangle + b^{(0)} + \xi^{(1)} \in (0, 1) \right] \quad (73)$$

By Lemma 11, the assumption on $p_{\mathrm{o}}$, and (63), we have

$$\Pr_{\phi_{-A}}\left[ \langle \phi_{-A}, q_{-A} \rangle + b^{(0)} \in (0, 1/2) \right] \geq \Omega(1) - O(1/k^{1/4}), \quad (74)$$

$$\Pr_{\xi^{(1)}}\left[ \xi^{(1)} \in (0, 1/2) \right] = 1/2 - \exp(-\Omega(k)), \quad (75)$$

This leads to

$$\beta := \mathbb{E}_{\xi^{(1)}}[I'_a] = \Pr_{\phi_{-A}, \xi^{(1)}} \left[ \langle \phi_{-A}, q_{-A} \rangle + b^{(0)} + \xi^{(1)} \in (0, 1) \right] \geq \Omega(1). \quad (76)$$

By Lemma 14, $\mathbb{E}_{\phi_A} \left\{ y\phi_j \right\} = \gamma/\tilde{\sigma}$. Therefore,

$$\left| T_j - \beta\gamma/\tilde{\sigma} \right| \leq O(\epsilon_{\mathrm{e}}/\tilde{\sigma}). \quad (77)$$

Now, consider $j \notin A$. Let $B$ denote $A \cup \{j\}$.

$$T_j = \mathbb{E}_{(x,y) \sim \mathcal{D}, \xi^{(1)}} \left\{ y\phi_j \mathbb{I}\left[ \langle \phi, q \rangle + b^{(0)} + \xi^{(1)} \in (0, 1) \right] \right\} \quad (78)$$

$$= \mathbb{E}_{\phi_B} \mathbb{E}_{\phi_{-B}, \xi^{(1)}} \left\{ y\phi_j \mathbb{I}\left[ \langle \phi, q \rangle + b^{(0)} + \xi^{(1)} \in (0, 1) \right] \right\} \quad (79)$$

$$= \mathbb{E}_{\phi_B, \xi^{(1)}} \left\{ y\phi_j \Pr_{\phi_{-B}} \left[ \langle \phi, q \rangle + b^{(0)} + \xi^{(1)} \in (0, 1) \right] \right\}. \quad (80)$$

Let

$$I_b := \Pr_{\phi_{-B}} \left[ \langle \phi, q \rangle + b^{(0)} + \xi^{(1)} \in (0, 1) \right], \tag{81}$$

$$I'_b := \Pr_{\phi_{-B}} \left[ \langle \phi_{-B}, q_{-B} \rangle + b^{(0)} + \xi^{(1)} \in (0, 1) \right]. \tag{82}$$

Similar as above, we have $|\mathbb{E}_{\xi^{(1)}}(I_b - I'_b)| \le O(\epsilon_e)$ by Lemma 13. Then by Lemma 14,

$$\left| T_j - \mathbb{E}_{\phi_B, \xi^{(1)}} \left\{ y \phi_j I'_b \right\} \right| \tag{83}$$

$$\le \mathbb{E}_{\phi_B} \left\{ |y \phi_j| |\mathbb{E}_{\xi^{(1)}}(I_b - I'_b)| \right\} \tag{84}$$

$$\le O(\epsilon_e) \mathbb{E}_{\phi_A} \{|y|\} \mathbb{E}_{\phi_j} \{|\phi_j|\} \tag{85}$$

$$\le O(\epsilon_e) \times O(\sigma_\phi^2 \tilde{\sigma}) \tag{86}$$

$$= O(\sigma_\phi^2 \epsilon_e \tilde{\sigma}). \tag{87}$$

Furthermore,

$$\mathbb{E}_{\phi_B, \xi^{(1)}} \left\{ y \phi_j I'_b \right\} = \mathbb{E}_{\phi_A} \{y\} \mathbb{E}_{\phi_j} \{\phi_j\} \mathbb{E}_{\xi^{(1)}} [I'_b] = 0. \tag{88}$$

Therefore,

$$|T_j| \le O(\sigma_\phi^2 \epsilon_e \tilde{\sigma}). \tag{89}$$

$\square$

**Lemma 16.** *Under the same assumptions as in Lemma 15,*

$$\frac{\partial}{\partial b_i} L_{\mathcal{D}}(g^{(0)}; \sigma_\xi^{(1)}) = -a_i^{(0)} T_b \tag{90}$$

*where* $|T_b| \le O(\epsilon_e)$.

*Proof of Lemma 16.* Consider one neuron index $i$ and omit the subscript $i$ in the parameters. Since the unbiased initialization leads to $g^{(0)}(x; \xi^{(1)}) = 0$, we have

$$\frac{\partial}{\partial b} L_{\mathcal{D}}(g^{(0)}; \sigma_\xi^{(1)}) \tag{91}$$

$$= -a^{(0)} \mathbb{E}_{(x,y) \sim \mathcal{D}} \left\{ y \mathbb{I}[yg^{(0)}(x; \xi) \le 1] \mathbb{E}_{\xi^{(1)}} \mathbb{I}[\langle w^{(0)}, x \rangle + b^{(0)} + \xi^{(1)} \in (0, 1)] \right\} \tag{92}$$

$$= -a^{(0)} \underbrace{\mathbb{E}_{(x,y) \sim \mathcal{D}, \xi^{(1)}} \left\{ y \mathbb{I}[\langle w^{(0)}, x \rangle + b^{(0)} + \xi^{(1)} \in (0, 1)] \right\}}_{:=T_b}. \tag{93}$$

Similar to the proof in Lemma 6,

$$\left| \Pr_{\phi_{-A}} [\langle \phi, q \rangle + b^{(0)} + \xi^{(1)} \in (0, 1)] - \Pr_{\phi_{-A}} [\langle \phi_{-A}, q_{-A} \rangle + b^{(0)} + \xi^{(1)} \in (0, 1)] \right| = O(\epsilon_e). \tag{94}$$

Then

$$\left| T_b - \mathbb{E}_{\phi_A, \xi^{(1)}} \left\{ y \Pr_{\phi_{-A}} [\langle \phi_{-A}, q_{-A} \rangle + b^{(0)} + \xi^{(1)} \in (0, 1)] \right\} \right| \tag{95}$$

$$= \mathbb{E}_{\phi_A, \xi^{(1)}} \left\{ |y| \left| \Pr_{\phi_{-A}} [\langle \phi, q \rangle + b^{(0)} + \xi^{(1)} \in (0, 1)] - \Pr_{\phi_{-A}} [\langle \phi_{-A}, q_{-A} \rangle + b^{(0)} + \xi^{(1)} \in (0, 1)] \right| \right\} \tag{96}$$

$$\le O(\epsilon_e) \mathbb{E}_{\phi_A} \{|y|\} \tag{97}$$

$$\le O(\epsilon_e). \tag{98}$$

Also,

$$\mathbb{E}_{\phi_A, \xi^{(1)}} \left\{ y \Pr_{\phi_{-A}} [\langle \phi_{-A}, q_{-A} \rangle + b^{(0)} + \xi^{(1)} \in (0, 1)] \right\} \tag{99}$$

$$= \mathbb{E}_{\phi_A} \{y\} \Pr_{\phi_{-A}, \xi^{(1)}} [\langle \phi_{-A}, q_{-A} \rangle + b^{(0)} + \xi^{(1)} \in (0, 1)] \tag{100}$$

$$= 0. \tag{101}$$

Therefore, $|T_b| \le O(\epsilon_e)$. $\square$

**Lemma 17.** *We have*

$$\frac{\partial}{\partial a_i} L_{\mathcal{D}}(g^{(0)}; \sigma_\xi^{(1)}) = -T_a \tag{102}$$

*where* $|T_a| \leq O(\max_\ell q_{i,\ell}^{(0)})$. *So if* $w_i^{(0)} \in \mathcal{G}(\delta)$, $|T_a| \leq O(\sigma_w \sqrt{\log(Dm/\delta)})$.

*Proof of Lemma 17.* Consider one neuron index $i$ and omit the subscript $i$ in the parameters. Since the unbiased initialization leads to $g^{(0)}(x; \xi^{(1)}) = 0$, we have

$$\frac{\partial}{\partial a} L_{\mathcal{D}}(g^{(0)}; \sigma_\xi^{(1)}) \tag{103}$$

$$= -\mathbb{E}_{(x,y)\sim\mathcal{D}} \left\{ y\mathbb{I}[yg^{(0)}(x; \xi^{(1)}) \leq 1]\mathbb{E}_{\xi^{(1)}}\sigma(\langle w^{(0)}, x\rangle + b^{(0)} + \xi^{(1)}) \right\} \tag{104}$$

$$= -\underbrace{\mathbb{E}_{(x,y)\sim\mathcal{D},\xi^{(1)}} \left\{ y\sigma(\langle w^{(0)}, x\rangle + b^{(0)} + \xi^{(1)}) \right\}}_{:=T_a}. \tag{105}$$

Let $\phi'_A$ be an independent copy of $\phi_A$, $\phi'$ be the vector obtained by replacing in $\phi$ the entries $\phi_A$ with $\phi'_A$, and let $x' = M\phi'$ and its label is $y'$. Then

$$|T_a| = \left| \mathbb{E}_{\phi_A} \left\{ y\mathbb{E}_{\phi_{-A},\xi^{(1)}}\sigma(\langle w^{(0)}, x\rangle + b^{(0)} + \xi^{(1)}) \right\} \right| \tag{106}$$

$$\leq \frac{1}{2} \left| \mathbb{E}_{\phi_A} \left\{ \mathbb{E}_{\phi_{-A},\xi^{(1)}}\sigma(\langle w^{(0)}, x\rangle + b^{(0)} + \xi^{(1)})|y = 1 \right\} \right. \tag{107}$$

$$\left. - \mathbb{E}_{\phi_A} \left\{ \mathbb{E}_{\phi_{-A},\xi^{(1)}}\sigma(\langle w^{(0)}, x\rangle + b^{(0)} + \xi^{(1)})|y = -1 \right\} \right| \tag{108}$$

$$\leq \frac{1}{2} \left| \mathbb{E}_{\phi_A} \left\{ \mathbb{E}_{\phi_{-A},\xi^{(1)}}\sigma(\langle w^{(0)}, x\rangle + b^{(0)} + \xi^{(1)})|y = 1 \right\} \right. \tag{109}$$

$$\left. - \mathbb{E}_{\phi'_A} \left\{ \mathbb{E}_{\phi_{-A},\xi^{(1)}}\sigma(\langle w^{(0)}, x'\rangle + b^{(0)} + \xi^{(1)})|y' = -1 \right\} \right|. \tag{110}$$

Since $\sigma$ is 1-Lipschitz,

$$|T_a| \leq \frac{1}{2}\mathbb{E}_{\phi_A,\phi'_A} \left\{ \mathbb{E}_{\phi_{-A}} \left| \langle w^{(0)}, x\rangle - \langle w^{(0)}, x'\rangle \right| |y = 1, y' = -1 \right\} \tag{111}$$

$$\leq \frac{1}{2}\mathbb{E}_{\phi_{-A}} \left( \mathbb{E}_{\phi_A} \left\{ \left| \langle w^{(0)}, x\rangle \right| |y = 1 \right\} + \mathbb{E}_{\phi'_A} \left\{ \left| \langle w^{(0)}, x'\rangle \right| |y' = -1 \right\} \right) \tag{112}$$

$$= \mathbb{E}_{\phi_{-A},\phi_A} \left| \langle w^{(0)}, x\rangle \right| \tag{113}$$

$$= \mathbb{E}_x \left| \langle w^{(0)}, x\rangle \right| \tag{114}$$

$$\leq \sqrt{\mathbb{E}_x \langle w^{(0)}, x\rangle^2} \tag{115}$$

$$\leq \max_\ell q_{i,\ell}^{(0)} \sqrt{\mathbb{E}_x \left( \sum_{\ell \in [D]} \phi_\ell^2 + \sum_{j \neq \ell: j, \ell \in A} |\phi_j \phi_\ell| \right)} \tag{116}$$

$$\leq \max_\ell q_{i,\ell}^{(0)} \sqrt{\mathbb{E}_x (1 + O(1))} \tag{117}$$

$$= \Theta(\max_\ell q_{i,\ell}^{(0)}). \tag{118}$$

$\square$

With the bounds on the gradient, we now summarize the results for the weights after the first gradient step.

**Lemma 18.** *Set*

$$\lambda_w^{(1)} = 1/(2\eta^{(1)}), \lambda_a^{(1)} = \lambda_b^{(1)} = 0, \sigma_\xi^{(1)} = 1/k^{3/2}.$$

*Fix* $\delta \in (0,1)$ *and suppose* $w_i^{(0)} \in \mathcal{G}_w(\delta), b_i^{(0)} \in \mathcal{G}_b(\delta)$ *for all* $i \in [2m]$. *If* $p_o = \Omega(k^2/D)$, $k = \Omega(\log^2(Dm/\delta))$, *then for all* $i \in [m]$, $w_i^{(1)} = \sum_{\ell=1}^{D} q_{i,\ell}^{(1)} M_\ell$ *satisfying*

- *if* $\ell \in A$, *then* $|q_{i,\ell}^{(1)} - \eta^{(1)} a_i^{(0)} \beta \gamma / \tilde{\sigma}| \leq O\left(\frac{|\eta^{(1)} a_i^{(0)}| \epsilon_e}{\tilde{\sigma}}\right)$, *where* $\beta \in [\Omega(1), 1]$ *and depends only on* $w_i^{(0)}, b_i^{(0)}$;

- *if* $\ell \notin A$, *then* $|q_{i,\ell}^{(1)}| \leq O\left(\sigma_\phi^2 |\eta^{(1)} a_i^{(0)}| \epsilon_e \tilde{\sigma}\right)$;

*and*

- $b_i^{(1)} = b_i^{(0)} + \eta^{(1)} a_i^{(0)} T_b$ *where* $|T_b| = O(\epsilon_e)$;

- $a_i^{(1)} = a_i^{(0)} + \eta^{(1)} T_a$ *where* $|T_a| = O(\sigma_w \sqrt{\log(Dm/\delta)})$.

*Proof of Lemma 18.* This follows from Lemma 10 and Lemma 15-17. $\square$

### B.5 FEATURE IMPROVEMENT: SECOND GRADIENT STEP

We first show that with properly set $\eta^{(1)}$, for most $x$, $|g^{(1)}(x; \sigma_\xi^{(2)})| < 1$ and thus $yg^{(1)}(x; \sigma_\xi^{(2)}) < 1$.

**Lemma 19.** *Fix* $\delta \in (0,1)$ *and suppose* $w_i^{(0)} \in \mathcal{G}_w(\delta), b_i^{(0)} \in \mathcal{G}_b(\delta), a_i^{(0)} \in \mathcal{G}_a(\delta)$ *for all* $i \in [2m]$. *If* $p_o = \Omega(k^2/D)$, $k = \Omega(\log^2(Dm/\delta))$, $\sigma_a \leq \tilde{\sigma}^2/(\gamma k^2)$, $\eta^{(1)} = O\left(\frac{\gamma}{km\sigma_a}\right)$, *and* $\sigma_\xi^{(2)} \leq 1/k$, *then with probability* $\geq 1 - \exp(-\Theta(k))$ *over* $(x,y)$, *we have* $yg^{(1)}(x; \sigma_\xi^{(2)}) < 1$. *Furthermore, for any* $i \in [2m]$, $\left|\langle w_i^{(1)}, x \rangle\right| = \left|\langle q_i^{(1)}, \phi \rangle\right| = O(\eta^{(1)} \tilde{\sigma}/\gamma)$, $\left|\langle (q_i^{(1)})_{-A}, \phi_{-A} \rangle\right| = O(\eta^{(1)} \tilde{\sigma}/\gamma)$, *and* $\left|b_i^{(1)} - b_{m+i}^{(1)}\right| = O(|\eta^{(1)} a_i^{(0)}| \epsilon_e)$.

*Proof of Lemma 19.* Note that $w_i^{(0)} = w_{m+i}^{(0)}$, $b_i^{(0)} = b_{m+i}^{(0)}$, and $a_i^{(0)} = -a_{m+i}^{(0)}$. Then the gradient for $w_i$ is the negation of that for $w_{m+i}$, the gradient for $b_i$ is the negation of that for $b_{m+i}$, and the gradient for $a_i$ is the same as that for $a_{m+i}$. With probability $\geq 1 - \exp(-\Theta(\max\{2p_o(D-k), k\}))$, among all $j \notin A$, we have that at most $2p_o(D-k) + k$ of $\phi_j$ are $(1 - p_o)/\tilde{\sigma}$, while the others are $-p_o/\tilde{\sigma}$. For data points with $\phi$ satisfying this, we have:

$$\left|g^{(1)}(x; \sigma_\xi^{(2)})\right| \tag{119}$$

$$= \left|\sum_{i=1}^{2m} a_i^{(1)} \mathbb{E}_{\xi^{(2)}} \sigma(\langle w_i^{(1)}, x \rangle + b_i^{(1)} + \xi_i^{(2)})\right| \tag{120}$$

$$= \left|\sum_{i=1}^{m} \left(a_i^{(1)} \mathbb{E}_{\xi^{(2)}} \sigma(\langle w_i^{(1)}, x \rangle + b_i^{(1)} + \xi_i^{(2)}) + a_{m+i}^{(1)} \mathbb{E}_{\xi^{(2)}} \sigma(\langle w_{m+i}^{(1)}, x \rangle + b_{m+i}^{(1)} + \xi_{m+i}^{(2)})\right)\right| \tag{121}$$

$$\leq \left|\sum_{i=1}^{m} \left(a_i^{(1)} \mathbb{E}_{\xi^{(2)}} \sigma(\langle w_i^{(1)}, x \rangle + b_i^{(1)} + \xi_i^{(2)}) + a_{m+i}^{(1)} \mathbb{E}_{\xi^{(2)}} \sigma(\langle w_i^{(1)}, x \rangle + b_i^{(1)} + \xi_i^{(2)})\right)\right| \tag{122}$$

$$+ \left|\sum_{i=1}^{m} \left(-a_{m+i}^{(1)} \mathbb{E}_{\xi^{(2)}} \sigma(\langle w_i^{(1)}, x \rangle + b_i^{(1)} + \xi_i^{(2)}) + a_{m+i}^{(1)} \mathbb{E}_{\xi^{(2)}} \sigma(\langle w_{m+i}^{(1)}, x \rangle + b_{m+i}^{(1)} + \xi_i^{(2)})\right)\right|. \tag{123}$$

Then we have

$$\left| g^{(1)}(x; \sigma_\xi^{(2)}) \right| \le \sum_{i=1}^{m} \left| 2\eta^{(1)} T_a \mathbb{E}_{\xi^{(2)}} \sigma(\langle w_i^{(1)}, x \rangle + b_i^{(1)} + \xi_i^{(2)}) \right| \tag{124}$$

$$+ \sum_{i=1}^{m} \left| a_{m+i}^{(1)} \right| \left( \left| \langle w_i^{(1)} - w_{m+i}^{(1)}, x \rangle \right| + \left| b_i^{(1)} - b_{m+i}^{(1)} \right| \right) \tag{125}$$

$$\le \sum_{i=1}^{m} \left| 2\eta^{(1)} T_a \right| \left( \left| \langle w_i^{(1)}, x \rangle + b_i^{(1)} \right| + \mathbb{E}_{\xi^{(2)}} \left| \xi_i^{(2)} \right| \right) \tag{126}$$

$$+ \sum_{i=1}^{m} \left| a_{m+i}^{(1)} \right| \left( \left| \langle w_i^{(1)} - w_{m+i}^{(1)}, x \rangle \right| + \left| b_i^{(1)} - b_{m+i}^{(1)} \right| \right). \tag{127}$$

We have $|T_a| = O(\sigma_w \sqrt{\log(Dm/\delta)})$, and

$$\left| \langle w_i^{(1)}, x \rangle \right| \le O(|\eta^{(1)} a_i^{(0)}|) (\beta\gamma/\tilde{\sigma} + \epsilon_e/\tilde{\sigma}) \frac{k}{\tilde{\sigma}} \tag{128}$$

$$+ O(|\eta^{(1)} a_i^{(0)}| \sigma_\phi^2 \epsilon_e \tilde{\sigma}) ((2p_o(D-k) + k)(1 - p_o)/\tilde{\sigma} + p_o D/\tilde{\sigma}) \tag{129}$$

$$\le O(|\eta^{(1)} a_i^{(0)}|) \left( k\gamma/\tilde{\sigma}^2 + \epsilon_e k/\tilde{\sigma}^2 + (k + p_o D)\sigma_\phi^2 \epsilon_e \right) \tag{130}$$

$$\le O(\eta^{(1)}(1 + p_o\tilde{\sigma})/\gamma). \tag{131}$$

$$\left| b_i^{(1)} \right| \le \left| b_i^{(0)} \right| + \left| \eta^{(1)} a_i^{(0)} T_b \right| \tag{132}$$

$$\le \frac{\sqrt{\log(m/\delta)}}{k^2} + \left| \eta^{(1)} a_i^{(0)} \frac{\epsilon_e}{\tilde{\sigma}} \right|. \tag{133}$$

$$\mathbb{E}_{\xi^{(2)}} \left| \xi_i^{(2)} \right| \le O(\sigma_\xi^{(2)}). \tag{134}$$

$$|a_{m+i}^{(1)}| \le |a_i^{(0)}| + |\eta^{(1)} T_a| \le |a_i^{(0)}| + O(\eta^{(1)} \sigma_w \sqrt{\log(Dm/\delta)}). \tag{135}$$

$$\left| \langle w_i^{(1)} - w_{m+i}^{(1)}, x \rangle \right| = 2 \left| \langle w_i^{(1)}, x \rangle \right| = O(\eta^{(1)}(1 + p_o\tilde{\sigma})/\gamma). \tag{136}$$

$$\left| b_i^{(1)} - b_{m+i}^{(1)} \right| = 2|\eta^{(1)} a_i^{(0)} T_b| = O(|\eta^{(1)} a_i^{(0)}| \epsilon_e). \tag{137}$$

Then we have

$$\left| g^{(1)}(x; \sigma_\xi^{(2)}) \right| \le O\left( m\eta^{(1)} \sigma_w \sqrt{\log(Dm/\delta)} \right) \left( \frac{\eta^{(1)}}{\gamma} + \frac{\sqrt{\log(m/\delta)}}{k^2} + \left| \eta^{(1)} a_i^{(0)} \frac{\epsilon_e}{\tilde{\sigma}} \right| + \sigma_\xi^{(2)} \right) \tag{138}$$

$$+ O\left( m(|a_i^{(0)}| + \eta^{(1)} \sigma_w \sqrt{\log(Dm/\delta)}) \right) \left( \frac{\eta^{(1)}}{\gamma} + \left| \eta^{(1)} a_i^{(0)} \frac{\epsilon_e}{\tilde{\sigma}} \right| \right) \tag{139}$$

$$= O\left( m\eta^{(1)} \sigma_w \frac{\log(Dm/\delta)}{k} + m|a_i^{(0)}| \left( \frac{\eta^{(1)}}{\gamma} + \left| \eta^{(1)} a_i^{(0)} \frac{\epsilon_e}{\tilde{\sigma}} \right| \right) \right) \tag{140}$$

$$= O\left( m\eta^{(1)} \sigma_w \frac{\log(Dm/\delta)}{k} + m|a_i^{(0)}| \frac{\eta^{(1)}}{\gamma} + m\sigma_a \eta^{(1)} \frac{k}{\gamma} \right) \tag{141}$$

$$< 1. \tag{142}$$

Then $\left| yg^{(1)}(x; \sigma_\xi^{(2)}) \right| < 1$. Finally, the statement on $\left| \langle (q_i^{(1)})_{-A}, \phi_{-A} \rangle \right|$ follows from a similar calculation on $\left| \langle w_i^{(1)}, x \rangle \right| = \left| \langle q_i^{(1)}, \phi \rangle \right|$. $\qquad\square$

We are now ready to analyze the gradients in the second gradient step.

**Lemma 20.** *Fix $\delta \in (0,1)$ and suppose $w_i^{(0)} \in \mathcal{G}_w(\delta), b_i^{(0)} \in \mathcal{G}_b(\delta), a_i^{(0)} \in \mathcal{G}_a(\delta)$ for all $i \in [2m]$. Let $\epsilon_{e2} := O\left( \frac{\eta^{(1)} |a_i^{(0)}| k(\gamma + \epsilon_e)}{\tilde{\sigma}^2 \sigma_\xi^{(2)}} \right) + \exp(-\Theta(k))$. If $k = \Omega(\log^2(Dm/\delta))$ and $k = O(D)$, $\sigma_a \le$*

$\tilde{\sigma}^2/(\gamma k^2)$, $\eta^{(1)} = O\left(\frac{\gamma}{km\sigma_a}\right)$, and $\sigma_\xi^{(2)} = 1/k^{3/2}$, then

$$\frac{\partial}{\partial w_i} L_\mathcal{D}(g^{(1)}; \sigma_\xi^{(2)}) = -a_i^{(1)} \sum_{j=1}^{D} M_j T_j \tag{143}$$

*where $T_j$ satisfies:*

- *if $j \in A$, then $|T_j - \beta\gamma/\tilde{\sigma}| \le O(\epsilon_{e2}/\tilde{\sigma} + \eta^{(1)}/\sigma_\xi^{(2)} + \eta^{(1)}|a_i^{(0)}|\epsilon_e/(\tilde{\sigma}\sigma_\xi^{(2)}))$, where $\beta \in [\Omega(1), 1]$ and depends only on $w_i^{(0)}, b_i^{(0)}$;*

- *if $j \notin A$, then $|T_j| \le \frac{1}{\tilde{\sigma}}\exp(-\Theta(k)) + O(\sigma_\phi^2\epsilon_{e2}\tilde{\sigma})$.*

*Proof of Lemma 20.* Consider one neuron index $i$ and omit the subscript $i$ in the parameters. By Lemma 19, $\Pr[yg^{(1)}(x; \xi^{(2)}) > 1] \le \exp(-\Theta(k))$. Let $\mathbb{I}_x = \mathbb{I}[yg^{(1)}(x; \xi^{(2)}) \le 1]$.

$$\frac{\partial}{\partial w} L_\mathcal{D}(g^{(1)}; \sigma_\xi^{(2)}) \tag{144}$$

$$= -a^{(1)} \mathbb{E}_{(x,y)\sim\mathcal{D}} \left\{ y\mathbb{I}_x \mathbb{E}_{\xi^{(2)}} \mathbb{I}[\langle w^{(1)}, x\rangle + b^{(1)} + \xi^{(2)} \in (0,1)]x \right\} \tag{145}$$

$$= -a^{(1)} \sum_{j=1}^{D} M_j \underbrace{\mathbb{E}_{(x,y)\sim\mathcal{D},\xi^{(2)}} \left\{ y\mathbb{I}_x \mathbb{I}[\langle w^{(1)}, x\rangle + b^{(1)} + \xi^{(2)} \in (0,1)]\phi_j \right\}}_{:=T_j}. \tag{146}$$

Let $T_{j1} := \mathbb{E}_{(x,y)\sim\mathcal{D},\xi^{(2)}} \left\{ y\mathbb{I}[\langle w^{(1)}, x\rangle + b^{(1)} + \xi^{(2)} \in (0,1)]\phi_j \right\}$. We have

$$|T_j - T_{j1}| \tag{147}$$

$$= \left| \mathbb{E}_{(x,y)\sim\mathcal{D},\xi^{(2)}} \left\{ y(1 - \mathbb{I}_x)\mathbb{I}[\langle w^{(1)}, x\rangle + b^{(1)} + \xi^{(2)} \in (0,1)]\phi_j \right\} \right| \tag{148}$$

$$\le \frac{1}{\tilde{\sigma}} \mathbb{E}_{(x,y)\sim\mathcal{D},\xi^{(2)}} |1 - \mathbb{I}_x| \tag{149}$$

$$\le \frac{1}{\tilde{\sigma}} \exp(-\Theta(k)). \tag{150}$$

So it is sufficient to bound $T_{j1}$. For simplicity, we use $q$ as a shorthand for $q_i^{(1)}$.

First, consider $j \in A$.

$$T_{j1} = \mathbb{E}_{(x,y)\sim\mathcal{D},\xi^{(2)}} \left\{ y\mathbb{I}[\langle w^{(1)}, x\rangle + b^{(1)} + \xi^{(2)} \in (0,1)]\phi_j \right\} \tag{151}$$

$$= \mathbb{E}_{\phi_A} \left\{ y\phi_j \Pr_{\phi_{-A},\xi^{(2)}} \left[ \langle\phi, q\rangle + b^{(1)} + \xi^{(2)} \in (0,1) \right] \right\}. \tag{152}$$

Let

$$I_a := \Pr_{\xi^{(2)}} \left[ \langle\phi, q\rangle + b^{(1)} + \xi^{(2)} \in (0,1) \right], \tag{153}$$

$$I_a' := \Pr_{\xi^{(2)}} \left[ \langle\phi_{-A}, q_{-A}\rangle + b^{(1)} + \xi^{(2)} \in (0,1) \right]. \tag{154}$$

By the property of the Gaussian $\xi^{(2)}$, that $|\langle\phi_A, q_A\rangle| = O(\frac{\eta^{(1)}|a_i^{(0)}|k(\gamma+\epsilon_e)}{\tilde{\sigma}^2})$, and that $|\langle\phi, q\rangle| = |\langle w_i^{(1)}, x\rangle| = O(\eta^{(1)}/\gamma) < O(1/k)$ and $|\langle\phi_{-A}, q_{-A}\rangle| = O(\eta^{(1)}/\gamma) < O(1/k)$, we have

$$|I_a - I_a'| \le \left| \Pr_{\xi^{(2)}} \left[ \langle\phi, q\rangle + b^{(1)} + \xi^{(2)} \ge 0 \right] - \Pr_{\xi^{(2)}} \left[ \langle\phi_{-A}, q_{-A}\rangle + b^{(1)} + \xi^{(2)} \ge 0 \right] \right| \tag{155}$$

$$+ \Pr_{\xi^{(2)}} \left[ \langle\phi, q\rangle + b^{(1)} + \xi^{(2)} \ge 1 \right] + \Pr_{\xi^{(2)}} \left[ \langle\phi_{-A}, q_{-A}\rangle + b^{(1)} + \xi^{(2)} \ge 1 \right] \tag{156}$$

$$= O\left( \frac{\eta^{(1)}|a_i^{(0)}|k(\gamma + \epsilon_e)}{\tilde{\sigma}^2\sigma_\xi^{(2)}} \right) + \exp(-\Theta(k)) = O(\epsilon_{e2}). \tag{157}$$

This leads to

$$\left| T_{j1} - \mathbb{E}_{\phi_A, \phi_{-A}} \left\{ y \phi_j I'_a \right\} \right| \tag{158}$$

$$\leq \mathbb{E}_{\phi_A} \left\{ |y \phi_j| \left| \mathbb{E}_{\phi_{-A}} (I_a - I'_a) \right| \right\} \tag{159}$$

$$\leq O(\epsilon_{e2}) \mathbb{E}_{\phi_A} \left\{ |y \phi_j| \right\} \tag{160}$$

$$\leq O(\epsilon_{e2}/\tilde{\sigma}) \tag{161}$$

where the last step is from Lemma 14. Furthermore,

$$\mathbb{E}_{\phi_A, \phi_{-A}} \left\{ y \phi_j I'_a \right\} \tag{162}$$

$$= \mathbb{E}_{\phi_A} \left\{ y \phi_j \right\} \mathbb{E}_{\phi_{-A}} [I'_a] \tag{163}$$

$$= \mathbb{E}_{\phi_A} \left\{ y \phi_j \right\} \Pr_{\phi_{-A}, \xi^{(2)}} \left[ \langle \phi_{-A}, q_{-A} \rangle + b^{(1)} + \xi^{(2)} \in (0,1) \right]. \tag{164}$$

By Lemma 19, we have $|\langle \phi_{-A}, q_{-A} \rangle| \leq O(\eta^{(1)} \tilde{\sigma}/\gamma)$. Also, $|b^{(1)} - b^{(0)}| \leq O(\eta^{(1)} |a_i^{(0)}| \epsilon_e)$. By the property of $\xi^{(2)}$,

$$\left| \Pr_{\xi^{(2)}} \left[ \langle \phi_{-A}, q_{-A} \rangle + b^{(1)} + \xi^{(2)} \in (0,1) \right] - \Pr_{\xi^{(2)}} \left[ b^{(0)} + \xi^{(2)} \in (0,1) \right] \right| \tag{165}$$

$$\leq O(\eta^{(1)} \tilde{\sigma}/(\gamma \sigma_\xi^{(2)})) + O(\eta^{(1)} |a_i^{(0)}| \epsilon_e / \sigma_\xi^{(2)}). \tag{166}$$

On the other hand,

$$\beta := \Pr_{\phi_{-A}, \xi^{(2)}} \left[ b^{(0)} + \xi^{(2)} \in (0,1) \right] = \Pr_{\xi^{(2)}} \left[ \xi^{(2)} \in (-b^{(0)}, 1 - b^{(0)}) \right] \tag{167}$$

$$= \Omega(1) \tag{168}$$

and $\beta$ only depends on $b^{(0)}$. By Lemma 14, $\mathbb{E}_{\phi_A} \left\{ y \phi_j \right\} = \gamma/\tilde{\sigma}$. Therefore,

$$|T_{j1} - \beta \gamma/\tilde{\sigma}| \leq O(\epsilon_{e2}/\tilde{\sigma}) + O(\eta^{(1)}/\sigma_\xi^{(2)}) + O(\eta^{(1)} |a_i^{(0)}| \epsilon_e / (\tilde{\sigma} \sigma_\xi^{(2)})). \tag{169}$$

Now, consider $j \notin A$. Let $B$ denote $A \cup \{j\}$.

$$T_{j1} = \mathbb{E}_{(x,y) \sim \mathcal{D}, \xi^{(2)}} \left\{ y \phi_j \mathbb{I} \left[ \langle \phi, q \rangle + b^{(1)} + \xi^{(2)} \in (0,1) \right] \right\} \tag{170}$$

$$= \mathbb{E}_{\phi_B} \mathbb{E}_{\phi_{-B}, \xi^{(2)}} \left\{ y \phi_j \mathbb{I} \left[ \langle \phi, q \rangle + b^{(1)} + \xi^{(2)} \in (0,1) \right] \right\} \tag{171}$$

$$= \mathbb{E}_{\phi_B} \left\{ y \phi_j \Pr_{\phi_{-B}, \xi^{(2)}} \left[ \langle \phi, q \rangle + b^{(1)} + \xi^{(2)} \in (0,1) \right] \right\}. \tag{172}$$

Let

$$I_b := \Pr_{\xi^{(2)}} \left[ \langle \phi, q \rangle + b^{(1)} + \xi^{(2)} \in (0,1) \right], \tag{173}$$

$$I'_b := \Pr_{\xi^{(2)}} \left[ \langle \phi_{-B}, q_{-B} \rangle + b^{(1)} + \xi^{(2)} \in (0,1) \right]. \tag{174}$$

Similar as above, we have $|I_b - I'_b| \leq \epsilon_{e2}$. Then by Lemma 14,

$$\left| T_{j1} - \mathbb{E}_{\phi_B, \phi_{-B}} \left\{ y \phi_j I'_b \right\} \right| \tag{175}$$

$$\leq \mathbb{E}_{\phi_B} \left\{ |y \phi_j| |\mathbb{E}_{\phi_{-B}} (I_b - I'_b)| \right\} \tag{176}$$

$$\leq O(\epsilon_{e2}) \mathbb{E}_{\phi_j} \left\{ |\phi_j| \right\} \tag{177}$$

$$\leq O(\epsilon_e) \times O(\sigma_\phi^2 \tilde{\sigma}) \tag{178}$$

$$= O(\sigma_\phi^2 \epsilon_{e2} \tilde{\sigma}). \tag{179}$$

Furthermore,

$$\mathbb{E}_{\phi_B, \phi_{-B}} \left\{ y \phi_j I'_b \right\} = \mathbb{E}_{\phi_A} \left\{ y \right\} \mathbb{E}_{\phi_j} \left\{ \phi_j \right\} \mathbb{E}_{\phi_{-B}} [I'_b] = 0. \tag{180}$$

Therefore,

$$|T_{j1}| \leq O(\sigma_\phi^2 \epsilon_{e2} \tilde{\sigma}). \tag{181}$$

$\square$

**Lemma 21.** *Under the same assumptions as in Lemma 20,*

$$\frac{\partial}{\partial b} L_{\mathcal{D}}(g^{(1)}; \sigma_\xi^{(2)}) = -a_i^{(1)} T_b \tag{182}$$

*where* $|T_b| \leq \exp(-\Omega(k)) + O(\epsilon_{e2})$.

*Proof of Lemma 21.* Consider one neuron index $i$ and omit the subscript $i$ in the parameters. By Lemma 19, $\Pr[yg^{(1)}(x; \xi^{(2)}) > 1] \leq \exp(-\Omega(k))$. Let $\mathbb{I}_x = \mathbb{I}[yg^{(1)}(x; \xi^{(2)}) \leq 1]$.

$$\frac{\partial}{\partial b} L_{\mathcal{D}}(g^{(1)}; \sigma_\xi^{(2)}) \tag{183}$$

$$= -a^{(1)} \underbrace{\mathbb{E}_{(x,y) \sim \mathcal{D}} \left\{ y \mathbb{I}_x \mathbb{E}_{\xi^{(2)}} \mathbb{I}[\langle w^{(1)}, x \rangle + b^{(1)} + \xi^{(2)} \in (0, 1)] \right\}}_{:=T_b}. \tag{184}$$

Let $T_{b1} := \mathbb{E}_{(x,y) \sim \mathcal{D}, \xi^{(2)}} \left\{ y \mathbb{I}[\langle w^{(1)}, x \rangle + b^{(1)} + \xi^{(2)} \in (0, 1)] \right\}$. We have

$$|T_b - T_{b1}| \tag{185}$$

$$= \left| \mathbb{E}_{(x,y) \sim \mathcal{D}, \xi^{(2)}} \left\{ y(1 - \mathbb{I}_x) \mathbb{I}[\langle w^{(1)}, x \rangle + b^{(1)} + \xi^{(2)} \in (0, 1)] \right\} \right| \tag{186}$$

$$\leq \mathbb{E}_{(x,y) \sim \mathcal{D}, \xi^{(2)}} |1 - \mathbb{I}_x| \tag{187}$$

$$\leq \exp(-\Omega(k)). \tag{188}$$

So it is sufficient to bound $T_{b1}$. For simplicity, we use $q$ as a shorthand for $q_i^{(1)}$.

$$T_{b1} = \mathbb{E}_{(x,y) \sim \mathcal{D}, \xi^{(2)}} \left\{ y \mathbb{I}\left[ \langle \phi, q \rangle + b^{(1)} + \xi^{(2)} \in (0, 1) \right] \right\} \tag{189}$$

$$= \mathbb{E}_{\phi_A} \mathbb{E}_{\phi_{-A}, \xi^{(2)}} \left\{ y \mathbb{I}\left[ \langle \phi, q \rangle + b^{(1)} + \xi^{(2)} \in (0, 1) \right] \right\} \tag{190}$$

$$= \mathbb{E}_{\phi_A} \left\{ y \Pr_{\phi_{-A}, \xi^{(2)}} \left[ \langle \phi, q \rangle + b^{(1)} + \xi^{(2)} \in (0, 1) \right] \right\}. \tag{191}$$

Let

$$I_b := \Pr_{\xi^{(2)}} \left[ \langle \phi, q \rangle + b^{(1)} + \xi^{(2)} \in (0, 1) \right], \tag{192}$$

$$I_b' := \Pr_{\xi^{(2)}} \left[ \langle \phi_{-A}, q_{-A} \rangle + b^{(1)} + \xi^{(2)} \in (0, 1) \right]. \tag{193}$$

Similar as in Lemma 20, we have $|I_b - I_b'| \leq \epsilon_{e2}$. Then by Lemma 14,

$$\left| T_{b1} - \mathbb{E}_{\phi_A, \phi_{-A}} \left\{ y I_b' \right\} \right| \tag{194}$$

$$\leq \mathbb{E}_{\phi_A} \left\{ |\mathbb{E}_{\phi_{-A}} (I_b - I_b')| \right\} \tag{195}$$

$$\leq O(\epsilon_{e2}). \tag{196}$$

Furthermore,

$$\mathbb{E}_{\phi_A, \phi_{-A}} \left\{ y I_b' \right\} = \mathbb{E}_{\phi_A} \left\{ y \right\} \mathbb{E}_{\phi_{-A}}[I_b'] = 0. \tag{197}$$

Therefore, $|T_{b1}| \leq O(\epsilon_{e2})$ and the statement follows. $\square$

**Lemma 22.** *Under the same assumptions as in Lemma 20,*

$$\frac{\partial}{\partial a_i} L_{\mathcal{D}}(g^{(1)}; \sigma_\xi^{(2)}) = -T_a \tag{198}$$

*where* $|T_a| = O(\eta^{(1)} \tilde{\sigma}/\gamma) + \exp(-\Omega(k)) \text{poly}(Dm)$.

*Proof of Lemma 22.* Consider one neuron index $i$ and omit the subscript $i$ in the parameters. By Lemma 19, $\Pr[yg^{(1)}(x;\xi^{(2)}) > 1] \leq \exp(-\Omega(k))$. Let $\mathbb{I}_x = \mathbb{I}[yg^{(1)}(x;\xi^{(2)}) \leq 1]$.

$$\frac{\partial}{\partial a} L_{\mathcal{D}}(g^{(1)};\sigma_\xi^{(2)}) \tag{199}$$

$$= -\underbrace{\mathbb{E}_{(x,y)\sim\mathcal{D}}\left\{y\mathbb{I}_x \mathbb{E}_{\xi^{(2)}}\sigma(\langle w^{(1)}, x\rangle + b^{(1)} + \xi^{(2)})\right\}}_{:=T_a}. \tag{200}$$

Let $T_{a1} := \mathbb{E}_{(x,y)\sim\mathcal{D}}\left\{y\mathbb{E}_{\xi^{(2)}}\sigma(\langle w^{(1)}, x\rangle + b^{(1)} + \xi^{(2)})\right\}$. We have

$$|T_a - T_{a1}| \tag{201}$$

$$= \left|\mathbb{E}_{(x,y)\sim\mathcal{D}}\left\{y(1-\mathbb{I}_x)\mathbb{E}_{\xi^{(2)}}\sigma(\langle w^{(1)}, x\rangle + b^{(1)} + \xi^{(2)})\right\}\right| \tag{202}$$

$$\leq \mathbb{E}_{(x,y)\sim\mathcal{D},\xi^{(2)}} |1 - \mathbb{I}_x| \tag{203}$$

$$\leq \exp(-\Omega(k)). \tag{204}$$

So it is sufficient to bound $T_{a1}$. For simplicity, we use $q$ as a shorthand for $q_i^{(1)}$.

Let $\phi_A'$ be an independent copy of $\phi_A$, $\phi'$ be the vector obtained by replacing in $\phi$ the entries $\phi_A$ with $\phi_A'$, and let $x' = M\phi'$ and its label is $y'$. Then

$$|T_{a1}| := \left|\mathbb{E}_{\phi_A}\left\{y\mathbb{E}_{\phi_{-A},\xi^{(2)}}\sigma(\langle w^{(1)}, x\rangle + b^{(1)} + \xi^{(2)})\right\}\right| \tag{205}$$

$$\leq \frac{1}{2}\left|\mathbb{E}_{\phi_A}\left\{\mathbb{E}_{\phi_{-A},\xi^{(2)}}\sigma(\langle w^{(1)}, x\rangle + b^{(1)} + \xi^{(1)})|y = 1\right\}\right. \tag{206}$$

$$\left. - \mathbb{E}_{\phi_A}\left\{\mathbb{E}_{\phi_{-A},\xi^{(2)}}\sigma(\langle w^{(1)}, x\rangle + b^{(1)} + \xi^{(2)})|y = -1\right\}\right| \tag{207}$$

$$\leq \frac{1}{2}\left|\mathbb{E}_{\phi_A}\left\{\mathbb{E}_{\phi_{-A},\xi^{(2)}}\sigma(\langle w^{(1)}, x\rangle + b^{(1)} + \xi^{(2)})|y = 1\right\}\right. \tag{208}$$

$$\left. - \mathbb{E}_{\phi_A'}\left\{\mathbb{E}_{\phi_{-A},\xi^{(2)}}\sigma(\langle w^{(1)}, x'\rangle + b^{(1)} + \xi^{(2)})|y' = -1\right\}\right| \tag{209}$$

$$\leq \frac{1}{2}\mathbb{E}_{\phi_A,\phi_A'}\left\{\mathbb{E}_{\phi_{-A}}\left|\langle w^{(1)}, x\rangle - \langle w^{(1)}, x'\rangle\right| \Big| y = 1, y' = -1\right\} \tag{210}$$

$$\leq \frac{1}{2}\mathbb{E}_{\phi_{-A}}\left(\mathbb{E}_{\phi_A}\left\{\left|\langle w^{(1)}, x\rangle\right| \Big| y = 1\right\} + \mathbb{E}_{\phi_A'}\left\{\left|\langle w^{(1)}, x'\rangle\right| \Big| y' = -1\right\}\right) \tag{211}$$

$$\leq \mathbb{E}_{\phi_{-A},\phi_A}\left|\langle w^{(1)}, x\rangle\right| \tag{212}$$

$$= \mathbb{E}_x\left|\langle w^{(1)}, x\rangle\right| \tag{213}$$

$$= O(\eta^{(1)}\tilde{\sigma}/\gamma) + \exp(-\Omega(k)) \times D \times \|q^{(1)}\|_\infty\|\phi\|_\infty \tag{214}$$

$$= O(\eta^{(1)}\tilde{\sigma}/\gamma) + \exp(-\Omega(k))\frac{D|\eta^{(1)}a^{(0)}|(\gamma + \epsilon_e)}{\tilde{\sigma}^2} \tag{215}$$

$$= O(\eta^{(1)}\tilde{\sigma}/\gamma) + \exp(-\Omega(k))\text{poly}(Dm) \tag{216}$$

where the fourth step follows from that $\sigma$ is 1-Lipschitz, the third to the last step from Lemma 19, and the second to the last step from Lemma 18. $\qquad\square$

With the above lemmas about the gradients, we are now ready to show that at the end of the second step, we get a good set of features for accurate prediction.

**Lemma 23.** *Set*

$$\eta^{(1)} = \frac{\gamma^2\tilde{\sigma}}{km^3}, \lambda_a^{(1)} = 0, \lambda_w^{(1)} = 1/(2\eta^{(1)}), \sigma_\xi^{(1)} = 1/k^{3/2}, \tag{217}$$

$$\eta^{(2)} = 1, \lambda_a^{(2)} = \lambda_w^{(2)} = 1/(2\eta^{(2)}), \sigma_\xi^{(2)} = 1/k^{3/2}. \tag{218}$$

*Fix* $\delta \in (0, O(1/k^3))$. *If* $p_o = \Omega(k^2/D)$, $k = \Omega\left(\log^2\left(\frac{Dm}{\delta\gamma}\right)\right)$, *and* $m \geq \max\{\Omega(k^4), D\}$, *then with probability at least* $1 - \delta$ *over the initialization, there exist* $\tilde{a}_i$'s *such that* $\tilde{g}(x) := \sum_{i=1}^{2m} \tilde{a}_i \sigma(\langle w_i^{(2)}, x \rangle + b_i^{(2)})$ *satisfies* $L_\mathcal{D}(\tilde{g}) = 0$. *Furthermore,* $\|\tilde{a}\|_0 = O(m/k)$, $\|\tilde{a}\|_\infty = O(k^5/m)$, *and* $\|\tilde{a}\|_2^2 = O(k^9/m)$. *Finally,* $\|a^{(2)}\|_\infty = O\left(\frac{1}{km^2}\right)$, $\|w_i^{(2)}\|_2 = O(\tilde{\sigma}/k)$, *and* $|b_i^{(2)}| = O(1/k^2)$ *for all* $i \in [2m]$.

*Proof of Lemma 23.* By Lemma 5, there exists a network $g^*(x) = \sum_{\ell=1}^{3(k+1)} a_\ell^* \sigma(\langle w_\ell^*, x \rangle + b_\ell^*)$ satisfying $g^*(x)$ for all $(x, y) \sim \mathcal{D}$. Furthermore, $|a_i^*| \leq 32k, 1/(32k) \leq |b_i^*| \leq 1/2$, $w_i^* = \tilde{\sigma} \sum_{j \in A} M_j/(4k)$, and $|\langle w_i^*, x \rangle + b_i^*| \leq 1$ for any $i \in [n]$ and $(x, y) \sim \mathcal{D}$. Now we fix an $\ell$, and show that with high probability there is a neuron in $g^{(2)}$ that can approximate the $\ell$-th neuron in $g^*$.

By Lemma 10, with probability $1 - 2\delta$ over $w_i^{(0)}$'s, they are all in $\mathcal{G}_w(\delta)$; with probability $1 - \delta$ over $a_i^{(0)}$'s, they are all in $\mathcal{G}_a(\delta)$; with probability $1 - \delta$ over $b_i^{(0)}$'s, they are all in $\mathcal{G}_b(\delta)$. Under these events, by Lemma 18, Lemma 20 and 21, for any neuron $i \in [2m]$, we have

$$w_i^{(2)} = a_i^{(1)} \sum_{j=1}^{D} M_j T_j, \tag{219}$$

$$b_i^{(2)} = b_i^{(1)} + a_i^{(1)} T_b. \tag{220}$$

where

- if $j \in A$, then $|T_j - \beta\gamma/\tilde{\sigma}| \leq \epsilon_{w1} := O(\epsilon_{e2}/\tilde{\sigma} + \eta^{(1)}/\sigma_\xi^{(2)} + \eta^{(1)}|a_i^{(0)}|\epsilon_e/(\tilde{\sigma}\sigma_\xi^{(2)}))$, where $\beta \in [\Omega(1), 1]$ and depends only on $w_i^{(0)}, b_i^{(0)}$;

- if $j \notin A$, then $|T_j| \leq \epsilon_{w2} := \frac{1}{\tilde{\sigma}} \exp(-\Theta(k)) + O(\sigma_\phi^2 \epsilon_{e2} \tilde{\sigma})$.

- $|T_b| \leq \epsilon_b := \frac{1}{\tilde{\sigma}} \exp(-\Theta(k)) + O(\epsilon_{e2})$.

Given the initialization, with probability $\Omega(1)$ over $b_i^{(0)}$, we have

$$|b_i^{(0)}| \in \left[\frac{1}{2k^2}, \frac{2}{k^2}\right], \text{sign}(b_i^{(0)}) = \text{sign}(b_\ell^*). \tag{221}$$

Finally, since $\frac{4k|b_\ell^*|\beta\gamma}{|b_i^{(0)}|\tilde{\sigma}^2} \in [\Omega(k^2\gamma/\tilde{\sigma}^2), O(k^3\gamma/\tilde{\sigma}^2)]$ and depends only on $w_i^{(0)}, b_i^{(0)}$, we have that for $\epsilon_a = \Theta(1/k^2)$, with probability $\Omega(\epsilon_a) > \delta$ over $a_i^{(0)}$,

$$\left|\frac{4k|b_\ell^*|\beta\gamma}{|b_i^{(0)}|\tilde{\sigma}^2} a_i^{(0)} - 1\right| \leq \epsilon_a, \quad |a_i^{(0)}| = O\left(\frac{\tilde{\sigma}^2}{k^2\gamma}\right). \tag{222}$$

Let $n_a = \epsilon_a m/4$. For the given value of $m$, by (219)-(222) we have with probability $\geq 1 - 5\delta$ over the initialization, for each $\ell$ there is a different set of neurons $I_\ell \subseteq [m]$ with $|I_\ell| = n_a$ and such that for each $i_\ell \in I_\ell$,

$$|b_{i_\ell}^{(0)}| \in \left[\frac{1}{2k^2}, \frac{2}{k^2}\right], \quad \text{sign}(b_{i_\ell}^{(0)}) = \text{sign}(b_\ell^*), \tag{223}$$

$$\left|\frac{4k|b_\ell^*|\beta\gamma}{|b_{i_\ell}^{(0)}|\tilde{\sigma}^2} a_{i_\ell}^{(0)} - 1\right| \leq \epsilon_a, \quad |a_{i_\ell}^{(0)}| = O\left(\frac{\tilde{\sigma}^2}{k^2\gamma}\right). \tag{224}$$

We also have

$$\left| \langle w_{i_\ell}^{(2)}, x \rangle - \frac{a_{i_\ell}^{(0)} \beta \gamma}{\tilde{\sigma}} \sum_{j \in A} \langle M_j, x \rangle \right| \tag{225}$$

$$\leq \left| \langle w_{i_\ell}^{(2)}, x \rangle - \frac{a_{i_\ell}^{(1)} \beta \gamma}{\tilde{\sigma}} \sum_{j \in A} \langle M_j, x \rangle \right| + \left| \frac{a_{i_\ell}^{(1)} \beta \gamma}{\tilde{\sigma}} \sum_{j \in A} \langle M_j, x \rangle - \frac{a_{i_\ell}^{(0)} \beta \gamma}{\tilde{\sigma}} \sum_{j \in A} \langle M_j, x \rangle \right| \tag{226}$$

$$= \left| a_{i_\ell}^{(1)} \sum_{j=1}^{D} T_j \phi_j - \frac{a_{i_\ell}^{(1)} \beta \gamma}{\tilde{\sigma}} \sum_{j \in A} \phi_j \right| + \left| a_{i_\ell}^{(1)} - a_{i_\ell}^{(0)} \right| \left| \frac{\beta \gamma}{\tilde{\sigma}} \sum_{j \in A} \phi_j \right| \tag{227}$$

$$= \left| a_{i_\ell}^{(1)} \right| \left| \sum_{j=1}^{D} T_j \phi_j - \frac{\beta \gamma}{\tilde{\sigma}} \sum_{j \in A} \phi_j \right| + \left| a_{i_\ell}^{(1)} - a_{i_\ell}^{(0)} \right| \left| \frac{\beta \gamma}{\tilde{\sigma}} \sum_{j \in A} \phi_j \right|. \tag{228}$$

We have $\left| a_{i_\ell}^{(1)} - a_{i_\ell}^{(0)} \right| = O(\eta^{(1)} \sigma_w \sqrt{\log(Dm/\delta)})$, and

$$\left| \sum_{j=1}^{D} T_j \phi_j - \frac{\beta \gamma}{\tilde{\sigma}} \sum_{j \in A} \phi_j \right| \leq \left| \sum_{j \in A} (T_j - \frac{\beta \gamma}{\tilde{\sigma}}) \phi_j \right| + \left| \sum_{j \notin A} T_j \phi_j \right| \tag{229}$$

$$\leq O(k \epsilon_{w1} / \tilde{\sigma}) + O(D \epsilon_{w2} / \tilde{\sigma}) =: \epsilon_\phi. \tag{230}$$

For the given values of parameters, we have

$$\epsilon_{e2} = O\left( \frac{\gamma}{m^2} \right), \tag{231}$$

$$\epsilon_{w1} = O\left( \frac{k\gamma}{m^2 \tilde{\sigma}} + \frac{\gamma \epsilon_e}{km^2} \right), \tag{232}$$

$$\epsilon_{w2} = O\left( \frac{\gamma}{m^2 \tilde{\sigma}} \right), \tag{233}$$

$$\epsilon_b = O\left( \frac{\gamma}{m^2} \right), \tag{234}$$

$$\epsilon_\phi = O\left( \frac{k^2 \gamma}{m^2 \tilde{\sigma}^2} + \frac{\gamma \epsilon_e}{m^2 \tilde{\sigma}} + \frac{\gamma}{m \tilde{\sigma}^2} \right). \tag{235}$$

Therefore,

$$\left| \langle w_{i_\ell}^{(2)}, x \rangle - \frac{a_{i_\ell}^{(0)} \beta \gamma}{\tilde{\sigma}} \sum_{j \in A} \langle M_j, x \rangle \right| \tag{236}$$

$$\leq \left| a_{i_\ell}^{(1)} \right| \epsilon_\phi + \left| a_{i_\ell}^{(1)} - a_{i_\ell}^{(0)} \right| \frac{k\gamma}{\tilde{\sigma}^2} \tag{237}$$

$$\leq O\left( \frac{\tilde{\sigma}^2}{k^2 \gamma} + \eta^{(1)} \sigma_w \sqrt{\log(Dm/\delta)} \right) \left( \frac{k^2 \gamma}{m^2 \tilde{\sigma}^2} + \frac{\gamma \epsilon_e}{m^2 \tilde{\sigma}} + \frac{\gamma}{m \tilde{\sigma}^2} \right) \tag{238}$$

$$+ O\left( \eta^{(1)} \sigma_w \sqrt{\log(Dm/\delta)} \right) \frac{k\gamma}{\tilde{\sigma}^2} \tag{239}$$

$$\leq O\left( \frac{1}{m} \right). \tag{240}$$

We also have by Lemma 18 and 21:

$$|b_{i_\ell}^{(2)} - b_{i_\ell}^{(0)}| \leq O\left( \eta^{(1)} |a_{i_\ell}^{(0)}| \epsilon_e + |a_{i_\ell}^{(1)}| \left( \frac{1}{\tilde{\sigma}} \exp(-\Theta(k)) + \epsilon_{e2} \right) \right) \leq O\left( \frac{1}{m} \right). \tag{241}$$

Now, construct $\tilde{a}$ such that $\tilde{a}_{i_\ell} = \frac{2a_\ell^* |b_\ell^*|}{|b_{i_\ell}^{(0)}| n_a}$ for each $\ell$ and each $i_\ell \in I_\ell$, and $\tilde{a}_i = 0$ elsewhere. Then

$$|\tilde{g}(x) - 2g^*(x)| \tag{242}$$

$$= \left| \sum_{\ell=1}^{3(k+1)} \sum_{i_\ell \in I_\ell} \tilde{a}_{i_\ell} \sigma\left(\langle w_{i_\ell}^{(2)}, x\rangle + b_{i_\ell}^{(2)}\right) - \sum_{\ell=1}^{3(k+1)} 2a_\ell^* \sigma\left(\langle w_\ell^*, x\rangle + b_\ell^*\right)\right| \tag{243}$$

$$= \left| \sum_{\ell=1}^{3(k+1)} \sum_{i_\ell \in I_\ell} \frac{2a_\ell^* |b_\ell^*|}{|b_{i_\ell}^{(0)}| n_a} \sigma\left(\langle w_{i_\ell}^{(2)}, x\rangle + b_{i_\ell}^{(2)}\right) - \sum_{\ell=1}^{3(k+1)} \sum_{i_\ell \in I_\ell} \frac{2a_\ell^* |b_\ell^*|}{|b_{i_\ell}^{(0)}| n_a} \sigma\left(\frac{|b_{i_\ell}^{(0)}|}{|b_\ell^*|}\langle w_\ell^*, x\rangle + b_{i_\ell}^{(0)}\right)\right| \tag{244}$$

$$\leq \left| \sum_{\ell=1}^{3(k+1)} \sum_{i_\ell \in I_\ell} \frac{1}{n_a} \left( \frac{2a_\ell^* |b_\ell^*|}{|b_{i_\ell}^{(0)}|} \sigma\left(\langle w_{i_\ell}^{(2)}, x\rangle + b_{i_\ell}^{(2)}\right) - \frac{2a_\ell^* |b_\ell^*|}{|b_{i_\ell}^{(0)}|} \sigma\left(\frac{a_{i_\ell}^{(0)} \beta\gamma}{\tilde{\sigma}} \sum_{j \in A}\langle M_j, x\rangle + b_{i_\ell}^{(0)}\right)\right)\right| \tag{245}$$

$$+ \left| \sum_{\ell=1}^{3(k+1)} \sum_{i_\ell \in I_\ell} \frac{1}{n_a} \left( \frac{2a_\ell^* |b_\ell^*|}{|b_{i_\ell}^{(0)}|} \sigma\left(\frac{a_{i_\ell}^{(0)} \beta\gamma}{\tilde{\sigma}} \sum_{j \in A}\langle M_j, x\rangle + b_{i_\ell}^{(0)}\right) - \frac{2a_\ell^* |b_\ell^*|}{|b_{i_\ell}^{(0)}|} \sigma\left(\frac{|b_{i_\ell}^{(0)}|}{|b_\ell^*|}\langle w_\ell^*, x\rangle + b_{i_\ell}^{(0)}\right)\right)\right| \tag{246}$$

$$\leq 3(k+1) \max_\ell \frac{2a_\ell^* |b_\ell^*|}{|b_{i_\ell}^{(0)}|} O\left(\frac{1}{m}\right) + \tag{247}$$

$$3(k+1) \max_\ell \frac{2a_\ell^* |b_\ell^*|}{|b_{i_\ell}^{(0)}|} \frac{\tilde{\sigma} |b_{i_\ell}^{(0)}|}{4k|b_\ell^*|} \left| \frac{4k a_{i_\ell}^{(0)} \beta\gamma |b_\ell^*|}{\tilde{\sigma}^2 |b_{i_\ell}^{(0)}|} - 1\right| \frac{k}{\tilde{\sigma}} \tag{248}$$

$$= O\left(\frac{k^4}{m} + k^2 \epsilon_a\right) \tag{249}$$

$$\leq 1. \tag{250}$$

Here the second equation follows from that $\sigma$ is positive-homogeneous in $[0, 1]$, $|\langle w_\ell^*, x\rangle + b_\ell^*| \leq 1$, $|b_{i_\ell}^{(0)}|/|b_\ell^*| \leq 1$. This guarantees $y\tilde{g}(x) \geq 1$. Changing the scaling of $\delta$ leads to the statement.

Finally, the bounds on $\tilde{a}$ follow from the above calculation. The bound on $\|a^{(2)}\|_2$ follows from Lemma 22, and those on $\|w_i^{(2)}\|_2$ and $\|b_i^{(2)}\|_2$ follow from (219)(220) and the bounds on $a_i^{(1)}$ and $b_i^{(1)}$ in Lemma 18. $\qquad\square$

## B.6 CLASSIFIER LEARNING STAGE

Once we have a set of good features, we are now ready to prove that the later steps will learn an accurate classifier. The intuition is that the first layer's weights do not change much and the second layer's weights get updated till achieving good accuracy. In particular, we will employ the online optimization technique from Daniely & Malach (2020).

We begin by showing that the first layer's weights do not change too much.

**Lemma 24.** *Assume the same conditions as in Lemma 23. Suppose for $t > 2$, $\lambda_a^{(t)} = \lambda_w^{(t)} = \lambda$, $\eta^{(t)} = \eta$ for some $\lambda, \eta \in (0, 1)$, and $\sigma_\xi^{(t)} = 0$. Then for any $t > 2$ and $i \in [2m]$,*

$$|a_i^{(t)}| \leq \eta t + O\left(\frac{1}{km^2}\right), \tag{251}$$

$$\|w_i^{(t)} - w_i^{(2)}\|_2 \leq O\left(\frac{t\eta\lambda\tilde{\sigma}}{k}\right) + \eta^2 t^2 + O\left(\frac{t}{km^2}\right), \tag{252}$$

$$|b_i^{(t)} - b_i^{(2)}| \leq O\left(\frac{t\eta\lambda}{k^2}\right) + \eta^2 t^2 + O\left(\frac{t}{km^2}\right). \tag{253}$$

*Proof of Lemma 24.* First, we bound the size of $|a_i^{(t)}|$:

$$|a_i^{(t)}| = \left| (1 - 2\eta\lambda)a_i^{(t-1)} - \eta\frac{\partial}{\partial a_i}L_\mathcal{D}(g^{(t-1)}) \right| \tag{254}$$

$$\leq \left| (1 - 2\eta\lambda)a_i^{(t-1)} - \eta\mathbb{E}_{(x,y)\sim\mathcal{D}}\left\{ y\mathbb{I}[yg^{(t-1)}(x) \leq 1]\sigma(\langle w_i^{(t-1)}, x\rangle + b_i^{(t-1)}) \right\} \right| \tag{255}$$

$$\leq |a_i^{(t-1)}| + \eta \tag{256}$$

which leads to

$$|a_i^{(t)}| \leq \eta t + |a_i^{(2)}| \tag{257}$$

where $|a_i^{(2)}| = O\left(\frac{1}{km^2}\right)$. We are now to bound the change of $w_i^{(t)}$ and $b_i^{(t)}$.

$$\|w_i^{(t)} - w_i^{(2)}\|_2 \tag{258}$$

$$= \left\| (1 - 2\eta\lambda)w_i^{(t-1)} - \eta\frac{\partial}{\partial w_i}L_\mathcal{D}(g^{(t-1)}) - w_i^{(2)} \right\|_2 \tag{259}$$

$$\leq \left\| (1 - 2\eta\lambda)w_i^{(t-1)} \right. \tag{260}$$

$$\left. + \eta a_i^{(t-1)}\mathbb{E}_{(x,y)\sim\mathcal{D}}\left\{ y\mathbb{I}[yg^{(t-1)}(x) \leq 1]\mathbb{I}[\langle w_i^{(t-1)}, x\rangle + b_i^{(t-1)} \in (0,1)]x \right\} - w_i^{(2)} \right\|_2 \tag{261}$$

$$\leq \left\| (1 - 2\eta\lambda)w_i^{(t-1)} - w_i^{(2)} \right\|_2 \tag{262}$$

$$+ \eta \left\| a_i^{(t-1)}\mathbb{E}_{(x,y)\sim\mathcal{D}}\left\{ y\mathbb{I}[yg^{(t-1)}(x) \leq 1]\mathbb{I}[\langle w_i^{(t-1)}, x\rangle + b_i^{(t-1)} \in (0,1)]x \right\} \right\|_2 \tag{263}$$

$$\leq (1 - 2\eta\lambda)\left\| w_i^{(t-1)} - w_i^{(2)} \right\|_2 + 2\eta\lambda\left\| w_i^{(2)} \right\|_2 + \eta\left| a_i^{(t-1)} \right| \tag{264}$$

leading to

$$\|w_i^{(t)} - w_i^{(2)}\|_2 \leq 2t\eta\lambda\left\| w_i^{(2)} \right\|_2 + \eta^2 t^2 + t|a_i^{(2)}|. \tag{265}$$

Note that $\|w_i^{(2)}\|_2 = O(\tilde{\sigma}/k)$.

$$|b_i^{(t)} - b_i^{(2)}| = \left| b_i^{(t-1)} - \eta\frac{\partial}{\partial b_i}L_\mathcal{D}(g^{(t-1)}) - b_i^{(2)} \right| \tag{266}$$

$$\leq \left| b_i^{(t-1)} - b_i^{(2)} \right| \tag{267}$$

$$+ \eta\left| a_i^{(t-1)}\mathbb{E}_{(x,y)\sim\mathcal{D}}\left\{ y\mathbb{I}[yg^{(t-1)}(x) \leq 1]\mathbb{I}[\langle w_i^{(t-1)}, x\rangle + b_i^{(t-1)} \in (0,1)] \right\} \right| \tag{268}$$

$$\leq \left| b_i^{(t-1)} - b_i^{(2)} \right| + \eta\left| a_i^{(t-1)} \right| \tag{269}$$

leading to

$$|b_i^{(t)} - b_i^{(2)}| \leq \eta^2 t^2 + t|a_i^{(2)}|. \tag{270}$$

Note that $\left| b_i^{(2)} \right| = O(1/k^2)$. $\qquad\square$

**Lemma 25.** *Assume the same conditions as in Lemma 24. Let $g_{\tilde{a}}^{(t)}(x) = \sum_{i=1}^m \tilde{a}_i\sigma(\langle w_i^{(t)}, x\rangle + b_i^{(t)})$. Then*

$$|\ell(g_{\tilde{a}}^{(t)}(x), y) - \ell(g_{\tilde{a}}^{(2)}(x), y)| \leq \|\tilde{a}\|_2\sqrt{\|\tilde{a}\|_0}\left( O\left(\frac{t\eta\lambda\tilde{\sigma}}{k}\right) + \eta^2 t^2 + O\left(\frac{t}{km^2}\right) \right). \tag{271}$$

*Proof of Lemma 25.* It follows from that

$$|\ell(g_{\tilde{a}}^{(t)}(x), y) - \ell(g_{\tilde{a}}^{(2)}(x))| \tag{272}$$

$$\leq |g_{\tilde{a}}^{(t)}(x) - g_{\tilde{a}}^{(2)}(x)| \tag{273}$$

$$\leq \|\tilde{a}\|_2 \sqrt{\|\tilde{a}\|_0} \max_{i \in [2m]} \left| \sigma(\langle w_i^{(t)}, x \rangle + b_i^{(t)}) - \sigma(\langle w_i^{(2)}, x \rangle + b_i^{(2)}) \right| \tag{274}$$

$$\leq \|\tilde{a}\|_2 \sqrt{\|\tilde{a}\|_0} \max_{i \in [2m]} \left( \left| \langle w_i^{(t)} - w_i^{(2)}, x \rangle \right| + \left| b_i^{(t)} - b_i^{(2)} \right| \right). \tag{275}$$

and Lemma 24. $\qquad \square$

## B.7 PROOF OF THEOREM 1

Based on the above lemmas, following the same argument as in the proof of Theorem 2 in Daniely & Malach (2020), we get our main theorem.

**Theorem 26** (Full version of Theorem 1). *Set*

$$\eta^{(1)} = \frac{\gamma^2 \tilde{\sigma}^2}{km^3}, \lambda_a^{(1)} = 0, \lambda_w^{(1)} = 1/(2\eta^{(1)}), \sigma_\xi^{(1)} = 1/k^2, \tag{276}$$

$$\eta^{(2)} = 1, \lambda_a^{(2)} = \lambda_w^{(2)} = 1/(2\eta^{(2)}), \sigma_\xi^{(2)} = 1/k^2, \tag{277}$$

$$\eta^{(t)} = \eta = \frac{k^2}{Tm^{1/3}}, \lambda_a^{(t)} = \lambda_w^{(t)} = \lambda \leq \frac{k^3}{\tilde{\sigma}m^{1/3}}, \sigma_\xi^{(t)} = 0, \text{ for } 2 < t \leq T. \tag{278}$$

*For any $\delta \in (0,1)$, if $p_o = \Omega(k^2/D)$, $k = \Omega\left(\log^2\left(\frac{D}{\delta\gamma}\right)\right)$, $\max\{\Omega(k^4), D\} \leq m \leq poly(D)$, then we have for any $\mathcal{D} \in \mathcal{F}_\Xi$, with probability at least $1 - \delta$, there exists $t \in [T]$ such that*

$$\Pr[\text{sign}(g^{(t)}(x)) \neq y] \leq L_\mathcal{D}(g^{(t)}) = O\left(\frac{k^8}{m^{2/3}} + \frac{k^3 T}{m^2} + \frac{k^2 m^{2/3}}{T}\right). \tag{279}$$

*Consequently, for any $\epsilon \in (0,1)$, if $T = m^{4/3}$, and $\max\{\Omega(k^{12}/\epsilon^{3/2}), D\} \leq m \leq poly(D)$, then*

$$\Pr[\text{sign}(g^{(t)}(x)) \neq y] \leq L_\mathcal{D}(g^{(t)}) \leq \epsilon. \tag{280}$$

*Proof of Theorem 1.* Consider $\tilde{L}_\mathcal{D}(g^{(t)}) = \mathbb{E}[\ell(g^{(t)}, y)] + \lambda_a^{(t)} \|a^{(t)}\|_2^2$. Note that the gradient update using $\tilde{L}_\mathcal{D}(g^{(t)})$ is the same as the update in our learning algorithm. Then by Theorem 27, Lemma 23, and Lemma 25,

$$\frac{1}{T} \sum_{t=3}^{T} \tilde{L}_\mathcal{D}(g^{(t)}) \leq \frac{\|\tilde{a}\|_2^2}{2} + \|\tilde{a}\|_2 \sqrt{\|\tilde{a}\|_0} \left( O\left(\frac{T\eta\lambda\tilde{\sigma}}{k}\right) + \eta^2 T^2 + O\left(\frac{T}{km^2}\right) \right) \tag{281}$$

$$+ \frac{\|\tilde{a}\|_2^2}{2\eta T} + \|a^{(2)}\|_2 \sqrt{m} + \eta m \tag{282}$$

$$\leq O\left(\frac{k^9}{m} + k^4 \eta^2 T^2 + \frac{k^3 T}{m^2} + \frac{k^9}{\eta T m} + \eta m\right). \tag{283}$$

$$\leq O\left(\frac{k^8}{m^{2/3}} + \frac{k^3 T}{m^2} + \frac{k^2 m^{2/3}}{T}\right). \tag{284}$$

The statement follows from that 0-1 classification error is bounded by the hinge-loss. $\qquad \square$

**Theorem 27** (Theorem 13 in Daniely & Malach (2020)). *Fix some $\eta$, and let $f_1, \ldots, f_T$ be some sequence of convex functions. Fix some $\theta_1$, and assume we update $\theta_{t+1} = \theta_t - \eta \nabla f_t(\theta_t)$. Then for every $\theta^*$ the following holds:*

$$\frac{1}{T} \sum_{t=1}^{T} f_t(\theta_t) \leq \frac{1}{T} \sum_{t=1}^{T} f_t(\theta^*) + \frac{1}{2\eta T} \|\theta^*\|_2^2 + \|\theta_1\|_2 \frac{1}{T} \sum_{t=1}^{T} \|\nabla f_t(\theta_t)\|_2 + \eta \frac{1}{T} \sum_{t=1}^{T} \|\nabla f_t(\theta_t)\|_2^2. \tag{285}$$

## C    LOWER BOUND FOR LINEAR MODELS ON FIXED FEATURE MAPPINGS

**Theorem 28** (Restatement of Theorem 2). *Suppose $\Psi$ is a data-independent feature mapping of dimension $N$ with bounded features, i.e., $\Psi : \mathcal{X} \to [-1, 1]^N$. Define for $B > 0$:*

$$\mathcal{H}_B = \{h(\tilde{x}) : h(\tilde{x}) = \langle \Psi(\tilde{x}), w \rangle, \|w\|_2 \leq B\}. \tag{286}$$

*Then, if $3 < k \leq D/16$ and $k$ is odd, then there exists $\mathcal{D} \in \mathcal{F}_\Xi$ such that all $h \in \mathcal{H}_B$ have hinge-loss at least $p_o \left(1 - \frac{\sqrt{2N}B}{2^k}\right)$.*

*Proof of Theorem 2.* We first show that $\mathcal{F}_\Xi$ contains some distributions that are essentially sparse parity learning problems, and then we invoke the lower bound result from existing work for such problems.

Consider $\mathcal{D}$ defined as follows.

- Let $P = \{i \in [k] : i \text{ is odd}\}$. That is, if there are odd numbers of 1's in $\tilde{\phi}_A$, then $y = +1$.

- Let $\mathcal{D}_{\tilde{\phi}}^{(0)}$ be a distribution where all entries $\tilde{\phi}_j$ are i.i.d. with $\Pr[\tilde{\phi}_j = 0] = \Pr[\tilde{\phi}_j = 1] = 1/2$. Let $\mathcal{D}^{(0)}$ be the distribution over $(\tilde{x}, y)$ induced by $\mathcal{D}_{\tilde{\phi}}^{(0)}$ and the above $P$.

- Let $\mathcal{D}_{\tilde{\phi}}^{(1)}$ be a distribution where all entries $\tilde{\phi}_j$ for $j \notin A$ are i.i.d. with $\Pr[\tilde{\phi}_j = 1] = p_o/(2 - 2p_o)$, while $\Pr[\tilde{\phi}_A = (0, 0, \ldots, 0)] = \Pr[\tilde{\phi}_A = (1, 1, \ldots, 1)] = 1/2$. Let $\mathcal{D}^{(1)}$ be the distribution over $(\tilde{x}, y)$ induced by $\mathcal{D}_{\tilde{\phi}}^{(1)}$ and the above $P$.

- Let $\mathcal{D}_A^{\text{mix}} = p_o \mathcal{D}^{(0)} + (1 - p_o)\mathcal{D}^{(1)}$.

It can be verified that such distributions are included in $\mathcal{F}_\Xi$ for $\gamma = \Theta(1)$.

Assume for contradiction that for all $\mathcal{D} \in \mathcal{F}_\Xi$, there exists $h^* \in \mathcal{H}_B$ such that $h = \langle \Psi, w^* \rangle$ loss smaller than $p_o \left(1 - \frac{\sqrt{2N}B}{2^k}\right)$. Then for all the distributions $\mathcal{D}_A^{\text{mix}}$ defined above, we have

$$\mathbb{E}_{\mathcal{D}^{(0)}}[\ell(h^*(\tilde{x}), y)] < 1 - \frac{\sqrt{2N}B}{2^k}. \tag{287}$$

Now let $\mathcal{D}_z$ be a distribution over $z \in \{-1, +1\}^D$ with i.i.d. entries $z_j$ and $\Pr[z_j = -1] = \Pr[z_j = +1] = 1/2$. Let $f_A(z) = \prod_{j \in A} z_j$ be the $k$-sparse parity functions. Let $\Psi'(z) = \Psi(M(z + 1)/2)$. Then we have $h'(z) = \langle \Psi'(z), w^* \rangle$ such that for all $A$,

$$\mathbb{E}_{\mathcal{D}_z}[\ell(h'(z), f_A(z))] < 1 - \frac{\sqrt{2N}B}{2^k}. \tag{288}$$

This is contradictory to Theorem 29. $\qquad\square$

The following theorem is implicit in the proof in Theorem 1 in Daniely & Malach (2020).

**Theorem 29.** *For a subset $A \subseteq [D]$ of size $k$, let the distribution $\mathcal{D}_A$ over $(z, y)$ defined as follows: $z$ is uniform over $\{\pm 1\}^D$ and $y = \prod_{i \in A} z_i$. Fix some $\Psi : \{\pm 1\}^D \to [-1, +1]^N$, and define:*

$$\mathcal{H}_\Psi^B = \{z \to \langle \Psi(z), w \rangle : \|w\|_2 \leq B\}.$$

*If $k$ is odd and $k \leq D/16$, then there exists some $A$ such that*

$$\min_{h \in \mathcal{H}_\Psi^B} \mathbb{E}_{\mathcal{D}_A}[\ell(h(z), y)] \geq 1 - \frac{\sqrt{2N}B}{2^k}.$$

We now prove the corollary.

**Corollary 30** (Restatement of Corollary 3). *For any function $f$ using a shift-invariant kernel $K$ with RKHS norm bounded by $L$, or $f(x) = \sum_i \alpha_i K(z_i, x)$ for some data points $z_i$ and $||\alpha||_2 \leq L$. If $3 < k \leq D/16$ and $k$ is odd, then there exists $\mathcal{D} \in \mathcal{F}_\Xi$ such that $f$ have hinge-loss at least $p_o(1 - \frac{\text{poly}(d,L)}{2^k}) - \frac{1}{\text{poly}(d,L)}$.*

*Proof.* By Claim 1 in Rahimi & Recht (2008), for any $\nu > 0$, there exists $N = \text{poly}(d, 1/\nu)$ Fourier features $\Psi_j$ that can approximate the shift-invariant kernel up to error $\nu$. For any $\epsilon > 0$, consider $\sum_i \alpha_i \langle \Psi(z_i), \Phi(x) \rangle = \langle \sum_i \alpha_i \Psi(z_i), \Psi(x) \rangle$. Let $w = \sum_i \alpha_i \Psi(z_i)$ and let $\nu = O(\frac{\epsilon}{L})$, then $\langle \Psi(x), w \rangle$ approximates $f(x)$ upto error $\epsilon$ and $N = \text{poly}(d, L, 1/\epsilon)$ and the norm of $w$ bounded by $B = \text{poly}(d, L, 1/\epsilon)$. The reasoning is the same for $f$ in the RKHS form, replacing sum with integral. By Theorem 2, $\langle \Psi(x), w \rangle$ has hinge-loss at least $p_0(1 - \frac{\sqrt{2N}B}{2^k})$. Thus, the function $f$ has loss at least $p_0(1 - \frac{\text{poly}(d,L,1/\epsilon)}{2^k}) - \epsilon$. Choose $\epsilon = \frac{1}{\text{poly}(d,L)}$, we get the bound. $\qquad\square$

# D   LOWER BOUND FOR LEARNING WITHOUT INPUT STRUCTURE

First recall the Statistical Query model (Kearns, 1998). In this model, the learning algorithm can only receive information about the data through statistical queries. A statistical query is specified by some property predicate $Q$ of labeled instances, and a tolerance parameter $\tau \in [0, 1]$. When the algorithm asks a statistical query $(Q, \tau)$, it receives a response $\hat{P}_Q \in [P_Q - \tau, P_Q + \tau]$, where $P_Q = \Pr[Q(x, y) \text{ is true}]$. $Q$ is also required to be polynomially computable, i.e., for any $(x, y)$ $Q(x, y)$ can be computed in polynomial time. Notice that a statistical query can be simulated by empirical average of a large random sample of data of size roughly $O(1/\tau^2)$ to assure the tolerance $\tau$ with high probability.

Blum et al. (1994) introduces the notion of Statistical Query dimension, which is convenient for our purpose.

**Definition 31** (Definition 2 in Blum et al. (1994)). *For concept class $C$ and distribution $\mathcal{D}$, the statistical query dimension SQ-DIM$(C, \mathcal{D})$ is the largest number $d$ such that $C$ contains $d$ concepts $c_1, \ldots, c_d$ that are nearly pairwise uncorrelated: specifically, for all $i \neq j$,*

$$| \Pr_{x \sim \mathcal{D}}[c_i(x) = c_j(x)] - \Pr_{x \sim \mathcal{D}}[c_i(x) \neq c_j(x)]| \leq 1/d^3. \tag{289}$$

**Theorem 32** (Theorem 12 in Blum et al. (1994)). *In order to learn $C$ to error less than $1/2 - 1/d^3$ in the Statistical Query model, where $d = SQ\text{-}DIM(C, \mathcal{D})$, either the number of queries or $1/\tau$ must be at least $\frac{1}{2}d^{1/3}$.*

We now use the above tools to prove our lower bound.

**Theorem 33** (Restatement of Theorem 4). *For any algorithm in the Statistical Query model that can learn over $\mathcal{F}_{\Xi_0}$ to error less than $\frac{1}{2} - \frac{1}{\binom{D}{k}^3}$, either the number of queries or $1/\tau$ must be at least $\frac{1}{2}\binom{D}{k}^{1/3}$.*

*Proof of Theorem 4.* Consider the following concept class and marginal distribution:

- Let $\mathcal{D}$ be the distribution over $\tilde{x}$, given by $\tilde{x} = M\tilde{\phi}$ and $\tilde{\phi}_j$ are i.i.d. with $\Pr[\tilde{\phi}_j = 0] = \Pr[\tilde{\phi}_j = 1] = 1/2$.

- Let $C$ be the class of functions $y = g_A(\tilde{\phi}) = \mathbb{I}[\sum_j (1 - \tilde{\phi}_j) \text{ is odd}]$ for different $A \subseteq [D]$.

The distributions over $(\tilde{x}, y)$ induced by $(C, \mathcal{D})$ are a subset of $\mathcal{F}_{\Xi_0}$. It is then sufficient to show that SQ-DIM$(C, \mathcal{D}) \geq \binom{D}{k}$.

It is easy to see that $C$ are essentially the sparse parity functions: if $z_j = 2\tilde{\phi}_j - 1$, then $g_A(\tilde{\phi}) = \prod_{j \in A} z_j$. This then implies that the $g_A$'s are uncorrelated, so SQ-DIM$(C, \mathcal{D}) \geq \binom{D}{k}$. $\qquad\square$

# E  COMPLETE EXPERIMENTAL RESULTS

Our experiments mainly focus on feature learning and the effect of the input structure. We first perform simulations on our learning problems to (1) verify our main theorems on the benefit of feature learning and the effect of input structure (2) verify our analysis of feature learning in networks. We then check if our insights carry over to real data: (3) whether similar feature learning is presented in real network/data; (4) whether damaging the input structure lowers the performance. The results are consistent with our analysis and provide positive support for the theory.

The experiments were ran 5 times with different random seeds, and the average results (accuracy) are reported. The standard deviations of the results are smaller than 0.5% and thus we do not present them for clarity. The hardware specifications are 4 Intel(R) Core(TM) i7-7700HQ CPU @ 2.80GHz, 16 GB RAM, and one NVIDIA GPU GTX1080.

## E.1  SIMULATION

We train a two-layer network following our learning process. We use two fixed feature methods: the NTK (Fang et al., 2021) and random feature (RF) methods based on the same network and random initialization as the network learning. More precisely, in the NTK method, we randomly initialize the network and take its NTK and learn a classifier on it. In the RF method, we freeze the first layer of the network, and train the second layer (on the random features given by the frozen neurons). The training step number is the same as that in network learning. We also test these three methods on the data distribution with input structure removed (i.e., $\mathcal{F}_{\Xi_0}$ in Theorem 4). For comparison, we take the representation of our two-layer network at step one/step two, named One Step/Two Step (fix the weight of the 1st layer after the first step/second step to train the weight of the second layer), and train the best classifiers on top of them.

Recall that our analysis is on the *directions* of the weights without considering their *scaling*, and thus it is important to choose cosine similarity rather than the typical $\ell_2$ distance. Thus, we use metric Cos Similarity $\max_{\{i \in [2m]\}} \cos(w_i, \sum_{j \in A} M_j)$ in our tables, and use Multidimensional Scaling to plot the weights distribution. The simulation dataset size is 50000. During training, the batch size is 1000, while for the first two steps we use the approximate full gradient (batch size is 50000). Each step is corresponding to one weights update.

### E.1.1  PARITY LABELING

**Setting.** We generate data according to the parity function data distributions used in our proof of the lower bound for fixed features (Theorem 2), with $d = 500, D = 100, k = 5, p_o = 1/2$, with a randomly sampled $A$. More precisely, we consider $\mathcal{D}$ defined as follows.

- Let $P = \{i \in [k] : i \text{ is odd}\}$. That is, if there are odd numbers of 1's in $\tilde{\phi}_A$, then $y = +1$.

- Let $\mathcal{D}_{\tilde{\phi}}^{(0)}$ be a distribution where all entries $\tilde{\phi}_j$ are i.i.d. with $\Pr[\tilde{\phi}_j = 0] = \Pr[\tilde{\phi}_j = 1] = 1/2$. Let $\mathcal{D}^{(0)}$ be the distribution over $(\tilde{x}, y)$ induced by $\mathcal{D}_{\tilde{\phi}}^{(0)}$ and the above $P$.

- Let $\mathcal{D}_{\tilde{\phi}}^{(1)}$ be a distribution where all entries $\tilde{\phi}_j$ for $j \notin A$ are i.i.d. with $\Pr[\tilde{\phi}_j = 1] = p_o/(2 - 2p_o)$, while $\Pr[\tilde{\phi}_A = (0, 0, \ldots, 0)] = \Pr[\tilde{\phi}_A = (1, 1, \ldots, 1)] = 1/2$. Let $\mathcal{D}^{(1)}$ be the distribution over $(\tilde{x}, y)$ induced by $\mathcal{D}_{\tilde{\phi}}^{(1)}$ and the above $P$.

- Let $\mathcal{D}_A^{\text{mix}} = p_o \mathcal{D}^{(0)} + (1 - p_o) \mathcal{D}^{(1)}$.

The network and the training follow Section 3, where the network size is $m = 300$ and the training time $T = 600$ steps.

**Verification of the Main Results.** Figure 5 shows that the results are consistent with our analysis. Network learning gets high test accuracy while the two fixed feature methods get significantly lower accuracy. Furthermore, when the input structure is removed, all three methods get test accuracy similar to random guessing.

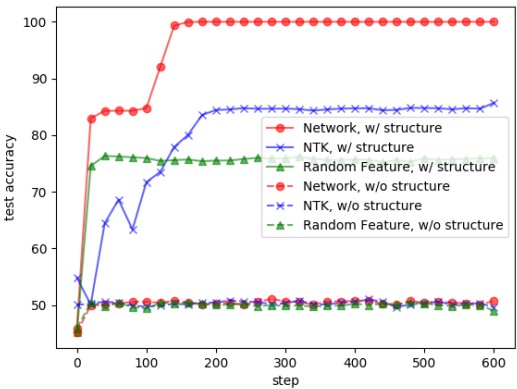

Figure 5: Test accuracy on simulated data under parity labeling with or without input structure.

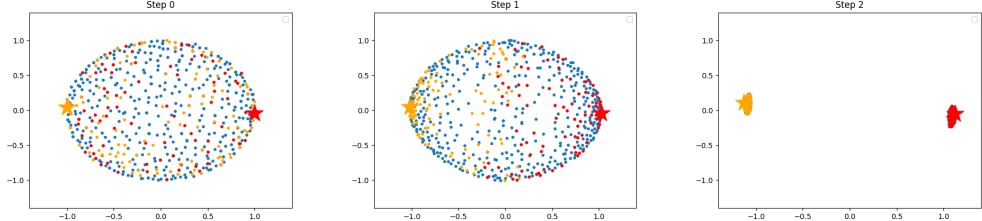

Figure 6: Visualization of the weights $w_i$'s after initialization/one gradient step/two gradient steps in network learning under parity labeling. The red star denotes the ground-truth $\sum_{j \in A} M_j$; the orange star is $-\sum_{j \in A} M_j$. The red dots are the weights closest to the red star after two steps; the orange ones are for the orange star.

| Model | Network | NTK | RF | One Step | Two Step | Network w/o structure |
|---|---|---|---|---|---|---|
| Train Acc (%) | 100.0 | 84.0 | 74.7 | 51.3 | 100.0 | 100.0 |
| Test Acc (%) | 100.0 | 86.4 | 76.0 | 52.2 | 100.0 | 52.0 |
| Cos Similarity | 0.997 | NA | 0.114 | 0.848 | 0.997 | 0.253 |

Table 1: Parity labeling results in six methods. The cosine similarity is computed between the ground-truth $\sum_{j \in A} M_j$ and the closest neuron weight.

**Feature Learning in Networks.** Figure 6 shows that the results are as predicted by our analysis. After the first gradient step, some weights begin to cluster around the ground-truth $\sum_{j \in A} M_j$ (or $-\sum_{j \in A} M_j$ due to we have $a_i$ in the gradient update which can be positive or negative). After the second step the weights get improved and well-aligned with the ground-truth (with cosine similarity $> 0.99$).

Table 1 shows the results for different methods. Recall that the Cos Similarity metric is $\max_{\{i \in [2m]\}} \cos(w_i, \sum_{j \in A} M_j)$, which reports the cosine value of the closest one. One Step refers to the method where we take the neurons after one gradient step, freeze their weights, and train a classifier on top; similar for Two Step. One Step gets test accuracy about $52\%$, while Two Step gets accuracy about $100\%$. This demonstrates that while some effective feature emerge in the first step, they need to be improved in the second step for accurate prediction. NTK, random feature, One Step all failed, while Network and Two Step can achieve 100% test accuracy. Network w/o structure refers to training the network on data without the input structure. It overfits the training dataset with 52% test accuracy.

### E.1.2 INTERVAL LABELING

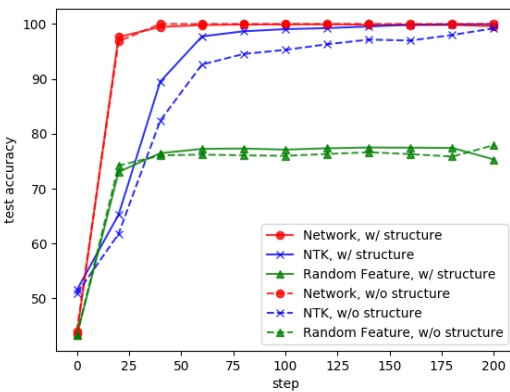

Figure 7: Test accuracy on simulated data under interval labeling with or without input structure.

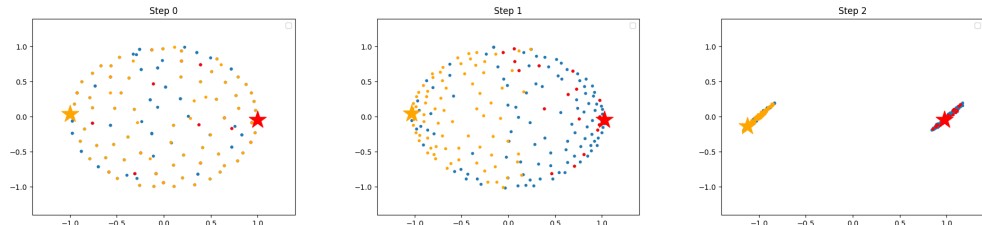

Figure 8: Visualization of the weights $w_i$'s after initialization/one gradient step/two gradient steps in network learning under interval labeling. The red star denotes the ground-truth $\sum_{j \in A} M_j$; the orange star is $-\sum_{j \in A} M_j$. The red dots are the weights closest to the red star after two steps; the orange ones are for the orange star.

| Model | Network | NTK | RF | One Step | Two Step | Network w/o structure |
|---|---|---|---|---|---|---|
| Train Acc (%) | 100.0 | 100.0 | 76.4 | 44.1 | 100.0 | 100.0 |
| Test Acc (%) | 100.0 | 100.0 | 73.2 | 41.0 | 100.0 | 100.0 |
| Cos Similarity | 1.00 | NA | 0.153 | 0.901 | 0.994 | 0.965 |

Table 2: Interval labeling results in six methods.

**Setting.** We also tried interval function, where $y = 1$ if $\sum_{i \in A} \tilde{\phi}_i$ is in the range $[t_1, t_2]$ with $t_1 = 20$ and $t_2 = 30$, otherwise $y = -1$. We use $d = 500, D = 100, k = 30$. The $\tilde{\phi}_i$'s are independent, and $\Pr[\tilde{\phi}_i = 1] = 2/3$ for any $i \in A$, and $\Pr[\tilde{\phi}_i = 1] = 1/2$ otherwise. When the input structure is removed, we set $\Pr[\tilde{\phi}_i = 1] = 1/2$ for all $i$'s.

The network and training again follows Section 3 with a network size $m = 100$ and the training time $T = 200$ steps.

**Verification of the Main Results.** Figure 7 shows that network learning learns the fastest, NTK learns slower but reaches similar test accuracy, while random feature can only reach a decent but lower accuracy. This is because for such simpler labeling functions, fixed feature methods can still achieve good performance (note that the lower bound does not hold for such a case), while the performance depends on what fixed features to use.

Furthermore, when the input structure is removed, the methods still get similar (or only slightly worse) performance as with input structure. This shows that when the labeling function is simple, the

help of the input structure for learning may not be needed. In the experiments on real data, we will show that when the input structure is changed, it indeed leads to lower performance which suggests that the labeling function in practice is typically more complicated than this interval labeling setting, and the help of the input structure is significant for learning.

**Feature Learning in Networks.** Figure 8 shows the phenomenon of feature learning similar to that in the parity labeling setting. Table 2 shows the test accuracy of six different methods. Random feature and One Step failed, while Network, NTK and Two Step succeed showing that interval labeling setting is a simpler case than parity labeling setting.

### E.2    MORE SIMULATION RESULT IN VARIOUS SETTINGS

We show the robustness of our simulation results by studying the learning behaviors in a variety of settings including different sample size, input data dimension and class imbalance. We reuse the same setting as the simulation in the main text (details in E.1.1), vary different parameters, and report the accuracy, the cosine similarities between the learned weights, and the visualization of the neuron weights.

### E.2.1    VARYING INPUT DATA DIMENSION

In the simulation experiments in the main text, the input data dimension $d$ is 500. Here we change the input data dimension to 100 and 2000. All other configurations follow E.1.1.

**Verification of the Main Results.** Figure 9 shows that our claim is robust under different input data dimensions. The performance of network learning is superior over NTK and random feature approaches on inputs with structure, and on inputs without structure, all three methods fail.

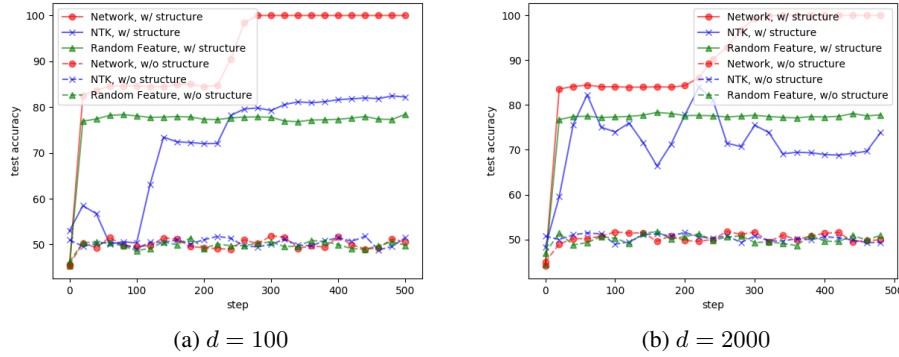

(a) $d = 100$             (b) $d = 2000$

Figure 9: Test accuracy on simulated data under different input data dimensions.

| $d = 100$ | Network | NTK | RF | One Step | Two Step | Network w/o structure |
|---|---|---|---|---|---|---|
| Train Acc | 100.0 | 83.1 | 78.9 | 53.0 | 100.0 | 100.0 |
| Test Acc | 100.0 | 81.5 | 78.3 | 51.1 | 100.0 | 51.0 |
| Cos Similarity | 1.000 | NA | 0.354 | 0.967 | 1.000 | 0.331 |

| $d = 2000$ | Network | NTK | RF | One Step | Two Step | Network w/o structure |
|---|---|---|---|---|---|---|
| Train Acc | 100.0 | 75.6 | 80.0 | 50.22 | 100.0 | 100.0 |
| Test Acc | 100.0 | 75.4 | 77.0 | 50.01 | 100.0 | 52.5 |
| Cos Similarity | 0.998 | NA | 0.056 | 0.560 | 0.998 | 0.309 |

Table 3: Results of six methods for different input data dimensions. The cosine similarity is computed between the ground-truth $\sum_{j \in A} M_j$ and the closest neuron weight.

**Feature Learning in Networks.** Figure 10 visualizes the neuron weights. It shows similar results to that in E.1.1: the weights gets updated to to the effective feature in the first two steps, forming clusters.

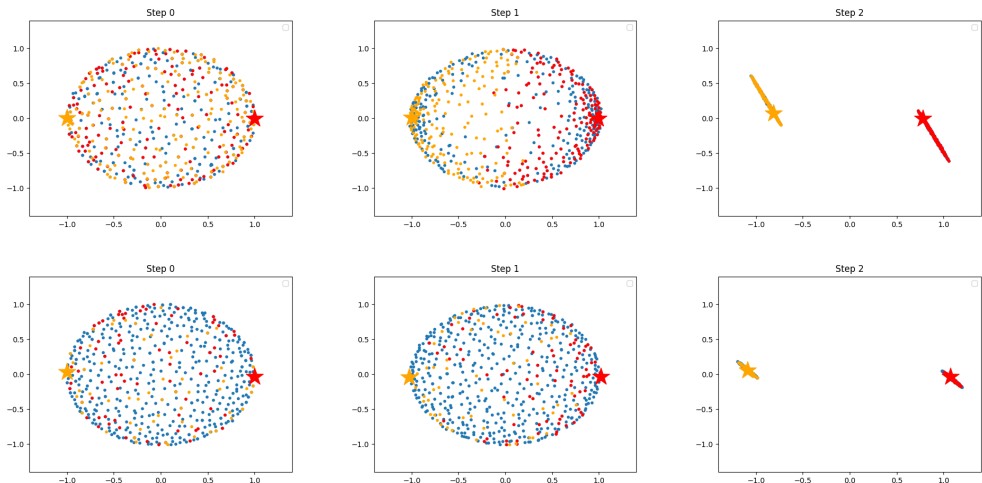

Figure 10: Visualization of the weights $w_i$'s in early steps under different input data dimensions. Upper row: input data dimension $d = 100$; lower row: $d = 2000$.

Table 3 shows some quantitative results. In particular, the average cosine similarities between neuron weights and the effective features after two steps are close to 1, showing that they match the effective features.

### E.2.2 Varying Class Imbalance Ratio

The experiments in the main text has 25000 training samples for each class. Here we keep the total sample size 50000 but use different class imbalance ratios, which is the class $-1$ sample size divide by the total sample size.

**Verification of the Main Results.** Figure 11 shows that our claim is robust under different class imbalance ratios. The results are similar to those for balanced classes, except that NTK becomes less stable.

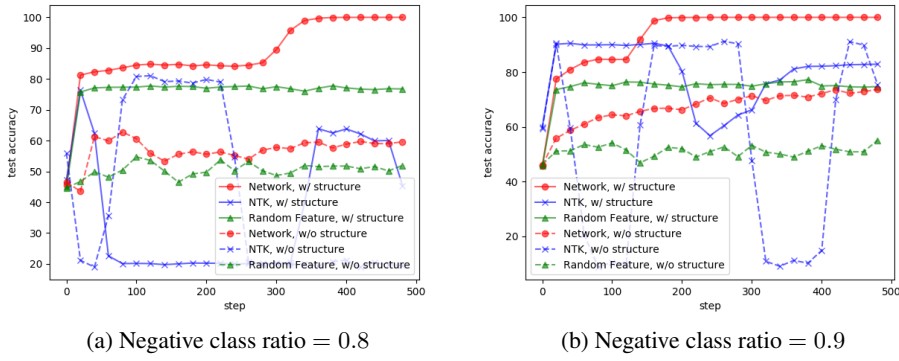

(a) Negative class ratio $= 0.8$        (b) Negative class ratio $= 0.9$

Figure 11: Test accuracy on simulated data under different negative class ratios.

**Feature Learning in Networks.** Figure 12 visualizes the neurons' weights. Again, the observation is similar to that for balanced classes. Table 4 shows some quantitative results which are also similar to those for balanced classes.

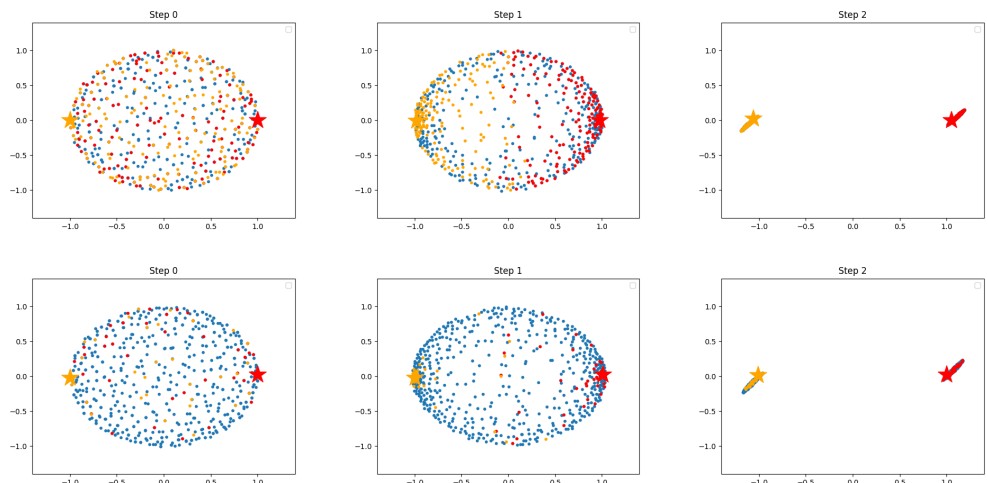

Figure 12: Visualization of the weights $w_i$'s in early steps under different class imbalance ratios. Upper row: negative class ratio 0.8; lower row: 0.9.

| ratio = 0.8 | Network | NTK | RF | One Step | Two Step | Network w/o structure |
|---|---|---|---|---|---|---|
| Train Acc | 100.0 | 62.9 | 72.7 | 78.3 | 100.0 | 100.0 |
| Test Acc | 100.0 | 82.7 | 70.4 | 75.7 | 100.0 | 61.7 |
| Cos Similarity | 0.999 | NA | 0.293 | 0.950 | 0.999 | 0.218 |
| ratio = 0.9 | Network | NTK | RF | One Step | Two Step | Network w/o structure |
| Train Acc | 100.0 | 84.0 | 73.6 | 92.3 | 100.0 | 100.0 |
| Test Acc | 100.0 | 81.7 | 72.4 | 89.2 | 100.0 | 71.8 |
| Cos Similarity | 0.997 | NA | 0.296 | 0.956 | 0.997 | 0.286 |

Table 4: Results of six methods under different negative class ratios.

### E.2.3 VARYING SAMPLE SIZE

Here we change the sample size 50000 in Section E.1.1 to be 25000 and 10000. For sample size 25000, we observe similar results. For sample size 10000, we observe over-fitting (test accuracy much lower than train accuracy). Therefore, for sample size 10000 we reduces the size of the network (i.e., number of hidden neurons) from $m = 300$ to $m = 50$.

**Verification of the Main Results.** Figure 13 shows that our claim is robust under different sample sizes. In particular, the network learning still outperforms the NTK and random feature approaches on structured inputs.

| $n = 25000$ | Network | NTK | RF | One Step | Two Step | Network w/o structure |
|---|---|---|---|---|---|---|
| Train Acc | 100.0 | 84.0 | 78.6 | 50.6 | 100.0 | 100 |
| Test Acc | 100.0 | 84.1 | 74.7 | 50.0 | 100.0 | 50.2 |
| Cos Similarity | 0.997 | NA | 0.105 | 0.851 | 0.997 | 0.230 |
| $n = 10000$ | Network | NTK | RF | One Step | Two Step | Network w/o structure |
| Train Acc | 100.0 | 73.9 | 71.6 | 50.7 | 100.0 | 100.0 |
| Test Acc | 100.0 | 75.0 | 74.3 | 50.3 | 100.0 | 52.2 |
| Cos Similarity | 0.995 | NA | 0.096 | 0.974 | 0.994 | 0.176 |

Table 5: Results of six methods for different sample size.

**Feature Learning in Networks.** Figure 14 and Table 5 show that the phenomenon of feature learning for different samples is similar to that in E.1.1.

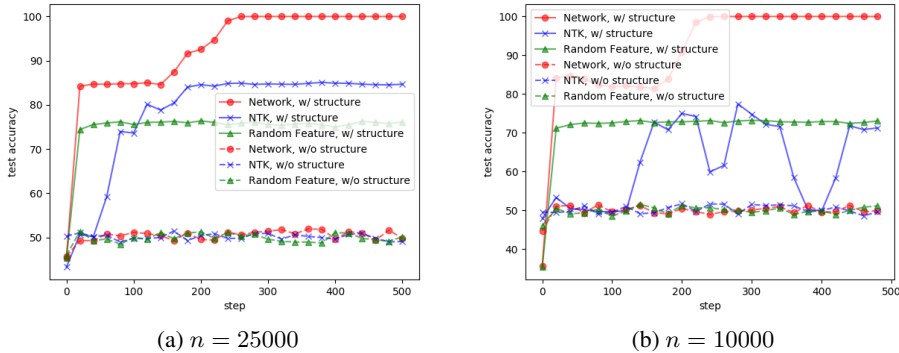

(a) $n = 25000$          (b) $n = 10000$

Figure 13: Test accuracy on simulated data under different sample sizes $n$.

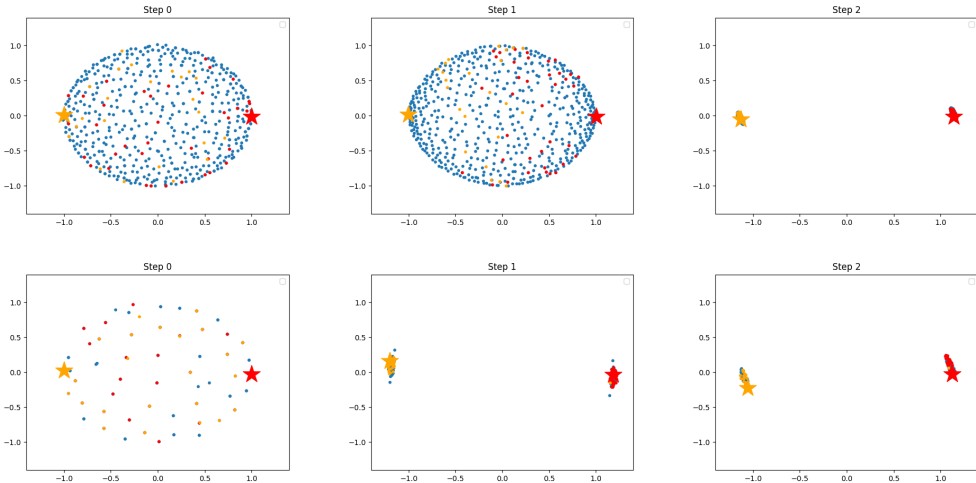

Figure 14: Visualization of the weights $w_i$'s in early steps under different sample sizes. Upper row: sample size 25000; lower row: 10000.

### E.3   EXPERIMENTS ON MORE DATA GENERATION MODELS

In this section we consider some additional data distributions and run the simulation experiments, in particular, focusing on the feature learning phenomenon. Note that our analysis is for the setting where the input distributions have structure revealing some information about the labeling function. (More precisely, the labeling function is specified by $A$ and $P$, while the input distribution also depends on them.) We therefore consider two other data generation mechanisms where the labeling function also has connections to the input distributions.

#### E.3.1   HIDDEN REPRESENTATION LABELING

Here we consider the following data model: first uniformly at random select $\tilde{\phi}_A$ from a set of binary vectors, and assign label 1 to some and -1 to others; sample irrelevant patterns $\tilde{\phi}_{-A}$ uniformly at random; generate the input $x = M\tilde{\phi}$. We randomly select 50 binary vectors for each label, with $d = 500, D = 250, k = 50, p_o = 1/2$.

This is a generalization of the distribution $\mathcal{D}^{(1)}$, a component in the distribution of our simulation experiments (see the proof of Theorem 2 for details). Recall the definition of $\mathcal{D}^{(1)}$: $\tilde{\phi}_A$ is uniform on only two values $[+1, \ldots, +1]$ and $[0, \ldots, 0]$, and uniform over irrelevant patterns; the value

$[+1, \dots, +1]$ corresponds to one class and $[0, \dots, 0]$ correspond to another class. Our data model here generalizes $\mathcal{D}^{(1)}$ to more than 2 values.

The visualization is shown in Figure 15. We can observe similar feature learning phenomena, and the neuron weights are updated to form clusters.

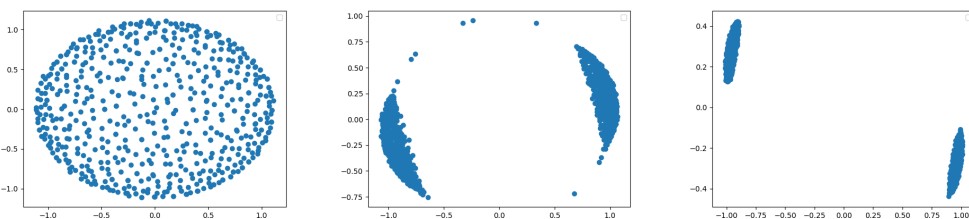

Figure 15: Visualization of the weights $w_i$'s after initialization/one gradient step/two gradient steps in network learning under hidden representation labeling.

### E.3.2 TWO-LAYER NETWORKS ON MIXTURE OF GAUSSIANS

To further support our intuition of feature learning, we run experiments on mixture of Gaussians.

**Data.** Let $\mathcal{X} = \mathbb{R}^d$ be the input space, and $\mathcal{Y} = \{\pm 1\}$ be the label space. Suppose $M \in \mathbb{R}^{d \times k}$ is an dictionary with $k$ orthonormal columns. Let $\varepsilon_i, i = 1, \dots, k$ be i.i.d symmetric Bernoulli random variables, and $g \sim \mathcal{N}(0, \sigma_r^2 \frac{k}{d} \mathbb{I}_d)$. Then we generate the input $x$ and class label $y$ by:

$$x = \sum_{i=1}^{k} \varepsilon_i M_{:i} + g, \quad y = \prod_{i=1}^{k} \varepsilon_i \tag{290}$$

In this case, $2^k$ Gaussian clusters will be created. The centers of the Gaussian clusters $\sum_{i=1}^{k} \pm M_{:i}$ lie on the vertices of a hyper cube, and the label of each Gaussian cluster is determined by the parity function on the vertices of the hyper cube.

Note that the labeling function is roughly equivalent to a network: $y = \sum_{i=1}^{n} a_i \text{ReLU}(\langle c_i, x \rangle)$ where $c_i$'s are the Gaussian centers, and $a_i \propto 1$ for Gaussian components with label 1 and $a_i \propto -1$ for those with label -1.

**Setting.** We then train a two-layer network with $m = 800$ hidden neurons on data sets generated as above with different chosen $k$'s and $d$'s. The training follows typical practice (not the hyperparameters in our analysis). In this setting, we expect the neural network to learn the effective features: the directions of Gaussian cluster centers.

**Result.** We run experiments with different settings. The parameters are shown in Table 6. From Figure 16 we can see that some neurons learn the directions of Gaussian centers, and each Gaussian center is covered by some neurons, which matches our expectation.

| Parameters | $d$ | $k$ | Number of Clusters | $\sigma_r$ |
|---|---|---|---|---|
| Experiment 1 | 100 | 4 | 16 | 1 |
| Experiment 2 | 25 | 4 | 16 | 0.7 |
| Experiment 3 | 100 | 5 | 32 | 1 |

Table 6: Gaussian mixture setting.

### E.4 REAL DATA: FEATURE LEARNING IN NETWORKS

We take the subset of MNIST (Deng, 2012) with labels 0/1, CIFAR10 (Krizhevsky, 2012) with labels airplane/automobile and SVHN (Netzer et al., 2011) with labels 0/1, and train a two-layer network

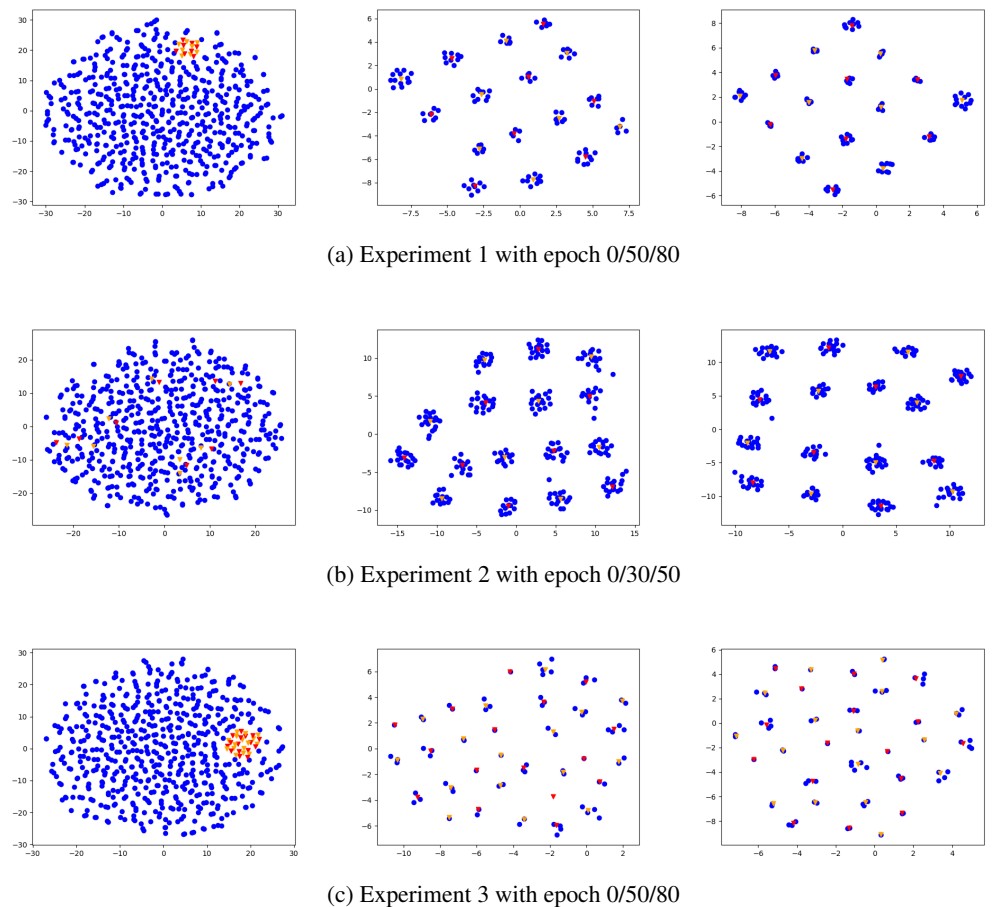

(a) Experiment 1 with epoch 0/50/80

(b) Experiment 2 with epoch 0/30/50

(c) Experiment 3 with epoch 0/50/80

Figure 16: Visualization of the weights $w_i$'s (blue dots) and Gaussian centers (red for positive labeled clusters and orange for negative labeled clusters).

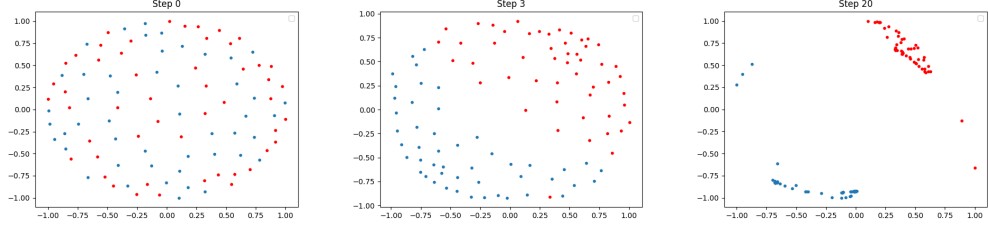

Figure 17: Visualization of the neurons' weights in a two-layer network trained on the subset of MNIST data with label 0/1. The weights gradually form two clusters.

with $m = 50$. We use traditional weight initialization method (random Gaussian) and training method (SGD with momentum $= 0.95$ without regularization) in this section, for our purpose of investigating the training dynamics in practice.

Then we visualize the neurons' weights following the same method in the simulation. Figure 17, Figure 18 and Figure 19 show a similar feature learning phenomenon: effective features emerge after a few steps, and then get improved to form clusters. This shows the insights obtained on our learning problems are also applicable to the real data.

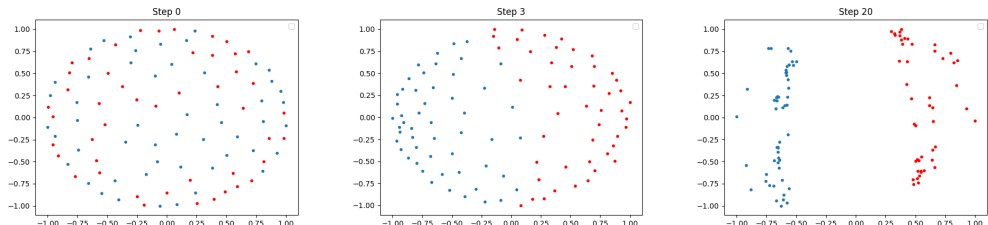

Figure 18: Visualization of the neurons' weights in a two-layer network trained on the subset of CIFAR10 data with label airplane/automobile. The weights gradually form two clusters.

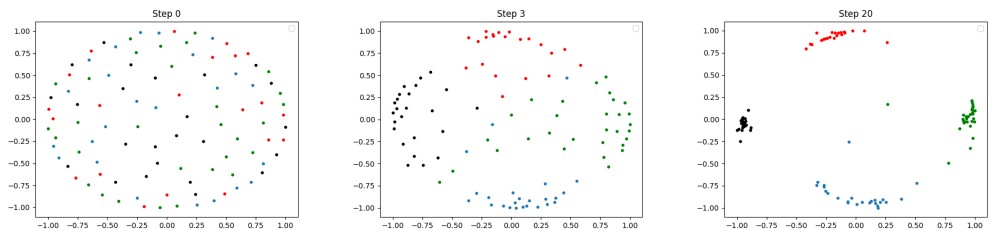

Figure 19: Visualization of the neurons' weights in a two-layer network trained on the subset of SVHN data with label 0/1. The weights gradually form four clusters.

| | $\cos(v_1, \bar{v})$ | $\cos(v_2, \bar{v})$ | $\cos(v_3, \bar{v})$ | $\cos(v_1, v_2)$ | $\cos(v_1, v_3)$ | $\cos(v_2, v_3)$ |
|---|---|---|---|---|---|---|
| ResNet(128) | 0.9727 | 0.8655 | 0.6549 | 0.7454 | 0.5083 | 0.6533 |
| ResNet(256) | 0.8646 | 0.9665 | 0.9121 | 0.7087 | 0.6919 | 0.9135 |

Table 7: Cosine similarities between the gradients in the early steps. We choose the neuron weight closest to the average weight of the green cluster at the end of the training (in Figure 20 for $\mathrm{ResNet}(128)$ and Figure 21 for $\mathrm{ResNet}(256)$). We record the gradients of the first 30 steps and divide them to three trunks of 10 steps evenly and sequentially. For the three trunks, we get the average gradients $v_1, v_2, v_3$. We calculate their cosine similarities to their average $\bar{v} = (v_1 + v_2 + v_3)/3$ and those between them.

### E.4.1 CNNs on Binary Cifar10: Feature Learning in Networks

**Setting.** We use $\mathrm{ResNet}(m)$, which is a ResNet-18 convolutional neural network (He et al., 2016) with $m$ filters in the first residual block. It is obtained by scaling the number of filters in each block proportionally from the standard ResNet-18 network which is $\mathrm{ResNet}(64)$. We use $\mathrm{ResNet}(128)$ and $\mathrm{ResNet}(256)$ in this experiment. We train our model on Binary CIFAR10 (Krizhevsky, 2012) with labels airplane/automobile for 20 epochs. The final test accuracy of $\mathrm{ResNet}(128)$ is 95.75% and that of $\mathrm{ResNet}(256)$ is 93.8%.

**Results.** Figure 20 visualizes the filters' weights of different residual blocks in $\mathrm{ResNet}(128)$ at Epoch 0, 3, and 20, and Figure 21 shows those in $\mathrm{ResNet}(256)$. They show that feature learning happens in the early stage, and show that there are some clusters of weights (e.g., the red and green points). These colored points are selected at Epoch 20. We first visualize the weights at Epoch 20, and then hand pick the points that roughly form two clusters (i.e., the points in the same cluster are close to each other while those in different clusters are far away). We assign red and green colors to the two clusters at Epoch 20, and then assign these weights with the same color in Epoch 0 and 3. Finally, we compute the cosine similarities and show that the hand picked points are indeed roughly clusters in the high-dimension.

In particular, we have the following three observations.

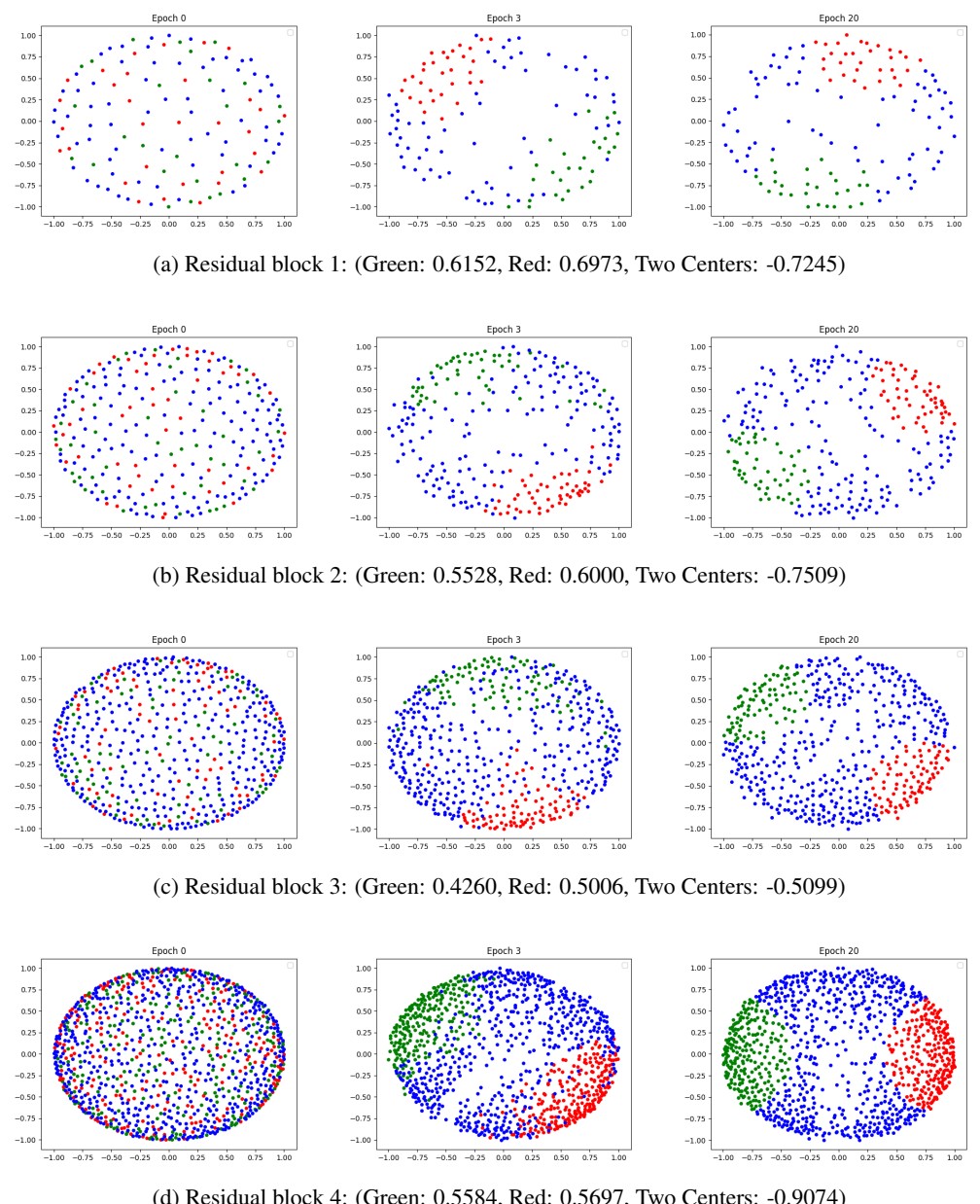

(a) Residual block 1: (Green: 0.6152, Red: 0.6973, Two Centers: -0.7245)

(b) Residual block 2: (Green: 0.5528, Red: 0.6000, Two Centers: -0.7509)

(c) Residual block 3: (Green: 0.4260, Red: 0.5006, Two Centers: -0.5099)

(d) Residual block 4: (Green: 0.5584, Red: 0.5697, Two Centers: -0.9074)

Figure 20: Visualization of the normalized convolution weights in all Residual block of $\mathrm{ResNet}(128)$ trained on the subset of CIFAR10 data with labels airplane/automobile. We show the weights after 0/3/20 epochs in network learning. The weights gradually form two clusters in all Residual blocks. We also report average cosine similarity between the green/red points in the clusters to their centers and cosine similarity between two cluster centers as (Green, Red, Two Centers).

First, we can see that the filter weights change significantly during the early stage of the training, indicating feature learning happens in the early stage: the change between Epoch 0 and Epoch 3 is much more significant than that between Epoch 3 and Epoch 20.

Second, we can also verify that the feature learning is guided by the gradients: the gradients of a filter in the early gradient steps point to similar directions (and thus the updated filter will learn this direction). More precisely, for a selected filter, we average the gradients every 10 gradient steps (so to reduce the variance due to mini-batch), and get $v_1, v_2$ and $v_3$ for the first 30 steps and compute their cosine similarities and those to their average. Table 7 shows the results. In general the similarities

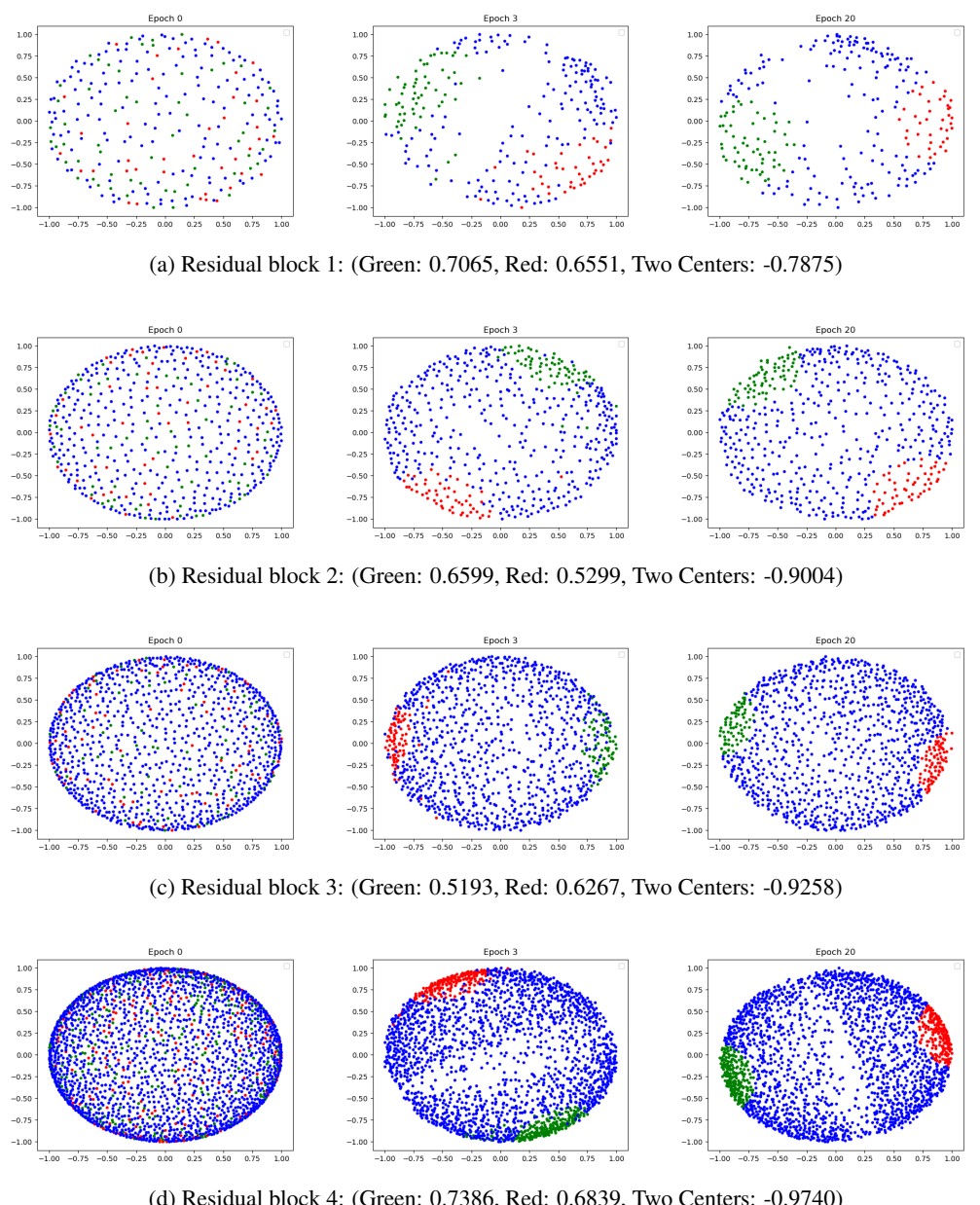

(a) Residual block 1: (Green: 0.7065, Red: 0.6551, Two Centers: -0.7875)

(b) Residual block 2: (Green: 0.6599, Red: 0.5299, Two Centers: -0.9004)

(c) Residual block 3: (Green: 0.5193, Red: 0.6267, Two Centers: -0.9258)

(d) Residual block 4: (Green: 0.7386, Red: 0.6839, Two Centers: -0.9740)

Figure 21: Visualization of the normalized convolution weights in all Residual block of $\mathrm{ResNet}(256)$ trained on the subset of CIFAR10 data with labels airplane/automobile. We show the weights after 0/3/20 epochs in network learning. The weights gradually form two clusters in all Residual blocks. We also report average cosine similarity between the green/red points in the clusters to their centers and cosine similarity between two cluster centers as (Green, Red, Two Centers).

are high indicating they point to similar directions. (Note that a similarity of $0.6$ is regarded as very significant as the filters are in a high dimension of $3 \times 3 \times 1024 = 9216$).

Third, we also observe some clustering effect of the filter weights, though not as significant as in our simulations. For example, in the red and green clusters in Figure 20(a) for the first residual block, the average cosine similarity for filter weights in the red cluster is about 0.62 and that for the green is about 0.7, while the cosine similarity between the two clusters' centers is about -0.72. This shows significant similarities within the cluster while difference between clusters.

Note that the clustering is less significant than our simulation experiments. This is because practical data have more patterns (i.e., effective feature directions) to be learned than our synthetic data, and also the practical network is not as overparameterized as in our simulation. Then filters are likely to learn different patterns (or their mixtures) without forming significant clusters. The results of $\mathrm{ResNet}(256)$ show more significant clustering than $\mathrm{ResNet}(128)$, which supports our explanation. On the other hand, we emphasize that the key insight of our analysis is that the gradient guides the learning of effective features in the early stage of training (rather than the clustering), which is verified as discussed above.

### E.5  REAL DATA: THE EFFECT OF INPUT STRUCTURE

To study the influence of the input structure, we propose to keep the labeling function unchanged, vary the input distributions, and exam the change of the loss surface and the training dynamics. We first describe the detailed experimental methodology, which allows us to generate data with similar labeling function but different input distributions. Then we perform experiments on the generated datasets to investigate the change of the learning due to the change in the input distributions, and present the experimental results. Finally, we also perform experiments to verify the intuition behind our experimental method.

#### E.5.1  EXPERIMENTAL METHODOLOGY

We consider the following experimental method. Given an original dataset $\mathcal{L} = \{(x_i, y_i)\}_{i=1}^{n}$ (e.g., CIFAR10) and an unlabeled dataset $\mathcal{U} = \{\tilde{x}_i\}_{i=1}^{m}$ from a proposed distribution $P_{\mathcal{U}}$ (e.g., Gaussians), first extend the labeling function of $\mathcal{L}$ to $\mathcal{U}$, giving synthetic labels $\tilde{y}_i$ to $\tilde{x}_i$. Then train a neural network on the union of $\mathcal{L}$ and the synthetic data $\mathcal{L}_{\mathcal{U}} = \{(\tilde{x}_i, \tilde{y}_i)\}_{i=1}^{m}$. By investigating the new training dynamics, in particular the difference on the original part $\mathcal{L}$ and the synthetic part $\mathcal{L}_{\mathcal{U}}$, we can see the effect of the input structure. The original dataset should be from real-world data, since one of our goals is to compare them with synthetic data, and identify the properties of real-world data important for the success of learning.

A natural idea is to first learn a powerful network $f(x)$ (called the teacher) on $\mathcal{L}$ to approximate the true labeling function, then apply $f$ on $\mathcal{U}$ to generate synthetic labels, and finally train another network (called the student) on the synthetic data and original data. However, we found that naïvely implementation of this idea fails miserably: the support of $\mathcal{L}$ and $\mathcal{U}$ can be typically different, and the powerful network learned over $\mathcal{L}$ can have entirely different behavior on $\mathcal{U}$. Therefore, we need to control the size of the teacher $f$ so that the labeling on $\mathcal{U}$ has similar complexity as that on $\mathcal{L}$. For our purpose, we can define the complexity of the labeling on $\mathcal{L}$ as the minimum size of the teacher achieving an approximation error $\epsilon$ for a chosen $\epsilon$, if the ground-truth data distribution of $\mathcal{L}$ is known. However, given only limited data, we cannot faithfully estimate the needed size of the teacher, and need to take into account the variance introduced by the finite data.

Our key idea is to use the U-shaped curve of the bias-variance trade-off and select the size of the teacher at the minimum of the U-shaped curve. Since recent works (Belkin et al., 2019; Nakkiran et al., 2020) show that neural networks can have a double descent curve for the error v.s. model complexity, we thus plot the double descent curve, and find the minimum in the classical regime (corresponding to the traditional U-shape curve).

Our method is designed based on the following two reasons. First, on the U-shaped curve, the complexity of the network is still roughly controlled by that of the number of parameters. The local minimum of the U-shaped curve is a good measurement of the complexity of the data. If the ground-truth is much more complicated than the teacher, then increasing the teacher's size leads to a significant decrease in the approximation error (bias) compared to a small increase in the variance, that is, we will be on the left-hand side of the U-shaped. In contrast, on the right-hand side of the U-shaped, increasing the teacher's size leads to a small decrease in the bias compared to a significant increase in the variance. That is, the complexity of the ground-truth is comparable to or lower than the teacher. So the local minimum approximates the complexity of the ground-truth labeling function.

Second, the local minimum point is chosen to get the best approximation of the true labels. This helps to maintain the labeling from the real-world data and thus helps our investigation on the input, since too drastic change in the labeling can affect the training.

We note that the method is not perfect. First, the teacher at the local minimum of U-shape may not have very high accuracy, especially on more complicated data. To alleviate this, we also use the teacher to give synthetic labels $y_i'$ to $x_i$ in $\mathcal{L}$, and train the student network on $\mathcal{L}' = \{(x_i, y_i')\}_{i=1}^n$. Though this introduces some differences from the original labels, it is acceptable for our purpose of studying the inputs. Furthermore, ensuring the consistency of the labels on the original input in $\mathcal{L}$ and $\mathcal{U}$ is important in our experiments. Second, the measurement is an approximation due to variance. Since only limited labeled data is available, it's important and necessary to calibrate the measurement w.r.t. the level of variance on the given dataset.

**Method Description.** Algorithm 1 presents the details. For a fixed network architecture for the teacher $f$, it first varies the network size and plots the double descent curve. Then it selects the local minimum in the classic regime of U-shape and trains the teacher with the corresponding size. In practice, we observed that the teacher might have unbalanced probabilities for different classes on $\mathcal{U}$ if its training does not take into account $\mathcal{U}$. Therefore, we propose the following heuristic regularization using $x \in \mathcal{U}$, where $\lambda$ is a regularization weight, and $f(x)$ is the probabilities over classes given by the teacher:

$$R(x) = R_1(x) + \lambda R_2(x) \tag{291}$$

$$R_1(x) = \sum_j \left( \frac{\sum_i f(x)_j}{m} \ln \frac{\sum_i f(x)_j}{m} \right) \tag{292}$$

$$R_2(x) = -\frac{1}{m} \sum_i \sum_j (f(x)_j \ln(f(x)_j)). \tag{293}$$

Here, $R_1(x)$ guarantees that each kind of label has the same average probability to be generated, and $R_2(x)$ pushes the probability away from uniform to avoid the case that the class probabilities for each data point converge to uniform.

---

**Algorithm 1** Learning the teacher network to generate synthetic labels for studying the effect of the input structure

---

**Input:** teacher architecture $f$, labeled dataset $\mathcal{L} = \{(x_i, y_i)\}_{i=1}^n$, unlabeled dataset $\mathcal{U} = \{\tilde{x}_i\}_{i=1}^m$.
Let $i$ to be the size of $f$, $f_i$ to be the teacher of size $i$.
**for** $i = 1$ to $n$ **do**
    Train $f_i$ on $\mathcal{L}$ and let $l_i$ denote the test loss
**end for**
Plot $l_i$ v.s. $i$, identify the classical regime, and the size $i_t$ corresponding to the local minimum in classical regime.
Train $f_{i_t}$ on $\mathcal{L}$ with a regularizer $R(x)$ on $\mathcal{U}$ defined in (291).
**Output:** $f_{i_t}$

---

### E.5.2 EXPERIMENTAL RESULTS

**Network models.** Here we use one-hidden-layer fully-connected networks with $m$ hidden units and quadratic activation functions. The network is denoted as $\mathrm{FC}(m)$. We use $\mathrm{ResNet}(m)$, which is a ResNet-18 convolutional neural network (He et al., 2016) with $m$ filters in the first residual block. It is obtained by scaling the number of filters in each block proportionally from the standard ResNet-18 network which is $\mathrm{ResNet}(64)$.

**Datasets.** We use MNIST (Deng, 2012), CIFAR10 (Krizhevsky, 2012) and SVHN (Netzer et al., 2011) as $\mathcal{L}$, and use Gaussian and images in Tiny ImageNet (Le & Yang, 2015) as $\mathcal{U}$. We generate the mixture data, where the fraction of the unlabeled data is denoted as $\alpha$.

**Setup.** We first use Algorithm 1 on the labeled data $\mathcal{L}$ and the unlabeled data $\mathcal{U}$ to get a synthetic labeling function (the teacher network) and then use it to give synthetic labels on a mixture of inputs from $\mathcal{L}$ and $\mathcal{U}$. For MNIST, the teacher network learned is $\mathrm{FC}(9)$, where the number of the hidden units is determined by Algorithm 1. See empirical verification in Figure 26. For CIFAR10 and SVHN, the teacher networks are $\mathrm{ResNet}(5)$ and $\mathrm{ResNet}(2)$, respectively, as determined by our method. The student network for MNIST is $\mathrm{FC}(9)$, and those for CIFAR10 and SVHN are $\mathrm{ResNet}(9)$ and $\mathrm{ResNet}(8)$, respectively. Finally, we train the student networks on these new datasets with perturbed input distributions.

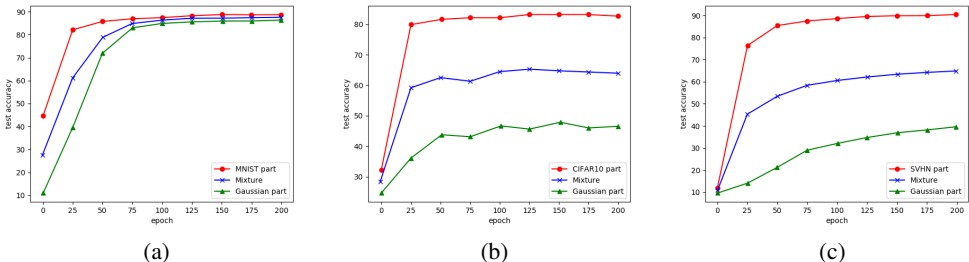

Figure 22: Test accuracy at different steps for an equal mixture $\alpha = 0.5$ of Gaussian inputs with data: (a) MNIST, (b) CIFAR10, (c) SVHN.

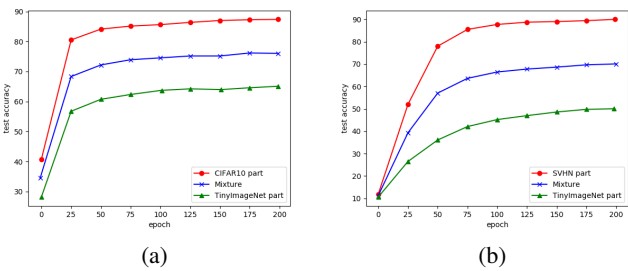

Figure 23: Test accuracy at different steps for an equal mixture $\alpha = 0.5$ of Tiny ImageNet inputs with data: (a) CIFAR10, (b) SVHN.

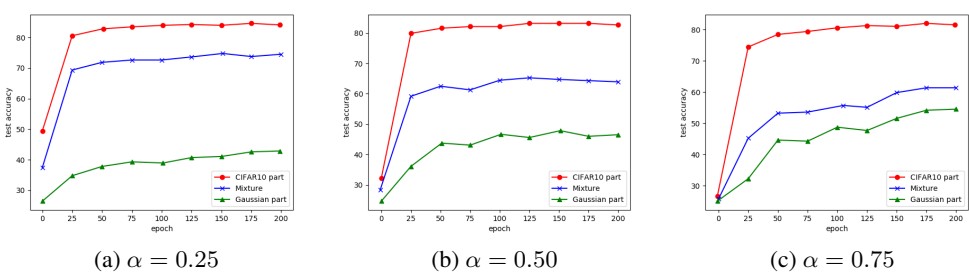

Figure 24: Test accuracy at different steps for varying mixture $\alpha$ of Gaussian inputs with CIFAR10.

Figure 22 shows the results on an equal mixture of data and Gaussian. It presents the test accuracy of the student on the original data part, the Gaussian part, and the whole mixture. For example, for CIFAR10, the test accuracy on the whole mixture is lower than that of training on the original CIFAR10, showing that the input structure indeed has a significant impact on the learning. Furthermore, the network learns well over the CIFAR10 part (with accuracy similar to that on the original data) but learns slower with worse accuracy on the Gaussian part. This suggests that the CIFAR10 input structure is still helping the network to learn effective features. While the results on MNIST+Gaussian do not show a significant trend (possibly because the tasks there are simpler), the results on SVHN+Gaussian show similar significant trends as CIFAR10+Gaussian.

Figure 24 shows the results when we vary the fraction of the Gaussian data $\alpha$. We observe that the test accuracy curve on the original part and that on the synthetic part have roughly the same trend for different $\alpha$ as before, further verifying our insights.

Figure 23 shows the results when mixed with Tiny ImageNet data instead of Gaussians. It shows a similar trend, while the performance on the Tiny ImageNet part is higher than that on the Gaussian part. This suggests that compared to Gaussians, the Tiny ImageNet data has helpful input structures, though not as helpful as that on the original data for learning the particular labeling.

### E.5.3 Larger Network on MNIST for Checking The Effect of Input Structure

Here we perform the experiment on MNIST as in E.5.2, but for a network with $m = 50$ hidden neurons rather than $m = 9$. Figure 25 shows similar results as those for $m = 9$: the learning on the MNIST input part is faster and better than that on the Gaussian input part. The separation between the two is actually more significant than that for $m = 9$. This then also supports our insight about the effect of input structures.

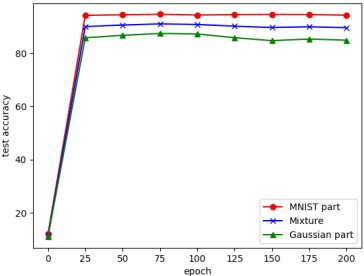

Figure 25: Test accuracy at different steps for an equal mixture $\alpha = 0.5$ of Gaussian inputs with MNIST, where $m = 50$.

### E.5.4 Empirical Verification of Our Method

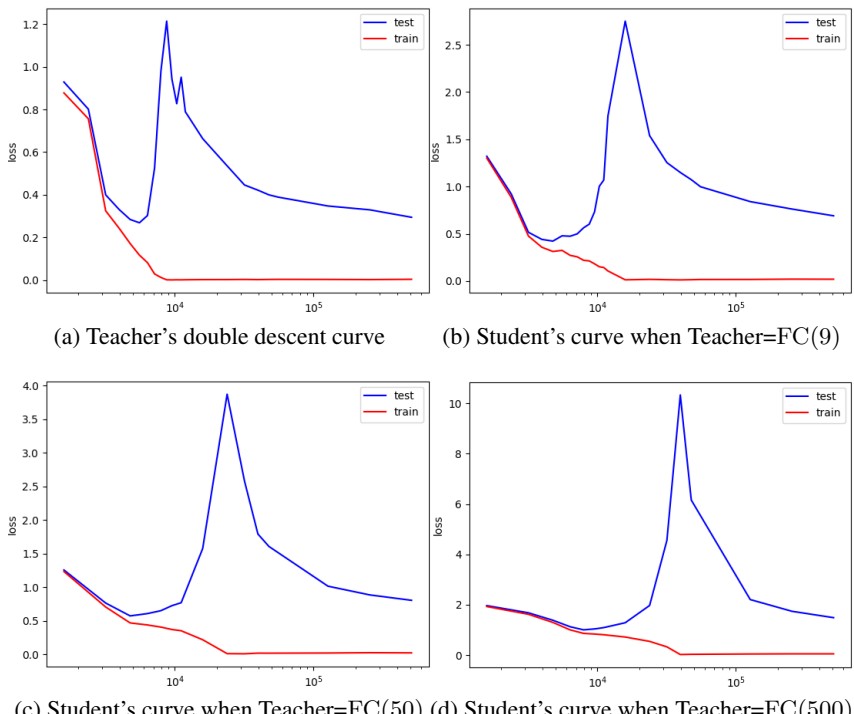

(a) Teacher's double descent curve    (b) Student's curve when Teacher=FC(9)

(c) Student's curve when Teacher=FC(50) (d) Student's curve when Teacher=FC(500)

Figure 26: Double descent curves of the students trained on data with synthetic labels (Loss v.s. Parameter number).

We also perform experiments to verify the intuition behind our methodology, i.e., the method gives a synthetic labeling function with roughly the same complexity on the original inputs and the injected inputs. We first use our method on MNIST and samples (of the same size as MNIST) from a Gaussian to get the teacher $FC(9)$; the double descent curve is in Figure 26(a). Then we train students on the Gaussian data with synthetic labels from the teacher, and plot the double descent curve for the students in Figure 26(b). The local minimums of the two U-shapes are roughly the same, matching our reasoning. Then we also train larger teachers and plot the double descent curve for students on Gaussian data. Figure 26(c) Teacher size $50$. Figure 26(d) Teacher size $500$. The local minimum of the U-shape becomes larger when the teacher gets larger, again matching our reasoning.

# F    PROVABLE GUARANTEES FOR NEURAL NETWORKS IN A MORE GENERAL SETTING

This section provides the analysis in a more general setting. We first describe the learning problems, and then provide the proofs following similar intuitions as for the simpler settings in the main text.

## F.1    PROBLEM SETUP

Let $\mathcal{X} = \mathbb{R}^d$ be the input space, and $\mathcal{Y} = \{\pm 1\}$ be the label space. Suppose $M \in \mathbb{R}^{d \times D}$ is a dictionary with $D$ elements, where each element $M_j$ can be regarded as a pattern. We assume quite general incoherent dictionary:

**(D)** $M$ is $\mu$-incoherent, i.e., the columns of $M$ are unit vectors, and for any $i \neq j, |\langle M_i, M_j \rangle| \leq \mu/\sqrt{d}$.

Note that the setting in the main text corresponds to $\mu = 0$.

Let $\tilde{\phi} \in \{0, 1\}^D$ be a hidden vector that indicates the presence of each pattern, and $\mathcal{D}_{\tilde{\phi}}$ a distribution for $\tilde{\phi}$. Let $A \subseteq [D]$ be a subset of size $k$ corresponding to the class relevant patterns. Let $P \subseteq [k]$. We first sample $\tilde{\phi}$ from $\mathcal{D}_{\tilde{\phi}}$, and then generate the input $\tilde{x}$ and the class label $y$ from $\tilde{\phi}, A, P$ by:

$$\tilde{x} = M\tilde{\phi} + \zeta, \quad y = \begin{cases} +1, & \text{if } \sum_{i \in A} \tilde{\phi}_i \in P, \\ -1, & \text{otherwise} \end{cases} \tag{294}$$

where the Gaussian noise $\zeta \sim \mathcal{N}(0, \sigma_\zeta^2 I_{d \times d})$ is independent from $\tilde{\phi}$. Note that the setting in the main text corresponds to $\sigma_\zeta = 0$.

We allow general $\mathcal{D}_{\tilde{\phi}}$ with the following assumptions:

**(A1)** The patterns in $A$ are correlated with the labels: for any $i \in A$, for $v \in \{\pm 1\}$ let $\gamma_v = \mathbb{E}[y\tilde{\phi}_i | y = v]$, then $\gamma := (\gamma_{+1} + \gamma_{-1})/2 > 0$.

**(A2)** The patterns outside $A$ are independent of the patterns in $A$.

Note that we allow imbalanced classes. Let $p_{\min} := \min(\Pr[y = -1], \Pr[y = +1])$. If the classes are balanced, then the assumption **(A1)** implies the assumption **(A1)** in the main text, so the setting here is more general. **(A2)** is also more general, in particular, allowing dependence between irrelevant patterns and non-identical distributions for them.

Let $\mathcal{D}(A, P, \mathcal{D}_{\tilde{\phi}})$ denote the distribution on $(\tilde{x}, y)$ corresponding to some $A, P$, and $\mathcal{D}_{\tilde{\phi}}$. Given parameters $\Xi = (d, D, k, \gamma, p_o, \mu, \sigma_\zeta)$, the family $\mathcal{F}_\Xi$ of distributions for learning is the set of all $\mathcal{D}(A, P, \mathcal{D}_{\tilde{\phi}})$ with $A \subseteq [D], P \subseteq [k]$, and $\mathcal{D}_{\tilde{\phi}}$ satisfying the above assumptions.

One special case is the mixture of two Gaussians.

*Example.* Suppose $M$ has one single column $v$, and $y = +1$ if $\tilde{\phi} = 1$ and $y = -1$ otherwise. Then the data distribution is simply a mixture of two Gaussians: $\tilde{x} \sim \frac{v}{2} + \mathcal{N}(y\frac{v}{2}, \sigma_\zeta^2 I_{d \times d})$.

### F.1.1    NEURAL NETWORK LEARNING

Again, we will normalize the data for learning: we first compute $x = (\tilde{x} - \mathbb{E}[\tilde{x}])/\tilde{\sigma}$ where $\tilde{\sigma}^2 := \sum_{i=1}^d (\tilde{x}_i - \mathbb{E}[\tilde{x}_i])^2 = \sum_{j \in [D]} \text{Var}(\tilde{\phi}_j) + d\sigma_\zeta^2$ is the variance of the data, and then train on $(x, y)$. This is equivalent to setting $\phi = (\tilde{\phi} - \mathbb{E}[\tilde{\phi}])/\tilde{\sigma}$ and generating $x = M\phi + \zeta/\sigma_\zeta$. For $(\tilde{x}, y)$ from $\mathcal{D}$ and the normalized $(x, y)$, we will simply say $(x, y) \sim \mathcal{D}$.

The learning will be the same as that in the main text, except the following. We will use a small $\sigma_w^2 = \tilde{\sigma}^2/\text{poly}(Dm)$. And we will use a weighted loss to handle the imbalanced classes in the first two steps for feature learning, and then use the unweighted loss in the remaining steps. Formally, the weighted loss is:

$$L_\mathcal{D}^\alpha(g; \sigma_\xi) = \mathbb{E}_{(x,y)}[\alpha_y \ell(y, g(x; \xi))], \tag{295}$$

where the class weights $\alpha_v = \frac{1}{2\Pr[y=v]}$ for $v \in \{\pm 1\}$.

## F.2   Main Result

In this setting, we have the following theorem:

**Theorem 34.** *Set*

$$\eta^{(1)} = \frac{\gamma^2 p_{\min}\tilde{\sigma}}{km^3}, \lambda_a^{(1)} = 0, \lambda_w^{(1)} = 1/(2\eta^{(1)}), \sigma_\xi^{(1)} = 1/k^{3/2}, \tag{296}$$

$$\eta^{(2)} = 1, \lambda_a^{(2)} = \lambda_w^{(2)} = 1/(2\eta^{(2)}), \sigma_\xi^{(2)} = 1/k^{3/2}, \tag{297}$$

$$\eta^{(t)} = \eta = \frac{k^2}{Tm^{1/3}}, \lambda_a^{(t)} = \lambda_w^{(t)} = \lambda \le \frac{k^3}{\tilde{\sigma}m^{1/3}}, \sigma_\xi^{(t)} = 0, \text{ for } 2 < t \le T. \tag{298}$$

*For any $\delta \in (0, O(1/k^3))$, if $\mu \le O(\sqrt{d}/D)$, $\sigma_\zeta \le O(\min\{1/\tilde{\sigma}, \tilde{\sigma}/\sqrt{d}\})$, $k = \Omega\left(\log^2\left(\frac{Dmd}{\delta\gamma p_{\min}}\right)\right)$, $m \ge \max\{\Omega(k^4), D, d\}$, then we have for any $\mathcal{D} \in \mathcal{F}_\Xi$, with probability at least $1 - \delta$, there exists $t \in [T]$ such that*

$$\Pr[\text{sign}(g^{(t)}(x)) \ne y] \le L_{\mathcal{D}}(g^{(t)}) = O\left(\frac{k^8}{m^{2/3}} + \frac{k^3 T}{m^2} + \frac{k^2 m^{2/3}}{T}\right). \tag{299}$$

*Consequently, for any $\epsilon \in (0, 1)$, if $T = m^{4/3}$, and $m \ge \max\{\Omega(k^{12}/\epsilon^{3/2}), D\}$, then*

$$\Pr[\text{sign}(g^{(t)}(x)) \ne y] \le L_{\mathcal{D}}(g^{(t)}) \le \epsilon. \tag{300}$$

The rest of the section is devoted to the proof of this theorem.

## F.3   Notations

Recall some notations that we will use throughout the analysis.

For a vector $v$ and an index set $I$, let $v_I$ denote the vector containing the entries of $v$ indexed by $I$, and $v_{-I}$ denote the vector containing the entries of $v$ with indices outside $I$.

Let $\rho := M^\top M$. Then we have $\rho_{jj} = 1$ for any $j$, and $|\rho_{j\ell}| \le \mu/\sqrt{d}$ for any $j \ne \ell$.

By initialization, $w_i^{(0)}$ for $i \in [m]$ are i.i.d. copies of the same random variable $w^{(0)} \sim \mathcal{N}(0, \sigma_w^2 I_{d \times d})$; similar for $a^{(0)}$ and $b^{(0)}$. Let $\sigma_{\phi_j}^2 := p_{oj}(1 - p_{oj})/\tilde{\sigma}^2$ denote the variance of $\phi_\ell$ for $\ell \notin A$, where $p_{oj} = \Pr[\tilde{\phi}_j = 1]$. Let $p_o$ be the value such that with probability $1 - \exp(-\Omega(k))$, $\sum_{j \notin A} \tilde{\phi}_j \le p_o(D - k)$ for some $p_o \in [0, 1]$. That is, $p_o$ is an upper bound on the density of $\tilde{\phi}_j$ with high probability.

Let $q_\ell := \langle w^{(0)}, M_\ell \rangle$. Similarly, define $q_{i,\ell}^{(t)} := \langle w_i^{(t)}, M_\ell \rangle$.

We also define the following sets to denote typical initialization. For a fixed $\delta \in (0, 1)$, define

$$\mathcal{G}_w(\delta) := \left\{ w \in \mathbb{R}^d : q_\ell = \langle w, M_\ell \rangle, \frac{\sigma_w^2 d}{2} \le \|w^{(0)}\|_2^2 \le \frac{3\sigma_w^2 d}{2}, \right. \tag{301}$$

$$\frac{\sigma_w^2(D - k)}{2} \le \sum_{\ell \notin A} q_\ell^2 \le \frac{3\sigma_w^2(D - k)}{2},$$

$$\left. \max_\ell |q_\ell| \le \sigma_w\sqrt{2\log(Dm/\delta)} \right\}, \tag{302}$$

$$\mathcal{G}_a(\delta) := \{a \in \mathbb{R} : |a| \le \sigma_a\sqrt{2\log(m/\delta)}\}. \tag{303}$$

$$\mathcal{G}_b(\delta) := \{b \in \mathbb{R} : |b| \le \sigma_b\sqrt{2\log(m/\delta)}\}. \tag{304}$$

## F.4 Existence of a Good Network

We first show that there exists a network that can fit the data distribution.

**Lemma 35.** *Suppose $\frac{k\mu}{\sqrt{d}}\frac{p_oD}{\tilde{\sigma}} \leq \frac{1}{16}$. For any $\mathcal{D} \in \mathcal{F}_\Xi$, there exists a network $g^*(x) = \sum_{i=1}^{n} a_i^* \sigma(\langle w_i^*, x\rangle + b_i^*)$ which satisfies*

$$\Pr_{(x,y)\sim\mathcal{D}}[yg^*(x) \leq 1] \leq \exp(-\Omega(k)) + \exp\left(-\Omega\left(\frac{1}{\sigma_\zeta^2(k+k^2\mu/\sqrt{d})}\right)\right).$$

*Furthermore, the number of neurons $n = 3(k+1)$, $|a_i^*| \leq 64k$, $1/(64k) \leq |b_i^*| \leq 1/4$, $w_i^* = \tilde{\sigma}\sum_{j\in A} M_j/(8k)$, and $|\langle w_i^*, x\rangle + b_i^*| \leq 1$ for any $i \in [n]$ and $(x,y) \sim \mathcal{D}$.*

*Consequently, if furthermore we have $k\mu/\sqrt{d} < 1$ and $\sigma_\zeta < 1/k$, then*

$$\Pr_{(x,y)\sim\mathcal{D}}[yg^*(x) \leq 1] \leq \exp(-\Omega(k)).$$

*Proof of Lemma 35.* Let $w = \tilde{\sigma}\sum_{j\in A} M_j$ and let $u = \sum_{j\in A}\mathbb{E}[\tilde{\phi}_j]$. We have

$$\langle w, x\rangle = \tilde{\sigma}\sum_{j\in A}\langle M_j, M\phi\rangle + \langle w, \zeta/\tilde{\sigma}\rangle \tag{305}$$

$$= \sum_{j\in A}\phi_j + \sum_{j\in A,\ell\neq j}\rho_{j\ell}\phi_\ell + \langle w, \zeta/\tilde{\sigma}\rangle \tag{306}$$

$$= \sum_{j\in A}\tilde{\phi}_j - u + \underbrace{\sum_{j\in A,\ell\neq j}\rho_{j\ell}\phi_\ell + \langle w, \zeta/\tilde{\sigma}\rangle}_{:=\epsilon_x}. \tag{307}$$

With probability $\geq 1 - \exp(-\Omega(k))$, among all $j \notin A$, we have that at most $p_o(D-k)$ of $\phi_j$ are $(1-p_o)/\tilde{\sigma}$, while the others are $-p_o/\tilde{\sigma}$, and thus

$$\left|\sum_{j\in A,\ell\neq j}\rho_{j\ell}\phi_\ell\right| \leq \frac{k\mu}{\sqrt{d}}\frac{p_oD}{\tilde{\sigma}} \leq \frac{1}{16}. \tag{308}$$

Furthermore, $\langle w, \zeta\rangle \sim \mathcal{N}(0, \sigma_\zeta^2\|w\|_2^2)$ and $\|w\|_2^2 \leq \tilde{\sigma}^2(k+k^2\mu/\sqrt{d})$, we have

$$\Pr[|\langle w, \zeta/\tilde{\sigma}\rangle| \leq 1/16] \geq 1 - \exp\left(-\Theta\left(\frac{1}{\sigma_\zeta^2\|w\|_2^2/\tilde{\sigma}^2}\right)\right) \tag{309}$$

$$\geq 1 - \exp\left(-\Theta\left(\frac{1}{\sigma_\zeta^2(k+k^2\mu/\sqrt{d})}\right)\right). \tag{310}$$

For good data points with $\phi$ and $\zeta$ satisfying the above, we have $|\epsilon_x| \leq 1/8$. By Lemma 7,

$$g_1^*(x) := \sum_{p\in P}\delta_{p-\mu,4,1/2}(\langle w, x\rangle) - \sum_{p\notin P, 0\leq p\leq k}\delta_{p-\mu,4,1/2}(\langle w, x\rangle) \tag{311}$$

$$= \sum_{p\in P}\delta_{p,4,1/2}(\langle w, x\rangle + u) - \sum_{p\notin P, 0\leq p\leq k}\delta_{p,4,1/2}(\langle w, x\rangle + u) \tag{312}$$

$$= \sum_{p\in P}\delta_{p,4,1/2}\left(\sum_{j\in A}\tilde{\phi}_j + \epsilon_x\right) - \sum_{p\notin P, 0\leq p\leq k}\delta_{p,4,1/2}\left(\sum_{j\in A}\tilde{\phi}_j + \epsilon_x\right). \tag{313}$$

Then for good data points, we have $yg_1^*(x) \geq 1$. Similarly,

$$g_2^*(x) := \sum_{p \in P} \delta_{p-\mu+1/4,8,1/2}(\langle w, x \rangle) - \sum_{p \notin P, 0 \leq p \leq k} \delta_{p-\mu+1/4,8,1/2}(\langle w, x \rangle) \tag{314}$$

$$= \sum_{p \in P} \delta_{p+1/4,8,1/2}(\langle w, x \rangle + u) - \sum_{p \notin P, 0 \leq p \leq k} \delta_{p+1/4,8,1/2}(\langle w, x \rangle + u) \tag{315}$$

$$= \sum_{p \in P} \delta_{p+1/4,8,1/2}\left(\sum_{j \in A} \tilde{\phi}_j + \epsilon_x\right) - \sum_{p \notin P, 0 \leq p \leq k} \delta_{p+1/4,8,1/2}\left(\sum_{j \in A} \tilde{\phi}_j + \epsilon_x\right). \tag{316}$$

Then for good data points, we have $yg_2^*(x) \geq 1$.

Note that the bias terms in $g_1^*$ and $g_2^*$ have distance at least $1/4$, then at least one of them satisfies that all its bias terms have absolute value $\geq 1/8$. Pick that one and denote it as $g(x) = \sum_{i=1}^n a_i \sigma_r(\langle w_i, x \rangle + b_i)$. By the positive homogeneity of $\sigma_r$, we have

$$g(x) = \sum_{i=1}^n 8ka_i \sigma_r(\langle w_i, x \rangle/(8k) + b_i/(8k)). \tag{317}$$

Since for any good data points, $|\langle w_i, x \rangle/(8k) + b_i/(8k)| \leq 1$, then

$$g(x) = \sum_{i=1}^n 8ka_i \sigma(\langle w_i, x \rangle/(8k) + b_i/(8k)) \tag{318}$$

where $\sigma$ is the truncated ReLU. Now we can set $a_i^* = 8ka_i, w_i^* = w_i/(8k), b_i^* = b_i/(8k)$, to get our final $g^*$. □

## F.5 INITIALIZATION

We first show that with high probability, the initial weights are in typical positions.

**Lemma 36.** *Suppose $D\mu/\sqrt{d} \leq 1/16$. For any $\delta \in (0, 1)$, with probability at least $1 - \delta - 2\exp(-\Theta(D - k))$ over $w^{(0)}$,*

$$\sigma_w^2 d/2 \leq \|w^{(0)}\|_2^2 \leq 3\sigma_w^2 d/2,$$

$$\sigma_w^2 (D - k)/2 \leq \sum_{\ell \notin A} q_\ell^2 \leq 3\sigma_w^2(D - k)/2,$$

$$\max_\ell |q_\ell| \leq \sigma_w \sqrt{2\log(D/\delta)}.$$

*With probability at least $1 - \delta$ over $b^{(0)}$,*

$$|b^{(0)}| \leq \sigma_b \sqrt{2\log(1/\delta)}.$$

*With probability at least $1 - \delta$ over $a^{(0)}$,*

$$|a^{(0)}| \leq \sigma_a \sqrt{2\log(1/\delta)}.$$

*Proof of Lemma 36.* The bound on $\|w^{(0)}\|_2^2$ follows from the property of Gaussians.

Note that $q = M^\top w^{(0)} \sim \mathcal{N}(0, \sigma_w^2 \rho)$ for the matrix $\rho = M^\top M$. We have with probability $\geq 1 - \delta/2$, $\max_\ell |q_\ell| \leq \sqrt{2\sigma_w^2 \log \frac{D}{\delta}}$.

For any subset $S \subseteq [D]$, let $\rho_S$ denote the submatrix of $\rho$ containing the rows and columns indexed by $S$. Then $q_S = M^\top w^{(0)} \sim \mathcal{N}(0, \sigma_w^2 \rho_S)$. By diagonalizing $\rho_S$ and then applying Bernstein's inequality, we have with probability $\geq 1 - 2\exp(-\Theta(|S|/\|\rho\|_2))$, $\|q_S\|_2^2 \in \left((\|\rho_S\|_F^2 - \frac{|S|}{4})\sigma_w^2, (\|\rho_S\|_F^2 + \frac{|S|}{4})\sigma_w^2\right)$. By Gershgorin circle theorem, we have

$$\|\rho\|_2 \leq 1 + (|S| - 1)\mu/\sqrt{d} \leq 17/16.$$

Similarly, we have

$$\frac{3}{4}|S| \le \left(\frac{15}{16}\right)^2 |S| \le \|\rho_S\|_F^2 \le \left(\frac{17}{16}\right)^2 |S| \le \frac{5}{4}|S|.$$

The bounds on $q$ then follow.

The bounds on $b^{(0)}$ and $a^{(0)}$ follow from the property of Gaussians. □

**Lemma 37.** *Suppose $D\mu/\sqrt{d} \le 1/16$. We have:*

- *With probability $\ge 1 - \delta - 2m \exp(-\Theta(D-k))$ over $w_i^{(0)}$'s, for all $i \in [2m]$, $w_i^{(0)} \in \mathcal{G}_w(\delta)$.*

- *With probability $\ge 1 - \delta$ over $b_i^{(0)}$'s, for all $i \in [2m]$, $b_i^{(0)} \in \mathcal{G}_b(\delta)$.*

- *With probability $\ge 1 - \delta$ over $a_i^{(0)}$'s, for all $i \in [2m]$, $a_i^{(0)} \in \mathcal{G}_a(\delta)$.*

*Proof of Lemma 37.* This follows from Lemma 36 by union bound. □

### F.6 SOME AUXILIARY LEMMAS

The expression of the gradients will be used frequently.

**Lemma 38.**

$$\frac{\partial}{\partial w_i} L_{\mathcal{D}}^{\alpha}(g; \sigma_{\xi}) = -a_i \mathbb{E}_{(x,y)\sim\mathcal{D}} \left\{\alpha_y y \mathbb{I}[yg(x;\xi) \le 1] \mathbb{E}_{\xi_i} \mathbb{I}[\langle w_i, x\rangle + b_i + \xi_i \in (0,1)]x\right\}, \quad (319)$$

$$\frac{\partial}{\partial b_i} L_{\mathcal{D}}^{\alpha}(g; \sigma_{\xi}) = -a_i \mathbb{E}_{(x,y)\sim\mathcal{D}} \left\{\alpha_y y \mathbb{I}[yg(x;\xi) \le 1] \mathbb{E}_{\xi_i} \mathbb{I}[\langle w_i, x\rangle + b_i \in (0,1)]\right\}, \quad (320)$$

$$\frac{\partial}{\partial a_i} L_{\mathcal{D}}^{\alpha}(g; \sigma_{\xi}) = -\mathbb{E}_{(x,y)\sim\mathcal{D}} \left\{\alpha_y y \mathbb{I}[yg(x;\xi) \le 1] \mathbb{E}_{\xi_i} \sigma(\langle w_i, x\rangle + b_i + \xi_i)\right\}. \quad (321)$$

*Proof of Lemma 38.* It follows from straightforward calculation. □

We also have the following auxiliary lemma for later calculations.

**Lemma 39.**

$$\mathbb{E}_{\phi_A} \{\alpha_y y\} = 0, \quad (322)$$

$$\mathbb{E}_{\phi_A} \{|\alpha_y y|\} = 1, \quad (323)$$

$$\mathbb{E}_{\phi_j} \{|\phi_j|\} = 2\sigma_{\phi_j}^2 \tilde{\sigma}, \text{ for } j \notin A, \quad (324)$$

$$\mathbb{E}_{\phi_A} \{\alpha_y y \phi_j\} = \frac{\gamma}{\tilde{\sigma}}, \text{ for } j \in A, \quad (325)$$

$$\mathbb{E}_{\phi_A} \{|\alpha_y y \phi_j|\} \le \frac{1}{\tilde{\sigma}}, \text{ for all } j \in [D]. \quad (326)$$

*Proof of Lemma 39.*

$$\mathbb{E}_{\phi_A}\{\alpha_y y\} = \sum_{v \in \{\pm 1\}} \mathbb{E}_{\phi_A}\{\alpha_y y | y = v\} \Pr[y = v] \tag{327}$$

$$= \frac{1}{2} \sum_{v \in \{\pm 1\}} \mathbb{E}_{\phi_A}\{y | y = v\} \tag{328}$$

$$= 0. \tag{329}$$

$$\mathbb{E}_{\phi_A}\{|\alpha_y y|\} = \sum_{v \in \{\pm 1\}} \mathbb{E}_{\phi_A}\{|\alpha_y y| \,| y = v\} \Pr[y = v] \tag{330}$$

$$= \frac{1}{2} \sum_{v \in \{\pm 1\}} \mathbb{E}_{\phi_A}\{|y| \,| y = v\} \tag{331}$$

$$= 1. \tag{332}$$

$$\mathbb{E}_{\phi_j}\{|\phi_j|\} = \frac{|-p_{\mathrm{o}j}|(1 - p_{\mathrm{o}j}) + |1 - p_{\mathrm{o}j}|p_{\mathrm{o}j}}{\tilde{\sigma}} = 2\sigma_{\phi_j}^2 \tilde{\sigma}. \tag{333}$$

$$\mathbb{E}_{\phi_A}\{\alpha_y y \phi_j\} = \sum_{v \in \{\pm 1\}} \mathbb{E}_{\phi_A}\{\alpha_y y \phi_j | \, y = v\} \Pr[y = v] \tag{334}$$

$$= \frac{1}{2} \sum_{v \in \{\pm 1\}} \mathbb{E}_{\phi_A}\{y \phi_j | \, y = v\} \tag{335}$$

$$= \frac{1}{2} \sum_{v \in \{\pm 1\}} \mathbb{E}_{\phi_A}\left\{ y \frac{\tilde{\phi}_j - \mathbb{E}[\tilde{\phi}_j]}{\tilde{\sigma}} \Bigg| y = v \right\} \tag{336}$$

$$= \frac{1}{2\tilde{\sigma}}(\gamma_{+1} + \gamma_{-1}) = \frac{\gamma}{\tilde{\sigma}}. \tag{337}$$

$$\mathbb{E}_{\phi_A}\{|\alpha_y y \phi_j|\} = \sum_{v \in \{\pm 1\}} \mathbb{E}_{\phi_A}\{|\alpha_v y \phi_j| \,| y = v\} \Pr[y = v] \tag{338}$$

$$\leq \frac{1}{2} \sum_{v \in \{\pm 1\}} \mathbb{E}_{\phi_A}\{|y \phi_j| \,| y = v\} \tag{339}$$

$$\leq \frac{1}{2} \sum_{v \in \{\pm 1\}} \mathbb{E}_{\phi_A}\{|y \phi_j| \,| y = v\} \tag{340}$$

$$\leq \frac{1}{\tilde{\sigma}}. \tag{341}$$

$\square$

## F.7 FEATURE EMERGENCE: FIRST GRADIENT STEP

We will show that w.h.p. over the initialization, after the first gradient step, there are neurons that represent good features.

We begin with analyzing the gradients.

**Lemma 40.** *Fix $\delta \in (0, 1)$ and suppose $w_i^{(0)} \in \mathcal{G}_w(\delta), b_i^{(0)} \in \mathcal{G}_b(\delta)$ for all $i \in [2m]$. Let*

$$\epsilon_e := \frac{D\sigma_w\sqrt{2\log(D/\delta)}}{\tilde{\sigma}^2 \sigma_\xi^{(1)}} + \frac{\sqrt{d}\sigma_\xi \sigma_w \sqrt{2\log(D/\delta)}}{\tilde{\sigma}\sigma_\xi^{(1)}}, \epsilon_\nu := \epsilon_e.$$

*If $\sigma_\zeta^2 \sigma_w^2 d/\tilde{\sigma}^2 = O(1/k)$, $p_o = \Omega(k^2/D)$, $k = \Omega(\log^2(Dmd/\delta))$, and $\sigma_\xi^{(1)} = O(1/k)$, then*

$$\frac{\partial}{\partial w_i} L_{\mathcal{D}}^\alpha(g^{(0)}; \sigma_\xi^{(1)}) = -a_i^{(0)} \left( \sum_{j=1}^{D} M_j T_j + \nu \right) \tag{342}$$

*where $T_j$ satisfies:*

- *if $j \in A$, then $|T_j - \beta\gamma/\tilde{\sigma}| \leq O(\epsilon_e/\tilde{\sigma})$, where $\beta \in [\Omega(1), 1]$ and depends only on $w_i^{(0)}, b_i^{(0)}$;*

- *if $j \notin A$, then $|T_j| \leq O(\sigma_{\phi_j}^2 \epsilon_e \tilde{\sigma})$;*

- $|\nu_j| \leq O\left(\frac{\sigma_\zeta \sqrt{\log(k)}}{\tilde{\sigma}} \epsilon_\nu\right) + \frac{\sigma_\zeta d}{\tilde{\sigma}} e^{-\Theta(k)}.$

*Proof of Lemma 40.* Consider one neuron index $i$ and omit the subscript $i$ in the parameters. Since the unbiased initialization leads to $g^{(0)}(x; \xi^{(1)}) = 0$, we have

$$\frac{\partial}{\partial w} L_{\mathcal{D}}^\alpha(g^{(0)}; \sigma_\xi^{(1)}) \tag{343}$$

$$= -a^{(0)} \mathbb{E}_{(x,y)\sim\mathcal{D}} \left\{ \alpha_y y \mathbb{I}[yg^{(0)}(x; \xi^{(1)}) \leq 1] \mathbb{E}_{\xi^{(1)}} \mathbb{I}[\langle w^{(0)}, x\rangle + b^{(0)} + \xi^{(1)} \in (0, 1)]x \right\} \tag{344}$$

$$= -a^{(0)} \mathbb{E}_{(x,y)\sim\mathcal{D},\xi^{(1)}} \left\{ \alpha_y y \mathbb{I}[\langle w^{(0)}, x\rangle + b^{(0)} + \xi^{(1)} \in (0, 1)]x \right\} \tag{345}$$

$$= -a^{(0)} \sum_{j=1}^D M_j \underbrace{\mathbb{E}_{(x,y)\sim\mathcal{D},\xi^{(1)}} \left\{ \alpha_y y \phi_j \mathbb{I}[\langle w^{(0)}, x\rangle + b^{(0)} + \xi^{(1)} \in (0, 1)] \right\}}_{:=T_j} \tag{346}$$

$$- a^{(0)} \underbrace{\mathbb{E}_{(x,y)\sim\mathcal{D},\xi^{(1)}} \left\{ \frac{\alpha_y y \zeta}{\tilde{\sigma}} \mathbb{I}[\langle w^{(0)}, x\rangle + b^{(0)} + \xi^{(1)} \in (0, 1)] \right\}}_{:=\nu} \tag{347}$$

First, consider $j \in A$.

$$T_j = \mathbb{E}_{(x,y)\sim\mathcal{D},\xi^{(1)}} \left\{ \alpha_y y \phi_j \mathbb{I}[\langle w^{(0)}, x\rangle + b^{(0)} + \xi^{(1)} \in (0, 1)] \right\} \tag{348}$$

$$= \mathbb{E}_{\phi_A,\zeta} \left\{ \alpha_y y \phi_j \Pr_{\phi_{-A},\xi^{(1)}} \left[ \langle \phi, q\rangle + \iota + b^{(0)} + \xi^{(1)} \in (0, 1) \right] \right\}. \tag{349}$$

where $\iota := \langle w^{(0)}, \zeta/\tilde{\sigma}\rangle$.

Let

$$I_a := \Pr_{\xi^{(1)}} \left[ \langle \phi, q\rangle + \iota + b^{(0)} + \xi^{(1)} \in (0, 1) \right], \tag{350}$$

$$I_a' := \Pr_{\xi^{(1)}} \left[ \langle \phi_{-A}, q_{-A}\rangle + \iota + b^{(0)} + \xi^{(1)} \in (0, 1) \right]. \tag{351}$$

Note that $|\langle \phi_A, q_A\rangle| = O(\frac{k\sigma_w \sqrt{2\log(D/\delta)}}{\tilde{\sigma}^2})$, and that $|\iota| = |\langle w_i^{(0)}, \zeta/\tilde{\sigma}\rangle| = O(\frac{\sqrt{d}\sigma_\xi \sigma_w \sqrt{2\log(D/\delta)}}{\tilde{\sigma}})$, and that $|\langle \phi, q\rangle|, |\langle \phi_{-A}, q_{-A}\rangle|$ are $O(\frac{D\sigma_w \sqrt{2\log(D/\delta)}}{\tilde{\sigma}^2})$. When $\sigma_w$ is sufficiently small, by the property of the Gaussian $\xi^{(1)}$, we have

$$|I_a - I_a'| \tag{352}$$

$$\leq \left| \Pr_{\xi^{(1)}} \left[ \langle \phi, q\rangle + \iota + b^{(0)} + \xi^{(1)} \geq 0 \right] - \Pr_{\xi^{(1)}} \left[ \langle \phi_{-A}, q_{-A}\rangle + \iota + b^{(0)} + \xi^{(1)} \geq 0 \right] \right| \tag{353}$$

$$+ \Pr_{\xi^{(1)}} \left[ \langle \phi, q\rangle + \iota + b^{(0)} + \xi^{(1)} \geq 1 \right] + \Pr_{\xi^{(1)}} \left[ \langle \phi_{-A}, q_{-A}\rangle + \iota + b^{(0)} + \xi^{(1)} \geq 1 \right] \tag{354}$$

$$= O(\epsilon_e). \tag{355}$$

In summary,

$$|\mathbb{E}_{\zeta,\phi_{-A}}(I_a - I_a')| = O(\epsilon_e). \tag{356}$$

Then we have

$$\left| T_j - \mathbb{E}_{\phi_A,\zeta,\phi_{-A}} \left\{ \alpha_y y \phi_j I_a' \right\} \right| \tag{357}$$

$$\leq \mathbb{E}_{\phi_A} \left\{ |\alpha_y y \phi_j| \left| \mathbb{E}_{\zeta,\phi_{-A}}(I_a - I_a') \right| \right\} \tag{358}$$

$$\leq O(\epsilon_e) \mathbb{E}_{\phi_A} \left\{ |\alpha_y y \phi_j| \right\} \tag{359}$$

$$\leq O(\epsilon_e/\tilde{\sigma}) \tag{360}$$

where the last step is from Lemma 39. Furthermore,

$$\mathbb{E}_{\phi_A, \zeta, \phi_{-A}} \{\alpha_y y \phi_j I'_a\} \tag{361}$$

$$= \mathbb{E}_{\phi_A} \{\alpha_y y \phi_j\} \, \mathbb{E}_{\zeta, \phi_{-A}} [I'_a] \tag{362}$$

$$= \mathbb{E}_{\phi_A} \{\alpha_y y \phi_j\} \Pr_{\phi_{-A}, \zeta, \phi_{-A}} \left[ \langle \phi_{-A}, q_{-A} \rangle + \iota + b^{(0)} + \xi^{(1)} \in (0, 1) \right] \tag{363}$$

When $\sigma_w$ is sufficiently small, we have

$$\Pr_{\phi_{-A}} \left[ \langle \phi_{-A}, q_{-A} \rangle + b^{(0)} \in (0, 1/2) \right] \geq \Omega(1), \tag{364}$$

$$\Pr_{\zeta, \xi^{(1)}} \left[ \iota + \xi^{(1)} \in (0, 1/2) \right] = 1/2 - \exp(-\Omega(k)), \tag{365}$$

This leads to

$$\beta := \mathbb{E}_{\zeta, \phi_{-A}} [I'_a] = \Pr_{\phi_{-A}, \zeta, \xi^{(1)}} \left[ \langle \phi_{-A}, q_{-A} \rangle + \iota + b^{(0)} + \xi^{(1)} \in (0, 1) \right] \geq \Omega(1). \tag{366}$$

By Lemma 39, $\mathbb{E}_{\phi_A} \{\alpha_y y \phi_j\} = \gamma / \tilde{\sigma}$. Therefore,

$$|T_j - \beta \gamma / \tilde{\sigma}| \leq O(\epsilon_e / \tilde{\sigma}). \tag{367}$$

Now, consider $j \notin A$. Let $B$ denote $A \cup \{j\}$.

$$T_j = \mathbb{E}_{(x,y) \sim \mathcal{D}, \zeta, \xi^{(1)}} \left\{ \alpha_y y \phi_j \mathbb{I} \left[ \langle \phi, q \rangle + \iota + b^{(0)} + \xi^{(1)} \in (0, 1) \right] \right\} \tag{368}$$

$$= \mathbb{E}_{\phi_B} \mathbb{E}_{\phi_{-B}, \zeta, \xi^{(1)}} \left\{ \alpha_y y \phi_j \mathbb{I} \left[ \langle \phi, q \rangle + \iota + b^{(0)} + \xi^{(1)} \in (0, 1) \right] \right\} \tag{369}$$

$$= \mathbb{E}_{\phi_B, \zeta} \left\{ \alpha_y y \phi_j \Pr_{\phi_{-B}, \xi^{(1)}} \left[ \langle \phi, q \rangle + \iota + b^{(0)} + \xi^{(1)} \in (0, 1) \right] \right\}. \tag{370}$$

Let

$$I_b := \Pr_{\xi^{(1)}} \left[ \langle \phi, q \rangle + \iota + b^{(0)} + \xi^{(1)} \in (0, 1) \right], \tag{371}$$

$$I'_b := \Pr_{\xi^{(1)}} \left[ \langle \phi_{-B}, q_{-B} \rangle + \iota + b^{(0)} + \xi^{(1)} \in (0, 1) \right]. \tag{372}$$

Similar as above, we have $|\mathbb{E}_{\zeta, \xi^{(1)}} (I_b - I'_b)| \leq O(\epsilon_e)$. Then by Lemma 39,

$$\left| T_j - \mathbb{E}_{\phi_B, \zeta, \phi_{-B}} \{\alpha_y y \phi_j I'_b\} \right| \tag{373}$$

$$\leq \mathbb{E}_{\phi_B} \left\{ |\alpha_y y \phi_j| |\mathbb{E}_{\zeta, \phi_{-B}} (I_b - I'_b)| \right\} \tag{374}$$

$$\leq O(\epsilon_e) \mathbb{E}_{\phi_A} \{|\alpha_y y|\} \, \mathbb{E}_{\phi_j} \{|\phi_j|\} \tag{375}$$

$$\leq O(\epsilon_e) \times 1 \times O(\sigma_{\phi_j}^2 \tilde{\sigma}) \tag{376}$$

$$= O(\sigma_{\phi_j}^2 \epsilon_e \tilde{\sigma}). \tag{377}$$

Furthermore,

$$\mathbb{E}_{\phi_B, \zeta, \phi_{-B}} \{\alpha_y y \phi_j I'_b\} = \mathbb{E}_{\phi_A} \{\alpha_y y\} \, \mathbb{E}_{\phi_j} \{\phi_j\} \, \mathbb{E}_{\zeta, \phi_{-B}} [I'_b] = 0. \tag{378}$$

Therefore,

$$|T_j| \leq O(\sigma_\phi^2 \epsilon_e \tilde{\sigma}). \tag{379}$$

Finally, consider $\nu_j$.

$$\nu_j = \mathbb{E}_{(x,y) \sim \mathcal{D}, \xi^{(1)}} \left\{ \frac{\alpha_y y \zeta_j}{\tilde{\sigma}} \mathbb{I}[\langle w^{(0)}, x \rangle + b^{(0)} + \xi^{(1)} \in (0, 1)] \right\} \tag{380}$$

$$= \mathbb{E}_{\phi_A, \phi_{-A}, \zeta, \xi^{(1)}} \left\{ \frac{\alpha_y y \zeta_j}{\tilde{\sigma}} \mathbb{I}[\langle \phi, q \rangle + \iota_j + \iota_{-j} + b^{(0)} + \xi^{(1)} \in (0, 1)] \right\} \tag{381}$$

$$= \mathbb{E}_{\phi_A, \zeta} \left\{ \frac{\alpha_y y \zeta_j}{\tilde{\sigma}} \Pr_{\phi_{-A}, \xi^{(1)}} [\langle \phi, q \rangle + \iota_j + \iota_{-j} + b^{(0)} + \xi^{(1)} \in (0, 1)] \right\} \tag{382}$$

where $\iota_j := w_j^{(0)}\zeta_j/\tilde{\sigma}$ and $\iota_{-j} := \langle w^{(0)}, \zeta/\tilde{\sigma}\rangle - \iota_j$.

With probability $\geq 1 - d\exp(-\Theta(k))$ over $\zeta$, for any $j$, $|\zeta_j| \leq O(\sigma_\zeta\sqrt{\log(k)})$. Let $\mathcal{G}_\zeta$ denote this event.

Let

$$I_j := \Pr_{\xi^{(1)}}\left[\langle\phi, q\rangle + \iota_j + \iota_{-j} + b^{(0)} + \xi^{(1)} \in (0,1)\right], \tag{383}$$

$$I_j' := \Pr_{\xi^{(1)}}\left[\langle\phi, q\rangle + \iota_{-j} + b^{(0)} + \xi^{(1)} \in (0,1)\right]. \tag{384}$$

Similar as above, we have $|\mathbb{E}_\zeta[I_j - I_j'|\mathcal{G}_\zeta]| \leq O(\epsilon_\nu)$. Then

$$|\mathbb{E}_{\zeta,\phi_{-A}}(I_j - I_j')| \leq |\mathbb{E}_{\zeta,\phi_{-A}}[(I_j - I_j')|\mathcal{G}_\zeta]| + \Pr[-\mathcal{G}_\zeta] \tag{385}$$

$$\leq O(\epsilon_\nu + d\exp(-\Theta(k))). \tag{386}$$

$$\left|\nu_j - \mathbb{E}_{\phi_A,\zeta,\phi_{-A}}\left\{\frac{\alpha_y y\zeta_j}{\tilde{\sigma}}I_j'\right\}\right| \tag{387}$$

$$= \left|\mathbb{E}_{\phi_A,\zeta,\phi_{-A}}\left\{\frac{\alpha_y y\zeta_j}{\tilde{\sigma}}(I_j - I_j')\right\}\right| \tag{388}$$

$$\leq \left|\mathbb{E}_{\phi_A,\zeta,\phi_{-A}}\left\{\frac{\alpha_y y\zeta_j}{\tilde{\sigma}}(I_j - I_j')|\mathcal{G}_\zeta\right\}\right| + \left|\mathbb{E}_{\phi_A,\zeta,\phi_{-A}}\left\{\frac{\alpha_y y\zeta_j}{\tilde{\sigma}}(I_j - I_j')| - \mathcal{G}_\zeta\right\}\right|\Pr[-\mathcal{G}_\zeta]. \tag{389}$$

The first term is bounded by

$$\left|\mathbb{E}_{\phi_A,\zeta,\phi_{-A}}\left\{\frac{\alpha_y y\zeta_j}{\tilde{\sigma}}(I_j - I_j')|\mathcal{G}_\zeta\right\}\right| \tag{390}$$

$$\leq \mathbb{E}_{\phi_A}\left\{\frac{\alpha_y y\sigma_\zeta\sqrt{\log(k)}}{\tilde{\sigma}}|\mathbb{E}_{\zeta,\phi_{-A}}[I_b - I_b'|\mathcal{G}_\zeta]|\right\} \tag{391}$$

$$\leq O(\epsilon_\nu)\mathbb{E}_{\phi_A}\{|\alpha_y y|\}\frac{\sigma_\zeta\sqrt{\log(k)}}{\tilde{\sigma}} \tag{392}$$

$$\leq O(\epsilon_\nu)\times 1 \times \frac{\sigma_\zeta\sqrt{\log(k)}}{\tilde{\sigma}} \tag{393}$$

$$= O\left(\frac{\sigma_\zeta\sqrt{\log(k)}}{\tilde{\sigma}}\epsilon_\nu\right). \tag{394}$$

The second term is bounded by

$$\left|\mathbb{E}_{\phi_A,\zeta,\phi_{-A}}\left\{\frac{\alpha_y y\zeta_j}{\tilde{\sigma}}(I_j - I_j')| - \mathcal{G}_\zeta\right\}\right|\Pr[-\mathcal{G}_\zeta] \tag{395}$$

$$\leq \left|\mathbb{E}_{\phi_A,\zeta,\phi_{-A}}\left\{\frac{\alpha_y y\zeta_j}{\tilde{\sigma}}(I_j - I_j')| - \mathcal{G}_\zeta\right\}\right| \times de^{-\Theta(k)} \tag{396}$$

$$\leq \mathbb{E}_{\phi_A}\left|\frac{\alpha_y y}{\tilde{\sigma}}\right| \times \mathbb{E}_\zeta\{|\zeta_j|| - \mathcal{G}_\zeta\} \times de^{-\Theta(k)} \tag{397}$$

$$\leq \frac{\sigma_\zeta}{\tilde{\sigma}} \times de^{-\Theta(k)} \tag{398}$$

$$\leq \frac{\sigma_\zeta d}{\tilde{\sigma}}e^{-\Theta(k)}. \tag{399}$$

Furthermore,

$$\mathbb{E}_{\phi_A,\zeta,\phi_{-A}}\left\{\frac{\alpha_y y\zeta_j}{\tilde{\sigma}}I_j'\right\} = \mathbb{E}_{\phi_A}\{\alpha_y y\}\mathbb{E}_{\zeta_j}\left\{\frac{\zeta_j}{\tilde{\sigma}}\right\}\mathbb{E}_{\zeta_{-j}}[I_j'] = 0. \tag{400}$$

Therefore,

$$|\nu_j| \leq O\left(\frac{\sigma_\zeta\sqrt{\log(k)}}{\tilde{\sigma}}\epsilon_\nu\right) + \frac{\sigma_\zeta d}{\tilde{\sigma}}e^{-\Theta(k)}. \tag{401}$$

$\square$

**Lemma 41.** *Under the same assumptions as in Lemma 40,*

$$\frac{\partial}{\partial b_i} L_{\mathcal{D}}^{\alpha}(g^{(0)}; \sigma_{\xi}^{(1)}) = -a_i^{(0)} T_b \tag{402}$$

*where $|T_b| \leq O(\epsilon_e)$.*

*Proof of Lemma 41.* Consider one neuron index $i$ and omit the subscript $i$ in the parameters. Since the unbiased initialization leads to $g^{(0)}(x; \xi^{(1)}) = 0$, we have

$$\frac{\partial}{\partial b} L_{\mathcal{D}}^{\alpha}(g^{(0)}; \sigma_{\xi}^{(1)}) \tag{403}$$

$$= -a^{(0)} \mathbb{E}_{(x,y) \sim \mathcal{D}} \left\{ \alpha_y y \mathbb{I}[yg^{(0)}(x; \xi^{(1)}) \leq 1] \mathbb{E}_{\xi^{(1)}} \mathbb{I}[\langle w^{(0)}, x \rangle + b^{(0)} + \xi^{(1)} \in (0,1)] \right\} \tag{404}$$

$$= -a^{(0)} \mathbb{E}_{(x,y) \sim \mathcal{D}, \xi^{(1)}} \left\{ \alpha_y y \mathbb{I}[\langle w^{(0)}, x \rangle + b^{(0)} + \xi^{(1)} \in (0,1)] \right\} \tag{405}$$

$$= -a^{(0)} \underbrace{\mathbb{E}_{\phi_A, \zeta, \xi^{(1)}} \left\{ \alpha_y y \Pr_{\phi_{-A}} \left[ \langle \phi, q \rangle + \iota + b^{(0)} + \xi^{(1)} \in (0,1) \right] \right\}}_{:=T_b}. \tag{406}$$

where $\iota := \langle w^{(0)}, \zeta/\tilde{\sigma} \rangle$. Similar to the proof in Lemma 40,

$$\left| \mathbb{E}_{\zeta} \left( \Pr_{\phi_{-A}, \xi^{(1)}} [\langle \phi, q \rangle + \iota + b^{(0)} + \xi^{(1)} \in (0,1)] \right. \right. \tag{407}$$

$$\left. \left. - \Pr_{\phi_{-A}, \xi^{(1)}} [\langle \phi_{-A}, q_{-A} \rangle + \iota + b^{(0)} + \xi^{(1)} \in (0,1)] \right) \right| = O(\epsilon_e). \tag{408}$$

Then

$$\left| T_b - \mathbb{E}_{\phi_A, \zeta} \left\{ \alpha_y y \Pr_{\phi_{-A}, \xi^{(1)}} [\langle \phi_{-A}, q_{-A} \rangle + \iota + b^{(0)} + \xi^{(1)} \in (0,1)] \right\} \right| \tag{409}$$

$$= \mathbb{E}_{\phi_A, \zeta} \left\{ |\alpha_y y| \left| \Pr_{\phi_{-A}, \xi^{(1)}} [\langle \phi, q \rangle + \iota + b^{(0)} + \xi^{(1)} \in (0,1)] \right. \right. \tag{410}$$

$$\left. \left. - \Pr_{\phi_{-A}, \xi^{(1)}} [\langle \phi_{-A}, q_{-A} \rangle + \iota + b^{(0)} + \xi^{(1)} \in (0,1)] \right| \right\} \tag{411}$$

$$\leq O(\epsilon_e) \mathbb{E}_{\phi_A} \{ |\alpha_y y| \} \tag{412}$$

$$\leq O(\epsilon_e). \tag{413}$$

Also,

$$\mathbb{E}_{\phi_A, \zeta} \left\{ \alpha_y y \Pr_{\phi_{-A}, \xi^{(1)}} [\langle \phi_{-A}, q_{-A} \rangle + \iota + b^{(0)} + \xi^{(1)} \in (0,1)] \right\} \tag{414}$$

$$= \mathbb{E}_{\phi_A} \{ \alpha_y y \} \Pr_{\phi_{-A}, \zeta, \xi^{(1)}} [\langle \phi_{-A}, q_{-A} \rangle + \iota + b^{(0)} + \xi^{(1)} \in (0,1)] \tag{415}$$

$$= 0. \tag{416}$$

Therefore, $|T_b| \leq O(\epsilon_e)$. □

**Lemma 42.** *We have*

$$\frac{\partial}{\partial a_i} L_{\mathcal{D}}^{\alpha}(g^{(0)}; \sigma_{\xi}^{(1)}) = -T_a \tag{417}$$

*where $|T_a| \leq O(\max_{\ell} q_{i,\ell}^{(0)})$. So if $w_i^{(0)} \in \mathcal{G}(\delta)$, $|T_a| \leq O(\sigma_w \sqrt{\log(Dm/\delta)})$.*

*Proof of Lemma 42.* Consider one neuron index $i$ and omit the subscript $i$ in the parameters. Since the unbiased initialization leads to $g^{(0)}(x; \xi^{(1)}) = 0$, we have

$$\frac{\partial}{\partial a} L_{\mathcal{D}}^{\alpha}(g^{(0)}; \sigma_{\xi}^{(1)}) \tag{418}$$

$$= -\mathbb{E}_{(x,y)\sim\mathcal{D}} \left\{ \alpha_y y \mathbb{I}[yg^{(0)}(x; \xi^{(1)}) \le 1] \mathbb{E}_{\xi^{(1)}} \sigma(\langle w^{(0)}, x\rangle + b^{(0)} + \xi^{(1)}) \right\} \tag{419}$$

$$= -\underbrace{\mathbb{E}_{(x,y)\sim\mathcal{D}, \xi^{(1)}} \left\{ \alpha_y y \sigma(\langle w^{(0)}, x\rangle + b^{(0)} + \xi^{(1)}) \right\}}_{:=T_a}. \tag{420}$$

Let $\phi'_A$ be an independent copy of $\phi_A$, $\phi'$ be the vector obtained by replacing in $\phi$ the entries $\phi_A$ with $\phi'_A$, and let $x' = M\phi' + \zeta/\tilde{\sigma}$ and its label is $y'$. Then

$$|T_a| = \left| \mathbb{E}_{\phi_A} \left\{ \alpha_y y \mathbb{E}_{\phi_{-A}, \zeta, \xi^{(1)}} \sigma(\langle w^{(0)}, x\rangle + b^{(0)} + \xi^{(1)}) \right\} \right| \tag{421}$$

$$\le \frac{1}{2} \left| \mathbb{E}_{\phi_A} \left\{ \mathbb{E}_{\phi_{-A}, \zeta, \xi^{(1)}} \sigma(\langle w^{(0)}, x\rangle + b^{(0)} + \xi^{(1)}) | y = 1 \right\} \right. \tag{422}$$

$$\left. - \mathbb{E}_{\phi_A} \left\{ \mathbb{E}_{\phi_{-A}, \zeta, \xi^{(1)}} \sigma(\langle w^{(0)}, x\rangle + b^{(0)} + \xi^{(1)}) | y = -1 \right\} \right| \tag{423}$$

$$\le \frac{1}{2} \left| \mathbb{E}_{\phi_A} \left\{ \mathbb{E}_{\phi_{-A}, \zeta, \xi^{(1)}} \sigma(\langle w^{(0)}, x\rangle + b^{(0)} + \xi^{(1)}) | y = 1 \right\} \right. \tag{424}$$

$$\left. - \mathbb{E}_{\phi'_A} \left\{ \mathbb{E}_{\phi_{-A}, \zeta, \xi^{(1)}} \sigma(\langle w^{(0)}, x'\rangle + b^{(0)} + \xi^{(1)}) | y' = -1 \right\} \right|. \tag{425}$$

Since $\sigma$ is 1-Lipschitz,

$$|T_a| \le \frac{1}{2} \mathbb{E}_{\phi_A, \phi'_A} \left\{ \mathbb{E}_{\phi_{-A}} \left| \langle w^{(0)}, M\phi\rangle - \langle w^{(0)}, M\phi'\rangle \right| | y = 1, y' = -1 \right\} \tag{426}$$

$$\le \frac{1}{2} \mathbb{E}_{\phi_{-A}} \left( \mathbb{E}_{\phi_A} \left\{ \left| \langle w^{(0)}, M\phi\rangle \right| | y = 1 \right\} + \mathbb{E}_{\phi'_A} \left\{ \left| \langle w^{(0)}, M\phi'\rangle \right| | y' = -1 \right\} \right) \tag{427}$$

$$\le \max_{\ell} q_{i,\ell}^{(0)} \sqrt{\mathbb{E}_{\phi} \left( \sum_{\ell \in [D]} \phi_{\ell}^2 + \sum_{j \ne \ell: j, \ell \in A} |\phi_j \phi_{\ell}| \right)} \tag{428}$$

$$\le \max_{\ell} q_{i,\ell}^{(0)} \sqrt{\mathbb{E}_{\phi} (1 + O(1))} \tag{429}$$

$$= \Theta(\max_{\ell} q_{i,\ell}^{(0)}). \tag{430}$$

$\square$

With the bounds on the gradient, we now summarize the results for the weights after the first gradient step.

**Lemma 43.** *Set*
$$\lambda_w^{(1)} = 1/(2\eta^{(1)}), \lambda_a^{(1)} = \lambda_b^{(1)} = 0, \sigma_{\xi}^{(1)} = 1/k^{3/2}.$$

*Fix $\delta \in (0, 1)$ and suppose $w_i^{(0)} \in \mathcal{G}_w(\delta), b_i^{(0)} \in \mathcal{G}_b(\delta)$ for all $i \in [2m]$. If $k = \Omega(\log^2(Dm/\delta))$, then for all $i \in [m]$, $w_i^{(1)} = \sum_{\ell=1}^{D} q_{i,\ell}^{(1)} M_{\ell} + \upsilon$ satisfying*

- *if $\ell \in A$, then $|q_{i,\ell}^{(1)} - \eta^{(1)} a_i^{(0)} \beta\gamma/\tilde{\sigma}| \le O\left(\frac{|\eta^{(1)} a_i^{(0)}|\epsilon_e}{\tilde{\sigma}}\right)$, where $\beta \in [\Omega(1), 1]$ and depends only on $w_i^{(0)}, b_i^{(0)}$;*

- *if $\ell \notin A$, then $|q_{i,\ell}^{(1)}| \le O\left(|\eta^{(1)} a_i^{(0)}|\sigma_{\phi_{\ell}}^2 \epsilon_e \tilde{\sigma}\right)$;*

- $|v_j| \leq O\left(|\eta^{(1)}a_i^{(0)}|\left(\frac{\sigma_\zeta\sqrt{\log(k)}}{\tilde{\sigma}}\epsilon_\nu + \frac{\sigma_\zeta d}{\tilde{\sigma}}e^{-\Theta(k)}\right)\right).$

*and*

- $b_i^{(1)} = b_i^{(0)} + \eta^{(1)}a_i^{(0)}T_b$ *where* $|T_b| = O(\epsilon_e)$;
- $a_i^{(1)} = a_i^{(0)} + \eta^{(1)}T_a$ *where* $|T_a| = O(\sigma_w\sqrt{\log(Dm/\delta)})$.

*Proof of Lemma 43.* This follows from Lemma 37 and Lemma 40-42. $\qquad\square$

### F.8 FEATURE IMPROVEMENT: SECOND GRADIENT STEP

We first show that with properly set $\eta^{(1)}$, for most $x$, $|g^{(1)}(x;\sigma_\xi^{(2)})| < 1$ and thus $yg^{(1)}(x;\sigma_\xi^{(2)}) < 1$.

**Lemma 44.** *Fix $\delta \in (0,1)$ and suppose $w_i^{(0)} \in \mathcal{G}_w(\delta), b_i^{(0)} \in \mathcal{G}_b(\delta), a_i^{(0)} \in \mathcal{G}_a(\delta)$ for all $i \in [2m]$. If $D\mu/\sqrt{d} \leq 1/16$, $\sigma_\zeta\tilde{\sigma} = O(1)$, $\sigma_\zeta^2 d/\tilde{\sigma}^2 = O(1)$, $k = \Omega(\log^2(Dm/\delta))$, $\sigma_a \leq \tilde{\sigma}^2/(\gamma k^2)$, $\eta^{(1)} = O\left(\frac{\gamma}{km\sigma_a\tilde{\sigma}}\right)$, and $\sigma_\xi^{(2)} \leq 1/k$, then with probability $\geq 1 - (d+D)\exp(-\Omega(k))$ over $(x,y)$, we have $yg^{(1)}(x;\sigma_\xi^{(2)}) < 1$. Furthermore, for any $i \in [2m]$, $\left|\langle w_i^{(1)}, \zeta/\tilde{\sigma}\rangle\right| = O(\eta^{(1)}\tilde{\sigma}/\gamma)$, $\left|\langle q_i^{(1)}, \phi\rangle\right| = O(\eta^{(1)}\tilde{\sigma}/\gamma)$, and $\left|\langle (q_i^{(1)})_{-A}, \phi_{-A}\rangle\right| = O(\eta^{(1)}\tilde{\sigma}/\gamma)$, and for any $j \in [d], \ell \in [D]$, $|\zeta_j| \leq O(\sigma_\zeta\sqrt{\log(k)})$ and $|\langle \zeta, D_\ell\rangle| \leq O(\sigma_\zeta\sqrt{\log(k)})$.*

*Proof of Lemma 44.* Note that $w_i^{(0)} = w_{m+i}^{(0)}$, $b_i^{(0)} = b_{m+i}^{(0)}$, and $a_i^{(0)} = a_{m+i}^{(0)}$. Then the gradient for $w_{m+i}$ is the negation of that for $w_{m+i}$, the gradient for $b_{m+i}$ is the negation of that for $b_{m+i}$, and the gradient for $a_{m+i}$ is the same as that for $a_{m+i}$.

$$\left|g^{(1)}(x;\sigma_\xi^{(2)})\right| \tag{431}$$

$$= \left|\sum_{i=1}^{2m} a_i^{(1)}\mathbb{E}_{\xi^{(2)}}\sigma(\langle w_i^{(1)}, x\rangle + b_i^{(1)} + \xi_i^{(2)})\right| \tag{432}$$

$$= \left|\sum_{i=1}^{m}\left(a_i^{(1)}\mathbb{E}_{\xi^{(2)}}\sigma(\langle w_i^{(1)}, x\rangle + b_i^{(1)} + \xi_i^{(2)}) + a_{m+i}^{(1)}\mathbb{E}_{\xi^{(2)}}\sigma(\langle w_{m+i}^{(1)}, x\rangle + b_{m+i}^{(1)} + \xi_{m+i}^{(2)})\right)\right| \tag{433}$$

$$\leq \left|\sum_{i=1}^{m}\left(a_i^{(1)}\mathbb{E}_{\xi^{(2)}}\sigma(\langle w_i^{(1)}, x\rangle + b_i^{(1)} + \xi_i^{(2)}) + a_{m+i}^{(1)}\mathbb{E}_{\xi^{(2)}}\sigma(\langle w_i^{(1)}, x\rangle + b_i^{(1)} + \xi_i^{(2)})\right)\right| \tag{434}$$

$$+ \left|\sum_{i=1}^{m}\left(-a_{m+i}^{(1)}\mathbb{E}_{\xi^{(2)}}\sigma(\langle w_i^{(1)}, x\rangle + b_i^{(1)} + \xi_i^{(2)}) + a_{m+i}^{(1)}\mathbb{E}_{\xi^{(2)}}\sigma(\langle w_{m+i}^{(1)}, x\rangle + b_{m+i}^{(1)} + \xi_i^{(2)})\right)\right|. \tag{435}$$

Then we have

$$\left|g^{(1)}(x;\sigma_\xi^{(2)})\right| \leq \sum_{i=1}^{m}\left|2\eta^{(1)}T_a\mathbb{E}_{\xi^{(2)}}\sigma(\langle w_i^{(1)}, x\rangle + b_i^{(1)} + \xi_i^{(2)})\right| \tag{436}$$

$$+ \sum_{i=1}^{m}\left|a_{m+i}^{(1)}\right|\left(\left|\langle w_i^{(1)} - w_{m+i}^{(1)}, x\rangle\right| + \left|b_i^{(1)} - b_{m+i}^{(1)}\right|\right) \tag{437}$$

$$\leq \sum_{i=1}^{m}\left|2\eta^{(1)}T_a\right|\left(\left|\langle w_i^{(1)}, x\rangle + b_i^{(1)}\right| + \mathbb{E}_{\xi^{(2)}}\left|\xi_i^{(2)}\right|\right) \tag{438}$$

$$+ \sum_{i=1}^{m}\left|a_{m+i}^{(1)}\right|\left(\left|\langle w_i^{(1)} - w_{m+i}^{(1)}, x\rangle\right| + \left|b_i^{(1)} - b_{m+i}^{(1)}\right|\right). \tag{439}$$

With probability $\geq 1 - \exp(-\Omega(k))$, among all $j \notin A$, we have that at most $p_{\mathrm{o}}(D - k)$ of $\phi_j$ are $(1 - p_{\mathrm{o}j})/\tilde{\sigma}$, while the others are $-p_{\mathrm{o}j}/\tilde{\sigma}$. With probability $\geq 1 - (d + D)\exp(-\Omega(k))$ over $\zeta$, for any $j$, $|\zeta_j| \leq O(\sigma_\zeta \sqrt{\log(k)})$ and $|\langle \zeta, D_\ell \rangle| \leq O(\sigma_\zeta \sqrt{\log(k)})$. For data points with $\phi$ and $\zeta$ satisfying these, we have:

**Claim 1.** $\left| \langle w_i^{(1)}, x \rangle \right| \leq O(\eta^{(1)}/\gamma)(1 + \tilde{\sigma} + \tilde{\sigma}/\sqrt{k})$.

*Proof of Claim 1.*

$$\left| \langle w_i^{(1)}, x \rangle \right| = \left| \langle \sum_{\ell=1}^{D} q_{i,\ell}^{(1)} M_\ell + \upsilon, \sum_{j=1}^{D} \phi_j M_j + \zeta/\tilde{\sigma} \rangle \right| \tag{440}$$

$$\leq \left| \langle \sum_{\ell=1}^{D} q_{i,\ell}^{(1)} M_\ell, \sum_{j=1}^{D} \phi_j M_j \rangle \right| + \left| \langle \sum_{\ell=1}^{D} q_{i,\ell}^{(1)} M_\ell, \zeta/\tilde{\sigma} \rangle \right| + \left| \langle \upsilon, \sum_{j=1}^{D} \phi_j M_j \rangle \right| + |\langle \upsilon, \zeta/\tilde{\sigma} \rangle|. \tag{441}$$

For each term above we bound as follows. Note that when $\sigma_w$ is sufficiently small, $\epsilon_{\mathrm{e}} = O(k \log^{1/2}(Dm/\delta)/\sqrt{D})$. Let

$$B_1 := \beta\gamma/\tilde{\sigma} + \epsilon_{\mathrm{e}}/\tilde{\sigma}, \tag{442}$$

$$B_2 := \sigma_\phi^2 \epsilon_{\mathrm{e}} \tilde{\sigma} = O(\epsilon_{\mathrm{e}}/\sqrt{D}), \tag{443}$$

$$C_1 = \frac{k}{\tilde{\sigma}}, \tag{444}$$

$$C_2 := p_{\mathrm{o}} D/\tilde{\sigma} = O(D/\tilde{\sigma}). \tag{445}$$

Then

$$|a_i^{(0)}| B_1 C_1 = O(\log(Dm/\delta)/k + \log(Dm/\delta)\epsilon_{\mathrm{e}}/(\gamma k)) = O(1/\gamma), \tag{446}$$

$$|a_i^{(0)}| B_2 C_2 = O(\tilde{\sigma}/\gamma), \tag{447}$$

$$|a_i^{(0)}| B_1 C_2 = O(D/k + \sqrt{D}/\gamma), \tag{448}$$

$$|a_i^{(0)}| B_2 C_1 = O(\epsilon_{\mathrm{e}}/(\gamma\sqrt{k})) = O(1/\gamma), \tag{449}$$

Then by the assumption on $\mu$,

$$\left| \langle \sum_{\ell=1}^{D} q_{i,\ell}^{(1)} M_\ell, \sum_{j=1}^{D} \phi_j M_j \rangle \right| \tag{450}$$

$$= \left| \sum_{\ell \in A} \langle q_{i,\ell}^{(1)} M_\ell, M_\ell \phi_\ell \rangle \right| + \left| \sum_{\ell \notin A} \langle q_{i,\ell}^{(1)} M_\ell, M_\ell \phi_\ell \rangle \right| + \left| \sum_{\ell \neq j} \langle q_{i,\ell}^{(1)} M_\ell, M_j \phi_j \rangle \right| \tag{451}$$

$$\leq O(|\eta^{(1)} a_i^{(0)}|) \left( B_1 C_1 + B_2 C_2 + \frac{\mu}{\sqrt{d}}(k B_1 (C_1 + C_2) + D B_2 (C_1 + C_2)) \right) \tag{452}$$

$$\leq O(|\eta^{(1)} a_i^{(0)}|) \left( B_1 C_1 + B_2 C_2 + \frac{k\mu}{\sqrt{d}} B_1 C_2 + B_2 C_1 \right) \tag{453}$$

$$\leq O(\eta^{(1)})(1/\gamma + \tilde{\sigma}/\gamma + 1/\gamma + 1/\gamma) \tag{454}$$

$$\leq O(\eta^{(1)}/\gamma)(1 + \tilde{\sigma}). \tag{455}$$

By the assumption on $\sigma_\zeta$,

$$\left| \langle \sum_{\ell=1}^{D} q_{i,\ell}^{(1)} M_\ell, \zeta/\tilde{\sigma} \rangle \right| \tag{456}$$

$$\leq O(|\eta^{(1)} a_i^{(0)}|)(kB_1 + DB_2)\frac{\sigma_\zeta \sqrt{\log(k)}}{\tilde{\sigma}} \tag{457}$$

$$\leq O(\eta^{(1)})(kB_1 + DB_2)\frac{\sigma_\zeta \sqrt{\log(k)}}{\tilde{\sigma}}\frac{\tilde{\sigma}^2 \sqrt{\log(Dm/\delta)}}{\gamma k^2} \tag{458}$$

$$\leq O(\eta^{(1)})\left( \sigma_\zeta \log(Dm/\delta)\left( \frac{1}{k} + \frac{\epsilon_e}{\gamma k} \right) + \sigma_\zeta \tilde{\sigma}\frac{\log(k)\log(Dm/\delta)}{\gamma k} \right) \tag{459}$$

$$\leq O(\eta^{(1)}/\gamma). \tag{460}$$

Also note that $|\nu_j| \leq O\left( \frac{\sigma_\zeta \log^2(Dm/\delta)}{\tilde{\sigma}\sqrt{D}} \right)$. Then by the assumption on $\sigma_\zeta$,

$$\left| \langle v, \sum_{j=1}^{D} \phi_j M_j \rangle \right| \tag{461}$$

$$\leq O(|\eta^{(1)} a_i^{(0)}|) \times \sqrt{d} \times O\left( \frac{\sigma_\zeta \log^2(Dm/\delta)}{\tilde{\sigma}\sqrt{D}} \right) \times (C_1 + C_2) \tag{462}$$

$$\leq O(|\eta^{(1)}/\gamma). \tag{463}$$

Finally, we have

$$|\langle v, \zeta/\tilde{\sigma} \rangle| \leq \sum_{j=1}^{d} |v_j||\zeta_j/\tilde{\sigma}| \tag{464}$$

$$\leq O(|\eta^{(1)} a_i^{(0)}|) \times d \times \frac{\sigma_\zeta \log^2(Dm/\delta)}{\tilde{\sigma}\sqrt{D}}\frac{\tilde{\sigma}\sqrt{\log(k)}}{\tilde{\sigma}} \tag{465}$$

$$\leq O(\eta^{(1)}/\gamma)\frac{\tilde{\sigma}}{\sqrt{k}}. \tag{466}$$

$$\square$$

We also have:

$$|T_a| = O(\sigma_w \sqrt{\log(Dm/\delta)}) \tag{467}$$

$$\left| b_i^{(1)} \right| \leq \left| b_i^{(0)} \right| + \left| \eta^{(1)} a_i^{(0)} T_b \right| \tag{468}$$

$$\leq \frac{\sqrt{\log(m/\delta)}}{k^2} + \left| \eta^{(1)} a_i^{(0)} \epsilon_e \right|. \tag{469}$$

$$\mathbb{E}_{\xi^{(2)}}\left| \xi_i^{(2)} \right| \leq O(\sigma_\xi^{(2)}). \tag{470}$$

$$|a_{m+i}^{(1)}| \leq |a_i^{(0)}| + |\eta^{(1)} T_a| \leq |a_i^{(0)}| + O(\eta^{(1)} \sigma_w \sqrt{\log(Dm/\delta)}). \tag{471}$$

$$\left| \langle w_i^{(1)} - w_{m+i}^{(1)}, x \rangle \right| = 2\left| \langle w_i^{(1)}, x \rangle \right| = O(\eta^{(1)} \tilde{\sigma}/\gamma). \tag{472}$$

$$\left| b_i^{(1)} - b_{m+i}^{(1)} \right| = 2|\eta^{(1)} a_i^{(0)} T_b| = O(|\eta^{(1)} a_i^{(0)}|\epsilon_e). \tag{473}$$

Then we have

$$\left| g^{(1)}(x; \sigma_\xi^{(2)}) \right| \leq O\left( m\eta^{(1)}\sigma_w \sqrt{\log(Dm/\delta)} \right) \left( \frac{\eta^{(1)}\tilde{\sigma}}{\gamma} + \frac{\sqrt{\log(m/\delta)}}{k^2} + \left| \eta^{(1)}a_i^{(0)}\epsilon_e \right| + \sigma_\xi^{(2)} \right) \tag{474}$$

$$+ O\left( m(|a_i^{(0)}| + \eta^{(1)}\sigma_w\sqrt{\log(Dm/\delta)}) \right) \left( \frac{\eta^{(1)}\tilde{\sigma}}{\gamma} + \left| \eta^{(1)}a_i^{(0)}\epsilon_e \right| \right) \tag{475}$$

$$= O\left( m\eta^{(1)}\sigma_w \frac{\log(Dm/\delta)}{k} + m|a_i^{(0)}| \left( \frac{\eta^{(1)}\tilde{\sigma}}{\gamma} + \left| \eta^{(1)}a_i^{(0)}\epsilon_e \right| \right) \right) \tag{476}$$

$$= O\left( m\eta^{(1)}\sigma_w \frac{\log(Dm/\delta)}{k} + m|a_i^{(0)}|\frac{\eta^{(1)}\tilde{\sigma}}{\gamma} + m|a_i^{(0)}|\frac{\eta^{(1)}\tilde{\sigma}}{\gamma\sqrt{k}} \right) \tag{477}$$

$$< 1. \tag{478}$$

Then $\left| yg^{(1)}(x; \sigma_\xi^{(2)}) \right| < 1$. Finally, the statement on $\left| \langle (q_i^{(1)})_{-A}, \phi_{-A} \rangle \right|$ follows from a similar calculation on $\left| \langle w_i^{(1)}, x \rangle \right| = \left| \langle q_i^{(1)}, \phi \rangle \right|$. $\qquad\square$

We are now ready to analyze the gradients in the second gradient step.

**Lemma 45.** *Fix $\delta \in (0,1)$ and suppose $w_i^{(0)} \in \mathcal{G}_w(\delta), b_i^{(0)} \in \mathcal{G}_b(\delta), a_i^{(0)} \in \mathcal{G}_a(\delta)$ for all $i \in [2m]$. Let $\epsilon_{e2} := O\left( \frac{\eta^{(1)}|a_i^{(0)}|k(\gamma + \epsilon_e)}{\tilde{\sigma}^2\sigma_\xi^{(2)}} \right) + \exp(-\Theta(k))$. If $D\mu/\sqrt{d} \leq 1/16$, $\sigma_\zeta\tilde{\sigma} = O(1)$, $\sigma_\zeta^2 d/\tilde{\sigma}^2 = O(1)$, $k = \Omega(\log^2(Dm/\delta))$, $\sigma_a \leq \tilde{\sigma}^2/(\gamma k^2)$, $\eta^{(1)} = O\left( \frac{\gamma}{km\sigma_a\tilde{\sigma}} \right)$, and $\sigma_\xi^{(2)} = 1/k^{3/2}$, then*

$$\frac{\partial}{\partial w_i}L_\mathcal{D}(g^{(1)}; \sigma_\xi^{(2)}) = -a_i^{(1)}\left( \sum_{j=1}^D M_j T_j + \nu \right) \tag{479}$$

*where $T_j$ satisfies:*

- *if $j \in A$, then $|T_j - \beta\gamma/\tilde{\sigma}| \leq O(\epsilon_{e2}/\tilde{\sigma} + \eta^{(1)}/\sigma_\xi^{(2)} + \eta^{(1)}|a_i^{(0)}|\epsilon_e/(\tilde{\sigma}\sigma_\xi^{(2)}))$, where $\beta \in [\Omega(1), 1]$ and depends only on $w_i^{(0)}, b_i^{(0)}$;*

- *if $j \notin A$, then $|T_j| \leq \frac{1}{\tilde{\sigma}}\exp(-\Omega(k)) + O(\sigma_\phi^2\tilde{\sigma}\epsilon_{e2})$;*

- *$|\nu_j| \leq O\left( \frac{\eta^{(1)}\sigma_\zeta}{\gamma\sigma_\xi^{(2)}} \right) + \exp(-\Omega(k))$.*

*Proof of Lemma 45.* Consider one neuron index $i$ and omit the subscript $i$ in the parameters. By Lemma 44, with probability at least $1 - (d + D)\exp(-\Omega(k)) = 1 - \exp(-\Omega(k))$ over $(x, y)$, $yg^{(1)}(x; \xi^{(2)}) > 1$ and furthermore, for any $i \in [2m]$, $\left| \langle w_i^{(1)}, \zeta/\tilde{\sigma} \rangle \right| = O(\eta^{(1)}\tilde{\sigma}/\gamma)$, $\left| \langle q_i^{(1)}, \phi \rangle \right| = O(\eta^{(1)}\tilde{\sigma}/\gamma)$, and $\left| \langle (q_i^{(1)})_{-A}, \phi_{-A} \rangle \right| = O(\eta^{(1)}\tilde{\sigma}/\gamma)$, and for any $j \in [d], \ell \in [D]$, $|\zeta_j| \leq O(\sigma_\zeta\sqrt{\log(k)})$ and $|\langle \zeta, D_\ell \rangle| \leq O(\sigma_\zeta\sqrt{\log(k)})$. Let $\mathbb{I}_x$ be the indicator of this event.

$$\frac{\partial}{\partial w}L_\mathcal{D}^\alpha(g^{(1)}; \sigma_\xi^{(2)}) \tag{480}$$

$$= -a^{(1)}\mathbb{E}_{(x,y)\sim\mathcal{D}}\left\{ \alpha_y y\mathbb{I}_x\mathbb{E}_{\xi^{(2)}}\mathbb{I}[\langle w^{(1)}, x \rangle + b^{(1)} + \xi^{(2)} \in (0,1)]x \right\} \tag{481}$$

$$= -a^{(1)}\sum_{j=1}^D M_j \underbrace{\mathbb{E}_{(x,y)\sim\mathcal{D},\xi^{(2)}}\left\{ \alpha_y y\mathbb{I}_x\mathbb{I}[\langle w^{(1)}, x \rangle + b^{(1)} + \xi^{(2)} \in (0,1)]\phi_j \right\}}_{:=T_j} \tag{482}$$

$$- a^{(1)}\underbrace{\mathbb{E}_{(x,y)\sim\mathcal{D},\xi^{(2)}}\left\{ \frac{\alpha_y y\zeta}{\tilde{\sigma}}\mathbb{I}_x\mathbb{I}[\langle w^{(1)}, x \rangle + b^{(1)} + \xi^{(2)} \in (0,1)] \right\}}_{:=\nu}. \tag{483}$$

Let $T_{j1} := \mathbb{E}_{(x,y)\sim\mathcal{D},\xi^{(2)}} \left\{ \alpha_y y \mathbb{I}[\langle w^{(1)}, x \rangle + b^{(1)} + \xi^{(2)} \in (0,1)]\phi_j \right\}$. We have

$$|T_j - T_{j1}| \tag{484}$$

$$= \left| \mathbb{E}_{(x,y)\sim\mathcal{D},\xi^{(2)}} \left\{ \alpha_y y (1 - \mathbb{I}_x) \mathbb{I}[\langle w^{(1)}, x \rangle + b^{(1)} + \xi^{(2)} \in (0,1)]\phi_j \right\} \right| \tag{485}$$

$$\leq \frac{1}{\tilde{\sigma}} \exp(-\Omega(k)). \tag{486}$$

Similarly, let $\nu' := \mathbb{E}_{(x,y)\sim\mathcal{D},\xi^{(2)}} \left\{ \frac{\alpha_y y \zeta}{\tilde{\sigma}} \mathbb{I}[\langle w^{(1)}, x \rangle + b^{(1)} + \xi^{(2)} \in (0,1)] \right\}$. We have

$$|\nu - \nu'| \tag{487}$$

$$= \left| \mathbb{E}_{(x,y)\sim\mathcal{D},\xi^{(2)}} \left\{ \frac{\alpha_y y \zeta}{\tilde{\sigma}} (1 - \mathbb{I}_x) \mathbb{I}[\langle w^{(1)}, x \rangle + b^{(1)} + \xi^{(2)} \in (0,1)] \right\} \right| \tag{488}$$

$$\leq \frac{\sigma_\zeta}{\tilde{\sigma}} \exp(-\Omega(k)). \tag{489}$$

So it is sufficient to bound $T_{j1}$ and $\nu'$. For simplicity, we use $q$ as a shorthand for $q_i^{(1)}$.

First, consider $j \in A$.

$$T_{j1} = \mathbb{E}_{(x,y)\sim\mathcal{D},\xi^{(2)}} \left\{ \alpha_y y \mathbb{I}[\langle w^{(1)}, x \rangle + b^{(1)} + \xi^{(2)} \in (0,1)]\phi_j \right\} \tag{490}$$

$$= \mathbb{E}_{\phi_A} \left\{ \alpha_y y \phi_j \Pr_{\phi_{-A},\xi^{(2)}} \left[ \langle \phi, q \rangle + \iota + b^{(1)} + \xi^{(2)} \in (0,1) \right] \right\} \tag{491}$$

where $\iota := \langle w^{(1)}, \zeta/\tilde{\sigma} \rangle$. Let

$$I_a := \Pr_{\xi^{(2)}} \left[ \langle \phi, q \rangle + \iota + b^{(1)} + \xi^{(2)} \in (0,1) \right], \tag{492}$$

$$I_a' := \Pr_{\xi^{(2)}} \left[ \langle \phi_{-A}, q_{-A} \rangle + \iota + b^{(1)} + \xi^{(2)} \in (0,1) \right]. \tag{493}$$

By the property of the Gaussian $\xi^{(2)}$, that $|\langle \phi_A, q_A \rangle| = O(\frac{\eta^{(1)} |a_i^{(0)}| k(\gamma+\epsilon_e)}{\tilde{\sigma}^2})$, and that $|\iota| = |\langle w_i^{(1)}, \zeta/\tilde{\sigma} \rangle|, |\langle \phi, q \rangle|, |\langle \phi_{-A}, q_{-A} \rangle|$ are all $O(\eta^{(1)}\tilde{\sigma}/\gamma) < O(1/k)$, we have

$$|I_a - I_a'| \tag{494}$$

$$\leq \left| \Pr_{\xi^{(2)}} \left[ \langle \phi, q \rangle + \iota + b^{(1)} + \xi^{(2)} \geq 0 \right] - \Pr_{\xi^{(2)}} \left[ \langle \phi_{-A}, q_{-A} \rangle + \iota + b^{(1)} + \xi^{(2)} \geq 0 \right] \right| \tag{495}$$

$$+ \Pr_{\xi^{(2)}} \left[ \langle \phi, q \rangle + \iota + b^{(1)} + \xi^{(2)} \geq 1 \right] + \Pr_{\xi^{(2)}} \left[ \langle \phi_{-A}, q_{-A} \rangle + \iota + b^{(1)} + \xi^{(2)} \geq 1 \right] \tag{496}$$

$$= O\left( \frac{\eta^{(1)} |a_i^{(0)}| k(\gamma + \epsilon_e)}{\tilde{\sigma}^2 \sigma_{\xi}^{(2)}} \right) + \exp(-\Theta(k)) = O(\epsilon_{e2}). \tag{497}$$

This leads to

$$\left| T_{j1} - \mathbb{E}_{\phi_A,\phi_{-A}} \left\{ \alpha_y y \phi_j I_a' \right\} \right| \tag{498}$$

$$\leq \mathbb{E}_{\phi_A} \left\{ |\alpha_y y \phi_j| \left| \mathbb{E}_{\phi_{-A}} (I_a - I_a') \right| \right\} \tag{499}$$

$$\leq O(\epsilon_{e2}) \mathbb{E}_{\phi_A} \left\{ |\alpha_y y \phi_j| \right\} \tag{500}$$

$$\leq O(\epsilon_{e2}/\tilde{\sigma}) \tag{501}$$

where the last step is from Lemma 39. Furthermore,

$$\mathbb{E}_{\phi_A,\phi_{-A}} \left\{ \alpha_y y \phi_j I_a' \right\} \tag{502}$$

$$= \mathbb{E}_{\phi_A} \left\{ \alpha_y y \phi_j \right\} \mathbb{E}_{\phi_{-A}} [I_a'] \tag{503}$$

$$= \mathbb{E}_{\phi_A} \left\{ \alpha_y y \phi_j \right\} \Pr_{\phi_{-A},\xi^{(2)}} \left[ \langle \phi_{-A}, q_{-A} \rangle + \iota + b^{(1)} + \xi^{(2)} \in (0,1) \right]. \tag{504}$$

By Lemma 19, we have $|\langle \phi_{-A}, q_{-A} \rangle + \iota| \leq O(\eta^{(1)} \tilde{\sigma} / \gamma)$. Also, $|b^{(1)} - b^{(0)}| \leq O(\eta^{(1)} |a_i^{(0)}| \epsilon_e)$. By the property of $\xi^{(2)}$,

$$\left| \Pr_{\xi^{(2)}} \left[ \langle \phi_{-A}, q_{-A} \rangle + \iota + b^{(1)} + \xi^{(2)} \in (0, 1) \right] - \Pr_{\xi^{(2)}} \left[ b^{(0)} + \xi^{(2)} \in (0, 1) \right] \right| \tag{505}$$

$$\leq O(\eta^{(1)} \tilde{\sigma} / (\gamma \sigma_\xi^{(2)})) + O(\eta^{(1)} |a_i^{(0)}| \epsilon_e / \sigma_\xi^{(2)}). \tag{506}$$

On the other hand,

$$\beta := \Pr_{\phi_{-A}, \xi^{(2)}} \left[ b^{(0)} + \xi^{(2)} \in (0, 1) \right] = \Pr_{\xi^{(2)}} \left[ \xi^{(2)} \in (-b^{(0)}, 1 - b^{(0)}) \right] \tag{507}$$

$$= \Omega(1) \tag{508}$$

and $\beta$ only depends on $b^{(0)}$. By Lemma 39, $\mathbb{E}_{\phi_A} \{\alpha_y y \phi_j\} = \gamma / \tilde{\sigma}$. Therefore,

$$|T_{j1} - \beta \gamma / \tilde{\sigma}| \leq O(\epsilon_{e2} / \tilde{\sigma}) + O(\eta^{(1)} / \sigma_\xi^{(2)}) + O(\eta^{(1)} |a_i^{(0)}| \epsilon_e / (\tilde{\sigma} \sigma_\xi^{(2)})). \tag{509}$$

Now, consider $j \notin A$. Let $B$ denote $A \cup \{j\}$.

$$T_{j1} = \mathbb{E}_{(x,y) \sim \mathcal{D}, \xi^{(2)}} \left\{ \alpha_y y \phi_j \mathbb{I} \left[ \langle \phi, q \rangle + \iota + b^{(1)} + \xi^{(2)} \in (0, 1) \right] \right\} \tag{510}$$

$$= \mathbb{E}_{\phi_B} \mathbb{E}_{\phi_{-B}, \zeta, \xi^{(2)}} \left\{ \alpha_y y \phi_j \mathbb{I} \left[ \langle \phi, q \rangle + \iota + b^{(1)} + \xi^{(2)} \in (0, 1) \right] \right\} \tag{511}$$

$$= \mathbb{E}_{\phi_B} \left\{ \alpha_y y \phi_j \Pr_{\phi_{-B}, \zeta, \xi^{(2)}} \left[ \langle \phi, q \rangle + \iota + b^{(1)} + \xi^{(2)} \in (0, 1) \right] \right\}. \tag{512}$$

Let

$$I_b := \Pr_{\xi^{(2)}} \left[ \langle \phi, q \rangle + \iota + b^{(1)} + \xi^{(2)} \in (0, 1) \right], \tag{513}$$

$$I_b' := \Pr_{\xi^{(2)}} \left[ \langle \phi_{-B}, q_{-B} \rangle + \iota + b^{(1)} + \xi^{(2)} \in (0, 1) \right]. \tag{514}$$

Similar as above, we have $|I_b - I_b'| \leq \epsilon_{e2}$. Then by Lemma 39,

$$\left| T_{j1} - \mathbb{E}_{\phi_B, \phi_{-B}, \zeta} \{\alpha_y y \phi_j I_b'\} \right| \tag{515}$$

$$\leq \mathbb{E}_{\phi_B} \left\{ |\alpha_y y \phi_j| |\mathbb{E}_{\phi_{-B}, \zeta} (I_b - I_b')| \right\} \tag{516}$$

$$\leq O(\epsilon_{e2}) \mathbb{E}_{\phi_A} \{|\alpha_y y|\} \mathbb{E}_{\phi_j} \{|\phi_j|\} \tag{517}$$

$$\leq O(\epsilon_{e2}) \times O(\sigma_\phi^2 \tilde{\sigma}) \tag{518}$$

$$= O(\sigma_\phi^2 \tilde{\sigma} \epsilon_{e2}). \tag{519}$$

Furthermore,

$$\mathbb{E}_{\phi_B, \phi_{-B}, \zeta} \{\alpha_y y \phi_j I_b'\} = \mathbb{E}_{\phi_A} \{\alpha_y y\} \mathbb{E}_{\phi_j} \{\phi_j\} \mathbb{E}_{\phi_{-B}} [I_b'] = 0. \tag{520}$$

Therefore,

$$|T_{j1}| \leq O(\sigma_\phi^2 \tilde{\sigma} \epsilon_{e2}). \tag{521}$$

Finally, consider $\nu_j'$.

$$\nu_j' = \mathbb{E}_{(x,y) \sim \mathcal{D}, \xi^{(2)}} \left\{ \frac{\alpha_y y \zeta_j}{\tilde{\sigma}} \mathbb{I}[\langle w^{(1)}, x \rangle + b^{(1)} + \xi^{(2)} \in (0, 1)] \right\} \tag{522}$$

$$= \mathbb{E}_{\phi_A, \phi_{-A}, \zeta, \xi^{(2)}} \left\{ \frac{\alpha_y y \zeta_j}{\tilde{\sigma}} \mathbb{I}[\langle \phi, q \rangle + \iota + b^{(1)} + \xi^{(2)} \in (0, 1)] \right\} \tag{523}$$

$$= \mathbb{E}_{\phi_A, \phi_{-A}, \zeta} \left\{ \frac{\alpha_y y \zeta_j}{\tilde{\sigma}} \Pr_{\xi^{(2)}} [\langle \phi, q \rangle + \iota + b^{(1)} + \xi^{(2)} \in (0, 1)] \right\} \tag{524}$$

Let

$$I_j := \Pr_{\xi^{(2)}} \left[ \langle \phi, q \rangle + \iota + b^{(0)} + \xi^{(1)} \in (0,1) \right], \tag{525}$$

$$I'_j := \Pr_{\xi^{(2)}} \left[ \langle \phi, q \rangle + b^{(0)} + \xi^{(1)} \in (0,1) \right]. \tag{526}$$

Since $|\iota| \leq O(\eta^{(1)} \tilde{\sigma} / \gamma)$, we have $|I_j - I'_j| \leq O(\eta^{(1)} \tilde{\sigma} / (\gamma \sigma_\xi^{(2)}))$. Then

$$\left| \nu'_j - \mathbb{E}_{\phi_A, \phi_{-A}, \zeta} \left\{ \frac{\alpha_y y \zeta_j}{\tilde{\sigma}} I'_j \right\} \right| \tag{527}$$

$$= \left| \mathbb{E}_{\phi_A, \phi_{-A}, \zeta} \left\{ \frac{\alpha_y y \zeta_j}{\tilde{\sigma}} (I_j - I'_j) \right\} \right| \tag{528}$$

$$\leq O(\eta^{(1)} \tilde{\sigma} / (\gamma \sigma_\xi^{(2)})) \mathbb{E}_{\phi_A, \phi_{-A}, \zeta} \left| \frac{\alpha_y y \zeta_j}{\tilde{\sigma}} \right| \tag{529}$$

$$\leq O(\eta^{(1)} \tilde{\sigma} / (\gamma \sigma_\xi^{(2)})) \mathbb{E}_{\phi_A} |\alpha_y y| \, \mathbb{E}_\zeta \left| \frac{\zeta_j}{\tilde{\sigma}} \right| \tag{530}$$

$$\leq O(\eta^{(1)} \tilde{\sigma} / (\gamma \sigma_\xi^{(2)})) \times 1 \times \frac{\sigma_\zeta}{\tilde{\sigma}} \tag{531}$$

$$\leq O(\eta^{(1)} \sigma_\zeta / (\gamma \sigma_\xi^{(2)})). \tag{532}$$

Furthermore,

$$\mathbb{E}_{\phi_A, \phi_{-A}, \zeta} \left\{ \frac{\alpha_y y \zeta_j}{\tilde{\sigma}} I'_j \right\} = \mathbb{E}_{\phi_A, \phi_{-A}} \left\{ \frac{\alpha_y y}{\tilde{\sigma}} I'_j \right\} \mathbb{E}_\zeta \left\{ \zeta_j \right\} = 0. \tag{533}$$

Therefore,

$$|\nu_j| \leq O \left( \frac{\eta^{(1)} \sigma_\zeta}{\gamma \sigma_\xi^{(2)}} \right) + \exp(-\Omega(k)). \tag{534}$$

$$\square$$

**Lemma 46.** *Under the same assumptions as in Lemma 45,*

$$\frac{\partial}{\partial b} L_{\mathcal{D}}(g^{(1)}; \sigma_\xi^{(2)}) = -a_i^{(1)} T_b \tag{535}$$

*where* $|T_b| \leq \exp(-\Omega(k)) + O(\epsilon_{e2})$.

*Proof of Lemma 46.* Consider one neuron index $i$ and omit the subscript $i$ in the parameters. By Lemma 44, $\Pr[y g^{(1)}(x; \xi^{(2)}) > 1] \leq \exp(-\Omega(k))$. Let $\mathbb{I}_x = \mathbb{I}[y g^{(1)}(x; \xi^{(2)}) \leq 1]$.

$$\frac{\partial}{\partial b} L_{\mathcal{D}}^\alpha(g^{(1)}; \sigma_\xi^{(2)}) \tag{536}$$

$$= -a^{(1)} \underbrace{\mathbb{E}_{(x,y) \sim \mathcal{D}} \left\{ \alpha_y y \mathbb{I}_x \mathbb{E}_{\xi^{(2)}} \mathbb{I}[\langle w^{(1)}, x \rangle + b^{(1)} + \xi^{(2)} \in (0,1)] \right\}}_{:= T_b}. \tag{537}$$

Let $T_{b1} := \mathbb{E}_{(x,y) \sim \mathcal{D}, \xi^{(2)}} \left\{ \alpha_y y \mathbb{I}[\langle w^{(1)}, x \rangle + b^{(1)} + \xi^{(2)} \in (0,1)] \right\}$. We have

$$|T_b - T_{b1}| \tag{538}$$

$$= \left| \mathbb{E}_{(x,y) \sim \mathcal{D}, \xi^{(2)}} \left\{ \alpha_y y (1 - \mathbb{I}_x) \mathbb{I}[\langle w^{(1)}, x \rangle + b^{(1)} + \xi^{(2)} \in (0,1)] \right\} \right| \tag{539}$$

$$\leq \exp(-\Omega(k)). \tag{540}$$

So it is sufficient to bound $T_{b1}$. For simplicity, we use $q$ as a shorthand for $q_i^{(1)}$.

$$T_{b1} = \mathbb{E}_{(x,y) \sim \mathcal{D}, \xi^{(2)}} \left\{ \alpha_y y \mathbb{I} \left[ \langle \phi, q \rangle + \iota + b^{(1)} + \xi^{(2)} \in (0,1) \right] \right\} \tag{541}$$

$$= \mathbb{E}_{\phi_A} \mathbb{E}_{\phi_{-A}, \xi^{(2)}} \left\{ \alpha_y y \mathbb{I} \left[ \langle \phi, q \rangle + \iota + b^{(1)} + \xi^{(2)} \in (0,1) \right] \right\} \tag{542}$$

$$= \mathbb{E}_{\phi_A} \left\{ \alpha_y y \Pr_{\phi_{-A}, \xi^{(2)}} \left[ \langle \phi, q \rangle + \iota + b^{(1)} + \xi^{(2)} \in (0,1) \right] \right\}. \tag{543}$$

Let

$$I_b := \Pr_{\xi^{(2)}} \left[ \langle \phi, q \rangle + \iota + b^{(1)} + \xi^{(2)} \in (0,1) \right],$$ (544)

$$I_b' := \Pr_{\xi^{(2)}} \left[ \langle \phi_{-A}, q_{-A} \rangle + \iota + b^{(1)} + \xi^{(2)} \in (0,1) \right].$$ (545)

Similar as in Lemma 20, we have $|I_b - I_b'| \le \epsilon_{e2}$. Then by Lemma 39,

$$\left| T_{b1} - \mathbb{E}_{\phi_A, \phi_{-A}} \left\{ \alpha_y y I_b' \right\} \right|$$ (546)

$$= \left| \mathbb{E}_{\phi_A, \phi_{-A}} \left\{ \alpha_y y (I_b - I_b') \right\} \right|$$ (547)

$$= O(\epsilon_{e2}) \mathbb{E}_{\phi_A, \phi_{-A}} |\alpha_y y|$$ (548)

$$\le O(\epsilon_{e2}).$$ (549)

Furthermore,

$$\mathbb{E}_{\phi_A, \phi_{-A}} \left\{ \alpha_y y I_b' \right\} = \mathbb{E}_{\phi_A} \left\{ \alpha_y y \right\} \mathbb{E}_{\phi_{-A}} [I_b'] = 0.$$ (550)

Therefore, $|T_{b1}| \le O(\epsilon_{e2})$ and the statement follows. $\qquad\square$

**Lemma 47.** *Under the same assumptions as in Lemma 45,*

$$\frac{\partial}{\partial a_i} L_{\mathcal{D}}(g^{(1)}; \sigma_\xi^{(2)}) = -T_a$$ (551)

*where* $|T_a| = O\left( \frac{\eta^{(1)} \tilde{\sigma}}{\gamma p_{\min}} \right) + \exp(-\Omega(k)) \mathrm{poly}\left( \frac{dD}{p_{\min}} \right).$

*Proof of Lemma 47.* Consider one neuron index $i$ and omit the subscript $i$ in the parameters. By Lemma 44, $\Pr[yg^{(1)}(x; \xi^{(2)}) > 1] \le \exp(-\Theta(k))$. Let $\mathbb{I}_x = \mathbb{I}[yg^{(1)}(x; \xi^{(2)}) \le 1]$.

$$\frac{\partial}{\partial a} L_{\mathcal{D}}^\alpha(g^{(1)}; \sigma_\xi^{(2)})$$ (552)

$$= -\underbrace{\mathbb{E}_{(x,y) \sim \mathcal{D}} \left\{ \alpha_y y \mathbb{I}_x \mathbb{E}_{\xi^{(2)}} \sigma(\langle w^{(1)}, x \rangle + b^{(1)} + \xi^{(2)}) \right\}}_{:= T_a}.$$ (553)

Let $T_{a1} := \mathbb{E}_{(x,y) \sim \mathcal{D}} \left\{ \alpha_y y \mathbb{E}_{\xi^{(2)}} \sigma(\langle w^{(1)}, x \rangle + b^{(1)} + \xi^{(2)}) \right\}$. We have

$$|T_a - T_{a1}|$$ (554)

$$= \left| \mathbb{E}_{(x,y) \sim \mathcal{D}} \left\{ \alpha_y y (1 - \mathbb{I}_x) \mathbb{E}_{\xi^{(2)}} \sigma(\langle w^{(1)}, x \rangle + b^{(1)} + \xi^{(2)}) \right\} \right|$$ (555)

$$\le \exp(-\Omega(k)).$$ (556)

So it is sufficient to bound $T_{a1}$. For simplicity, we use $q$ as a shorthand for $q_i^{(1)}$.

Let $\phi'_A$ be an independent copy of $\phi_A$, $\phi'$ be the vector obtained by replacing in $\phi$ the entries $\phi_A$ with $\phi'_A$, and let $x' = M\phi' + \zeta$ and its label is $y'$. Then

$$|T_{a1}| := \left| \mathbb{E}_{\phi_A} \left\{ \alpha_y y \mathbb{E}_{\phi_{-A}, \zeta, \xi^{(2)}} \sigma(\langle w^{(1)}, x \rangle + b^{(1)} + \xi^{(2)}) \right\} \right| \tag{557}$$

$$\leq \frac{1}{2} \left| \mathbb{E}_{\phi_A} \left\{ \mathbb{E}_{\phi_{-A}, \zeta, \xi^{(2)}} \sigma(\langle w^{(1)}, x \rangle + b^{(1)} + \xi^{(1)}) | y = 1 \right\} \right. \tag{558}$$

$$\left. - \mathbb{E}_{\phi_A} \left\{ \mathbb{E}_{\phi_{-A}, \zeta, \xi^{(2)}} \sigma(\langle w^{(1)}, x \rangle + b^{(1)} + \xi^{(2)}) | y = -1 \right\} \right| \tag{559}$$

$$\leq \frac{1}{2} \left| \mathbb{E}_{\phi_A} \left\{ \mathbb{E}_{\phi_{-A}, \zeta, \xi^{(2)}} \sigma(\langle w^{(1)}, x \rangle + b^{(1)} + \xi^{(2)}) | y = 1 \right\} \right. \tag{560}$$

$$\left. - \mathbb{E}_{\phi'_A} \left\{ \mathbb{E}_{\phi_{-A}, \zeta, \xi^{(2)}} \sigma(\langle w^{(1)}, x' \rangle + b^{(1)} + \xi^{(2)}) | y' = -1 \right\} \right| \tag{561}$$

$$\leq \frac{1}{2} \mathbb{E}_{\phi_A, \phi'_A} \left\{ \mathbb{E}_{\phi_{-A}} \left| \langle w^{(1)}, x \rangle - \langle w^{(1)}, x' \rangle \right| | y = 1, y' = -1 \right\} \tag{562}$$

$$\leq \frac{1}{2} \mathbb{E}_{\phi_{-A}} \left( \mathbb{E}_{\phi_A} \left\{ \left| \langle w^{(1)}, M\phi \rangle \right| | y = 1 \right\} + \mathbb{E}_{\phi'_A} \left\{ \left| \langle w^{(1)}, M\phi' \rangle \right| | y' = -1 \right\} \right) \tag{563}$$

$$\leq \mathbb{E}_{\phi_{-A}, \phi_A} \left| \alpha_y \langle w^{(1)}, M\phi \rangle \right| \tag{564}$$

$$= \mathbb{E}_\phi \left| \alpha_y \langle w^{(1)}, M\phi \rangle \right| \tag{565}$$

$$= O(\eta^{(1)} \tilde{\sigma}/\gamma) + \exp(-\Omega(k)) \times \frac{\sqrt{dD}}{\tilde{\sigma}} \times \|w^{(1)}\|_\infty \tag{566}$$

$$= O\left( \frac{\eta^{(1)} \tilde{\sigma}}{\gamma p_{\min}} \right) + \exp(-\Omega(k)) \text{poly}(dD/p_{\min}) \tag{567}$$

where the fourth step follows from that $\sigma$ is 1-Lipschitz, and the second to the last line from Lemma 44 and that $\left| \langle w^{(1)}, M\phi \rangle \right| \leq \|w^{(1)}\|_\infty \sqrt{d} \|M\phi\|_2^2$. $\qquad \square$

With the above lemmas about the gradients, we are now ready to show that at the end of the second step, we get a good set of features for accurate prediction.

**Lemma 48.** *Set*

$$\eta^{(1)} = \frac{\gamma^2 p_{\min} \tilde{\sigma}}{km^3}, \lambda_a^{(1)} = 0, \lambda_w^{(1)} = 1/(2\eta^{(1)}), \sigma_\xi^{(1)} = 1/k^{3/2}, \tag{568}$$

$$\eta^{(2)} = 1, \lambda_a^{(2)} = \lambda_w^{(2)} = 1/(2\eta^{(2)}), \sigma_\xi^{(2)} = 1/k^{3/2}. \tag{569}$$

*Fix $\delta \in (0, O(1/k^3))$. If $D\mu/\sqrt{d} \leq 1/16$, $\sigma_\zeta \tilde{\sigma} = O(1)$, $\sigma_\zeta^2 d/\tilde{\sigma}^2 = O(1)$, $k = \Omega\left(\log^2\left(\frac{Dmd}{\delta \gamma p_{\min}}\right)\right)$, and $m \geq \max\{\Omega(k^4), D, d\}$, then with probability at least $1 - \delta$ over the initialization, there exist $\tilde{a}_i$'s such that $\tilde{g}(x) := \sum_{i=1}^{2m} \tilde{a}_i \sigma(\langle w_i^{(2)}, x \rangle + b_i^{(2)})$ satisfies $L_\mathcal{D}(\tilde{g}) \leq \exp(-\Omega(k))$. Furthermore, $\|\tilde{a}\|_0 = O(m/k)$, $\|\tilde{a}\|_\infty = O(k^5/m)$, and $\|\tilde{a}\|_2^2 = O(k^9/m)$. Finally, $\|a^{(2)}\|_\infty = O\left(\frac{1}{km^2}\right)$, $\|w_i^{(2)}\|_2 = O(\tilde{\sigma}/k)$, and $|b_i^{(2)}| = O(1/k^2)$ for all $i \in [2m]$.*

*Proof of Lemma 48.* By Lemma 35, there exists a network $g^*(x) = \sum_{\ell=1}^{3(k+1)} a_\ell^* \sigma(\langle w_\ell^*, x \rangle + b_\ell^*)$ satisfying

$$\Pr_{(x,y) \sim \mathcal{D}} [y g^*(x) \leq 1] \leq \exp(-\Omega(k)).$$

Furthermore, the number of neurons $n = 3(k+1)$, $|a_i^*| \leq 64k$, $1/(64k) \leq |b_i^*| \leq 1/4$, $w_i^* = \tilde{\sigma} \sum_{j \in A} M_j/(8k)$, and $|\langle w_i^*, x \rangle + b_i^*| \leq 1$ for any $i \in [n]$ and $(x, y) \sim \mathcal{D}$. Now we fix an $\ell$, and show that with high probability there is a neuron in $g^{(2)}$ that can approximate the $\ell$-th neuron in $g^*$.

With probability $\geq 1 - \exp(-\Omega(\max\{2p_o(D-k), k\}))$, among all $j \notin A$, we have that at most $2p_o(D-k) + k$ of $\phi_j$ are $(1-p_o)/\tilde{\sigma}$, while the others are $-p_o/\tilde{\sigma}$. With probability $\geq 1 - (d +$

$D$) $\exp(-\Omega(k))$ over $\zeta$, for any $j$, $|\zeta_j| \leq O(\sigma_\zeta \sqrt{\log(k)})$ and $|\langle \zeta, D_\ell \rangle| \leq O(\sigma_\zeta \sqrt{\log(k)})$. Below we consider data points with $\phi$ and $\zeta$ satisfying these.

By Lemma 37, with probability $1 - 2\delta$ over $w_i^{(0)}$'s, they are all in $\mathcal{G}_w(\delta)$; with probability $1 - \delta$ over $a_i^{(0)}$'s, they are all in $\mathcal{G}_a(\delta)$; with probability $1 - \delta$ over $b_i^{(0)}$'s, they are all in $\mathcal{G}_b(\delta)$. Under these events, by Lemma 43, Lemma 45 and 46, for any neuron $i \in [2m]$, we have

$$w_i^{(2)} = a_i^{(1)} \left( \sum_{j=1}^{D} M_j T_j + \nu \right), \tag{570}$$

$$b_i^{(2)} = b_i^{(1)} + a_i^{(1)} T_b. \tag{571}$$

where

- if $j \in A$, then $|T_j - \beta\gamma/\tilde{\sigma}| \leq \epsilon_{w1} := O(\epsilon_{e2}/\tilde{\sigma} + \eta^{(1)}/\sigma_\xi^{(2)} + \eta^{(1)}|a_i^{(0)}|\epsilon_e/(\tilde{\sigma}\sigma_\xi^{(2)}))$, where $\beta \in [\Omega(1), 1]$ and depends only on $w_i^{(0)}, b_i^{(0)}$;

- if $j \notin A$, then $|T_j| \leq \epsilon_{w2} := \frac{1}{\tilde{\sigma}} \exp(-\Omega(k)) + O(\sigma_\phi^2 \tilde{\sigma} \epsilon_{e2})$;

- $|\nu_j| \leq \epsilon_\nu := O\left( \frac{\eta^{(1)}\sigma_\zeta}{\gamma\sigma_\xi^{(2)}} \right) + \exp(-\Omega(k))$.

- $|T_b| \leq \epsilon_b := \exp(-\Omega(k)) + O(\epsilon_{e2})$.

Given the initialization, with probability $\Omega(1)$ over $b_i^{(0)}$, we have

$$|b_i^{(0)}| \in \left[ \frac{1}{2k^2}, \frac{2}{k^2} \right], \mathrm{sign}(b_i^{(0)}) = \mathrm{sign}(b_\ell^*). \tag{572}$$

Finally, since $\frac{8k|b_\ell^*|\beta\gamma}{|b_i^{(0)}|\tilde{\sigma}^2} \in [\Omega(k^2\gamma/\tilde{\sigma}^2), O(k^3\gamma/\tilde{\sigma}^2)]$ and depends only on $w_i^{(0)}, b_i^{(0)}$, we have that for $\epsilon_a = \Theta(1/k^2)$, with probability $\Omega(\epsilon_a) > \delta$ over $a_i^{(0)}$,

$$\left| \frac{8k|b_\ell^*|\beta\gamma}{|b_i^{(0)}|\tilde{\sigma}^2} a_i^{(0)} - 1 \right| \leq \epsilon_a, \quad |a_i^{(0)}| = O\left( \frac{\tilde{\sigma}^2}{k^2\gamma} \right). \tag{573}$$

Let $n_a = \epsilon_a m/4$. For the given value of $m$, by (570)-(573) we have with probability $\geq 1 - 5\delta$ over the initialization, for each $\ell$ there is a different set of neurons $I_\ell \subseteq [m]$ with $|I_\ell| = n_a$ and such that for each $i_\ell \in I_\ell$,

$$|b_{i_\ell}^{(0)}| \in \left[ \frac{1}{2k^2}, \frac{2}{k^2} \right], \quad \mathrm{sign}(b_{i_\ell}^{(0)}) = \mathrm{sign}(b_\ell^*), \tag{574}$$

$$\left| \frac{8k|b_\ell^*|\beta\gamma}{|b_{i_\ell}^{(0)}|\tilde{\sigma}^2} a_{i_\ell}^{(0)} - 1 \right| \leq \epsilon_a, \quad |a_{i_\ell}^{(0)}| = O\left( \frac{\tilde{\sigma}^2}{k^2\gamma} \right). \tag{575}$$

Now, construct $\tilde{a}$ such that $\tilde{a}_{i_\ell} = \frac{2a_\ell^*|b_\ell^*|}{|b_{i_\ell}^{(0)}|n_a}$ for each $\ell$ and each $i_\ell \in I_\ell$, and $\tilde{a}_i = 0$ elsewhere. To show that it gives accurate predictions, we first consider bounding some error terms.

For the given values of parameters, we have

$$\epsilon_{e2} = O\left( \frac{\gamma}{m^2} \right), \tag{576}$$

$$\epsilon_{w1} = O\left( \frac{k\gamma}{m^2\tilde{\sigma}} + \frac{\gamma\epsilon_e}{km^2} \right), \tag{577}$$

$$\epsilon_{w2} = O\left( \frac{\gamma}{m^2\tilde{\sigma}} \right), \tag{578}$$

$$\epsilon_\nu = O\left( \frac{\gamma k}{m^3} \right), \tag{579}$$

$$\epsilon_b = O\left( \frac{\gamma}{m^2} \right). \tag{580}$$

We also have the following useful claims.

**Claim 2.** $\sum_{\ell \in A} |\langle M_\ell, x \rangle| \leq O\left(\frac{k}{\tilde{\sigma}}\right)$.

*Proof of Claim 2.*

$$\sum_{\ell \in A} |\langle M_\ell, x \rangle| \tag{581}$$

$$\leq \sum_{\ell \in A} \left( |\phi_j| + \left| \sum_{j \neq \ell} M_\ell^\top M_j \phi_j \right| + |M_\ell^\top \zeta / \tilde{\sigma}| \right) \tag{582}$$

$$\leq O\left(\frac{k}{\tilde{\sigma}}\right) + O\left(kD\frac{\mu}{\sqrt{d}\tilde{\sigma}}\right) + O\left(k\frac{\sigma_\zeta \sqrt{\log(k)}}{\tilde{\sigma}}\right) \tag{583}$$

$$\leq O\left(\frac{k}{\tilde{\sigma}}\right). \tag{584}$$

$\square$

**Claim 3.**

$$\left| \langle w_{i_\ell}^{(2)}, x \rangle - \frac{a_{i_\ell}^{(0)} \beta \gamma}{\tilde{\sigma}} \sum_{j \in A} \langle M_j, x \rangle \right| \leq O\left(\frac{1}{m}\right). \tag{585}$$

*Proof of Claim 3.*

$$\left| \langle w_{i_\ell}^{(2)}, x \rangle \right| = \left| \langle \sum_{\ell=1}^{D} a_{i_\ell}^{(1)} T_\ell M_\ell + \upsilon, x \rangle \right| \tag{586}$$

$$\leq \left| \langle \sum_{\ell \in A} a_{i_\ell}^{(1)} T_\ell M_\ell, x \rangle \right| + \left| \langle \sum_{\ell \notin A} a_{i_\ell}^{(1)} T_\ell M_\ell, x \rangle \right| + |\langle \upsilon, x \rangle|. \tag{587}$$

Then

$$\left| \langle w_{i_\ell}^{(2)}, x \rangle - \frac{a_{i_\ell}^{(0)} \beta \gamma}{\tilde{\sigma}} \sum_{j \in A} \langle M_j, x \rangle \right| \tag{588}$$

$$\leq \left| \langle w_{i_\ell}^{(2)}, x \rangle - \frac{a_{i_\ell}^{(1)} \beta \gamma}{\tilde{\sigma}} \sum_{j \in A} \langle M_j, x \rangle \right| + \left| \frac{a_{i_\ell}^{(1)} \beta \gamma}{\tilde{\sigma}} \sum_{j \in A} \langle M_j, x \rangle - \frac{a_{i_\ell}^{(0)} \beta \gamma}{\tilde{\sigma}} \sum_{j \in A} \langle M_j, x \rangle \right| \tag{589}$$

$$\leq \left| \langle w_{i_\ell}^{(2)}, x \rangle - \frac{a_{i_\ell}^{(1)} \beta \gamma}{\tilde{\sigma}} \sum_{j \in A} \langle M_j, x \rangle \right| + \left| a_{i_\ell}^{(1)} - a_{i_\ell}^{(0)} \right| \left| \frac{\beta \gamma}{\tilde{\sigma}} \sum_{j \in A} \langle M_j, x \rangle \right|. \tag{590}$$

The first term is

$$\left| \langle w_{i_\ell}^{(2)}, x \rangle - \frac{a_{i_\ell}^{(1)} \beta \gamma}{\tilde{\sigma}} \sum_{j \in A} \langle M_j, x \rangle \right| \tag{591}$$

$$\leq \left| a_{i_\ell}^{(1)} \right| \left( \left| \langle \sum_{\ell \in A} \left( T_\ell - \frac{\beta \gamma}{\tilde{\sigma}} \right) M_\ell, x \rangle \right| + \left| \langle \sum_{\ell \notin A} T_\ell M_\ell, x \rangle \right| + |\langle \nu, x \rangle| \right). \tag{592}$$

By Claim 2,

$$\left| \left\langle \sum_{\ell \in A} \left( T_\ell - \frac{\beta\gamma}{\tilde{\sigma}} \right) M_\ell, x \right\rangle \right| \leq \sum_{\ell \in A} \left| T_\ell - \frac{\beta\gamma}{\tilde{\sigma}} \right| |\langle M_\ell, x \rangle| \tag{593}$$

$$\leq \epsilon_{w1} \sum_{\ell \in A} |\langle M_\ell, x \rangle| \tag{594}$$

$$\leq O\left( \frac{k\epsilon_{w1}}{\tilde{\sigma}} \right). \tag{595}$$

$$\left| \left\langle \sum_{\ell \notin A} T_\ell M_\ell, x \right\rangle \right| \leq \left| \sum_{\ell \notin A} T_\ell \phi_j \right| + \left| \sum_{\ell \notin A, j \neq \ell} T_\ell M_\ell^\top M_j \phi_j \right| + \left| \sum_{\ell \notin A} T_\ell M_\ell^\top \zeta / \tilde{\sigma} \right| \tag{596}$$

$$\leq O\left( \frac{D\epsilon_{w2}}{\tilde{\sigma}} \right) + O\left( D^2 \epsilon_{w2} \frac{\mu}{\sqrt{d}\tilde{\sigma}} \right) + O\left( D\epsilon_{w2} \frac{\sigma_\zeta \sqrt{\log(k)}}{\tilde{\sigma}} \right) \tag{597}$$

$$\leq O\left( \frac{D\epsilon_{w2}}{\tilde{\sigma}} \right). \tag{598}$$

$$|\langle \nu, x \rangle| \leq \left| \left\langle \nu, \sum_{\ell \in [D]} M_\ell \phi_\ell \right\rangle \right| + |\langle \nu, \zeta / \tilde{\sigma} \rangle| \tag{599}$$

$$\leq \sum_{\ell \in [D]} |\phi_\ell \langle \nu, M_\ell \rangle| + |\langle \nu, \zeta / \tilde{\sigma} \rangle| \tag{600}$$

$$\leq O\left( \frac{p_{\mathrm{o}} D + k}{\tilde{\sigma}} \right) \epsilon_\nu \sqrt{d} + d\epsilon_\nu \frac{O(\sigma_\zeta \sqrt{\log(k)})}{\tilde{\sigma}} \tag{601}$$

$$\leq O\left( \frac{\epsilon_\nu \sqrt{d}}{\tilde{\sigma}} \right) \left( p_{\mathrm{o}} D + \sqrt{d}\epsilon_\nu \sqrt{\log(k)} \right) \tag{602}$$

$$\leq O\left( \frac{\epsilon_\nu \sqrt{d}}{\tilde{\sigma}} \right) \left( p_{\mathrm{o}} D + \tilde{\sigma} \sqrt{\log(k)} \right) \tag{603}$$

$$\leq O\left( \frac{p_{\mathrm{o}} D \epsilon_\nu \sqrt{d}}{\tilde{\sigma}} \right). \tag{604}$$

Then by (576)-(580),

$$\epsilon_\phi := \left( \frac{k\epsilon_{w1}}{\tilde{\sigma}} + \frac{D\epsilon_{w2}}{\tilde{\sigma}} + \frac{p_{\mathrm{o}} D \epsilon_\nu \sqrt{d}}{\tilde{\sigma}} \right) = O\left( \frac{k^2 \gamma}{m^2 \tilde{\sigma}^2} + \frac{\gamma \epsilon_{\mathrm{e}}}{m^2 \tilde{\sigma}} + \frac{\gamma}{m\tilde{\sigma}^2} + \frac{\gamma k}{m^{3/2}\tilde{\sigma}} \right). \tag{605}$$

We have $\left| a_{i_\ell}^{(1)} - a_{i_\ell}^{(0)} \right| = O(\eta^{(1)} \sigma_w \sqrt{\log(Dm/\delta)})$. So the first term is bounded by

$$\left| \langle w_{i_\ell}^{(2)}, x \rangle - \frac{a_{i_\ell}^{(0)} \beta\gamma}{\tilde{\sigma}} \sum_{j \in A} \langle M_j, x \rangle \right| \leq \left| a_{i_\ell}^{(1)} \right| \epsilon_\phi \tag{606}$$

$$\leq O\left( \frac{\tilde{\sigma}^2}{k^2 \gamma} + \eta^{(1)} \sigma_w \sqrt{\log(Dm/\delta)} \right) \left( \frac{k^2 \gamma}{m^2 \tilde{\sigma}^2} + \frac{\gamma \epsilon_{\mathrm{e}}}{m^2 \tilde{\sigma}} + \frac{\gamma}{m\tilde{\sigma}^2} + \frac{\gamma k}{m^{3/2}\tilde{\sigma}} \right) \leq O\left( \frac{1}{m} \right). \tag{607}$$

By Claim 2, the second term is bounded by

$$\left| a_{i_\ell}^{(1)} - a_{i_\ell}^{(0)} \right| \left| \frac{\beta\gamma}{\tilde{\sigma}} \sum_{j \in A} \langle M_j, x \rangle \right| \leq O\left( \frac{k\eta^{(1)} \sigma_w \sqrt{\log(Dm/\delta)}\gamma}{\tilde{\sigma}^2} \right) \leq O\left( \frac{\gamma^3}{m^3} \right). \tag{608}$$

Combining the bounds on the two terms leads to the claim. $\qquad\square$

**Claim 4.**

$$|b_{i_\ell}^{(2)} - b_{i_\ell}^{(0)}| \leq O\left(\frac{1}{k^2 m}\right). \tag{609}$$

*Proof of Claim 4.* By Lemma 43 and 46:

$$|b_{i_\ell}^{(2)} - b_{i_\ell}^{(0)}| \leq |b_{i_\ell}^{(2)} - b_{i_\ell}^{(1)}| + |b_{i_\ell}^{(1)} - b_{i_\ell}^{(0)}| \tag{610}$$

$$\leq O\left(\eta^{(1)}|a_{i_\ell}^{(0)}|\epsilon_{\mathrm{e}} + |a_{i_\ell}^{(1)}|\left(\exp(-\Omega(k)) + \epsilon_{\mathrm{e2}}\right)\right) \tag{611}$$

$$\leq O\left(\frac{\gamma}{km^2} + \frac{1}{k^2 m}\right) \leq O\left(\frac{1}{k^2 m}\right). \tag{612}$$

$\square$

We are now ready to show $\tilde{g}$ is close to $2g^*$.

$$|\tilde{g}(x) - 2g^*(x)| \tag{613}$$

$$= \left|\sum_{\ell=1}^{3(k+1)} \sum_{i_\ell \in I_\ell} \tilde{a}_{i_\ell} \sigma\left(\langle w_{i_\ell}^{(2)}, x\rangle + b_{i_\ell}^{(2)}\right) - \sum_{\ell=1}^{3(k+1)} 2a_\ell^* \sigma\left(\langle w_\ell^*, x\rangle + b_\ell^*\right)\right| \tag{614}$$

$$= \left|\sum_{\ell=1}^{3(k+1)} \sum_{i_\ell \in I_\ell} \frac{2a_\ell^*|b_\ell^*|}{|b_{i_\ell}^{(0)}|n_a} \sigma\left(\langle w_{i_\ell}^{(2)}, x\rangle + b_{i_\ell}^{(2)}\right) - \sum_{\ell=1}^{3(k+1)} \sum_{i_\ell \in I_\ell} \frac{2a_\ell^*|b_\ell^*|}{|b_{i_\ell}^{(0)}|n_a} \sigma\left(\frac{|b_{i_\ell}^{(0)}|}{|b_\ell^*|}\langle w_\ell^*, x\rangle + b_{i_\ell}^{(0)}\right)\right| \tag{615}$$

$$\leq \left|\sum_{\ell=1}^{3(k+1)} \sum_{i_\ell \in I_\ell} \frac{1}{n_a}\left(\frac{2a_\ell^*|b_\ell^*|}{|b_{i_\ell}^{(0)}|} \sigma\left(\langle w_{i_\ell}^{(2)}, x\rangle + b_{i_\ell}^{(2)}\right) - \frac{2a_\ell^*|b_\ell^*|}{|b_{i_\ell}^{(0)}|} \sigma\left(\frac{a_{i_\ell}^{(0)}\beta\gamma}{\tilde{\sigma}}\sum_{j\in A}\langle M_j, x\rangle + b_{i_\ell}^{(0)}\right)\right)\right| \tag{616}$$

$$+ \left|\sum_{\ell=1}^{3(k+1)} \sum_{i_\ell \in I_\ell} \frac{1}{n_a}\left(\frac{2a_\ell^*|b_\ell^*|}{|b_{i_\ell}^{(0)}|} \sigma\left(\frac{a_{i_\ell}^{(0)}\beta\gamma}{\tilde{\sigma}}\sum_{j\in A}\langle M_j, x\rangle + b_{i_\ell}^{(0)}\right) - \frac{2a_\ell^*|b_\ell^*|}{|b_{i_\ell}^{(0)}|} \sigma\left(\frac{|b_{i_\ell}^{(0)}|}{|b_\ell^*|}\langle w_\ell^*, x\rangle + b_{i_\ell}^{(0)}\right)\right)\right|. \tag{617}$$

Here the second equation follows from that $\sigma$ is positive-homogeneous in $[0, 1]$, $|\langle w_\ell^*, x\rangle + b_\ell^*| \leq 1$, $|b_{i_\ell}^{(0)}|/|b_\ell^*| \leq 1$.

By Claim 3 and 4, the first term is bounded by:

$$3(k+1)\max_\ell \frac{2a_\ell^*|b_\ell^*|}{|b_{i_\ell}^{(0)}|}\left(\left|\langle w_{i_\ell}^{(2)}, x\rangle - \frac{a_{i_\ell}^{(0)}\beta\gamma}{\tilde{\sigma}}\sum_{j\in A}\langle M_j, x\rangle\right| + |b_{i_\ell}^{(2)} - b_{i_\ell}^{(0)}|\right) \tag{618}$$

$$\leq 3(k+1)\max_\ell \frac{2a_\ell^*|b_\ell^*|}{|b_{i_\ell}^{(0)}|} O\left(\frac{1}{m}\right) \tag{619}$$

$$\leq O\left(\frac{k^4}{m}\right). \tag{620}$$

By Claim 2, the second term is bounded by:

$$3(k+1)\max_\ell \frac{2a_\ell^*|b_\ell^*|}{|b_{i_\ell}^{(0)}|} \left| \frac{a_{i_\ell}^{(0)}\beta\gamma}{\tilde{\sigma}} \sum_{j\in A}\langle M_j, x\rangle - \frac{|b_{i_\ell}^{(0)}|}{|b_\ell^*|}\langle w_\ell^*, x\rangle \right| \tag{621}$$

$$\leq 3(k+1)\max_\ell \frac{2a_\ell^*|b_\ell^*|}{|b_{i_\ell}^{(0)}|} \left| \frac{a_{i_\ell}^{(0)}\beta\gamma}{\tilde{\sigma}} \sum_{j\in A}\langle M_j, x\rangle - \frac{|b_{i_\ell}^{(0)}|\tilde{\sigma}}{8k|b_\ell^*|} \sum_{j\in A}\langle M_j, x\rangle \right| \tag{622}$$

$$\leq 3(k+1)\max_\ell \frac{2a_\ell^*|b_\ell^*|}{|b_{i_\ell}^{(0)}|} \frac{\tilde{\sigma}|b_{i_\ell}^{(0)}|}{8k|b_\ell^*|} \left| \frac{8ka_{i_\ell}\beta\gamma|b_\ell^*|}{\tilde{\sigma}^2|b_{i_\ell}^{(0)}|} - 1 \right| \left| \sum_{j\in A}\langle M_j, x\rangle \right| \tag{623}$$

$$\leq 3(k+1)\max_\ell O(a_\ell^*\epsilon_a) \tag{624}$$

$$\leq O\left(k^2\epsilon_a\right). \tag{625}$$

Then

$$|\tilde{g}(x) - 2g^*(x)| = O\left(\frac{k^4}{m} + k^2\epsilon_a\right) \leq 1. \tag{626}$$

This guarantees $y\tilde{g}(x) \geq 1$. Changing the scaling of $\delta$ leads to the statement.

Finally, the bounds on $\tilde{a}$ follow from the above calculation. The bound on $\|a^{(2)}\|_2$ follows from Lemma 47, and those on $\|w_i^{(2)}\|_2$ and $\|b_i^{(2)}\|_2$ follow from (570)(571) and the bounds on $a_i^{(1)}$ and $b_i^{(1)}$ in Lemma 43. $\qquad\square$

### F.9 CLASSIFIER LEARNING STAGE AND MAIN THEOREM

Once we have a good set of features in Lemma 48, we can follow exactly the same argument as in Section B.6 and B.7 for the simplified setting, and arrive at the main theorem for the general setting:

**Theorem 49** (Restatement of Theorem 34). *Set*

$$\eta^{(1)} = \frac{\gamma^2 p_{\min}\tilde{\sigma}}{km^3}, \lambda_a^{(1)} = 0, \lambda_w^{(1)} = 1/(2\eta^{(1)}), \sigma_\xi^{(1)} = 1/k^{3/2}, \tag{627}$$

$$\eta^{(2)} = 1, \lambda_a^{(2)} = \lambda_w^{(2)} = 1/(2\eta^{(2)}), \sigma_\xi^{(2)} = 1/k^{3/2}, \tag{628}$$

$$\eta^{(t)} = \eta = \frac{k^2}{Tm^{1/3}}, \lambda_a^{(t)} = \lambda_w^{(t)} = \lambda \leq \frac{k^3}{\tilde{\sigma}m^{1/3}}, \sigma_\xi^{(t)} = 0, \text{ for } 2 < t \leq T. \tag{629}$$

*For any $\delta \in (0, O(1/k^3))$, if $\mu \leq O(\sqrt{d}/D)$, $\sigma_\zeta \leq O(\min\{1/\tilde{\sigma}, \tilde{\sigma}/\sqrt{d}\})$, $k = \Omega\left(\log^2\left(\frac{Dmd}{\delta\gamma p_{\min}}\right)\right)$, $m \geq \max\{\Omega(k^4), D, d\}$, then we have for any $\mathcal{D} \in \mathcal{F}_\Xi$, with probability at least $1 - \delta$, there exists $t \in [T]$ such that*

$$\Pr[\text{sign}(g^{(t)}(x)) \neq y] \leq L_\mathcal{D}(g^{(t)}) = O\left(\frac{k^8}{m^{2/3}} + \frac{k^3 T}{m^2} + \frac{k^2 m^{2/3}}{T}\right). \tag{630}$$

*Consequently, for any $\epsilon \in (0, 1)$, if $T = m^{4/3}$, and $m \geq \max\{\Omega(k^{12}/\epsilon^{3/2}), D\}$, then*

$$\Pr[\text{sign}(g^{(t)}(x)) \neq y] \leq L_\mathcal{D}(g^{(t)}) \leq \epsilon. \tag{631}$$

