# OpenReview forum: "A Theoretical Analysis on Feature Learning in Neural Networks: Emergence from Inputs and Advantage over Fixed Features"
_ICLR.cc/2022/Conference — ICLR 2022 Poster_

### Official Review · Reviewer_bp8c · 2021-10-31

**Correctness:** 4
**Technical Novelty And Significance:** 2
**Empirical Novelty And Significance:** 3
**Recommendation:** 5
**Confidence:** 4

**Main Review:**

+  a simple model
+ a positive learning result analyzing GD
+ a relevant lower bound

- the model is very specific with many assumptions ensuring that the GD analysis works out (e.g., irrelevant attribute effects are balanced out)
- all the interesting action in the upper bound happens in the first two rounds of GD, which suggests that this particular problem can also be solved by simple combinatorial methods
- the claimed SQ lower bound simply consists of showing that the parity function fits the model (i.e., no new lower bound insight here)
- it is not clear that kernel based learning cannot solve this problem; the text suggests this but I could not find any proof/evidence in the paper

**Summary Of The Paper:**

This paper gives results for learning a simple classification task by training a 2-layer NN with truncated ReLU activations. They also prove some lower bounds suggesting the necessity of the some of the model assumptions. From these results they try to infer that (a) their algorithm goes beyond the kernel/NTK regime and (b) the features learned are data/input specific and the task cannot be accomplished with oblivious features (such as random features).

**Summary Of The Review:**

The contributions of this paper, in my view, are a simple model that seems to require learning input-specific features, and a careful detailed analysis of the first two steps of GD applied to a 2-layer network with truncated ReLU activations. I think these are interesting, but might not be sufficient to justify acceptance.

---

> ### Author Response · Authors · 2021-11-12
> **Clarification for the results and model assumptions, Part 1/2**
>
> We thank the reviewer for the comments. We will address your concerns below.
>
> **Kernel based learning**
>
> Indeed, Theorem 2 can imply that kernel methods (of polynomial sizes like polynomially large RKHS norms) cannot solve the problem.
> The key is that these models can be approximated by the fixed feature models in Theorem 2 to arbitrarily small error with polynomially large dimensions and weight norms. For example, Claim 1 in [1] shows that random Fourier feature maps can approximate any shift-invariant kernel (e.g., RBF) to arbitrary error, and the number of Fourier features needed is polynomial in the dimension of the input data and the approximation (and some other related parameters). That is, they can be approximated by fixed feature models of polynomial sizes. The study in [2] has more discussions on approximating kernels with finite dimensional fixed features.
>
> Recall that Theorem 2 shows that fixed-feature models need to be exponentially large to get good accuracy. Now if there exist polynomial-size kernel-based models with good accuracy, then using the above approximation gives polynomial-size fixed feature models with similarly good accuracy, a contradiction to Theorem 2. So models using the kernels that can be approximated should be exponentially large to get good accuracy.
>
> We have only briefly mentioned the approximation before Theorem 2 as this is not our contribution. We will elaborate on this in the revision.
>
> [1] Ali Rahimi and Benjamin Recht. Random features for large-scale kernel machines. NeurIPS'08.
>
> [2] Pritish Kamath, Omar Montasser, and Nathan Srebro. Approximate is good enough: Probabilistic variants of dimensional and margin complexity. COLT'20.
>
> **Model assumptions**
>
> We acknowledge that our problem setup is simplistic, an abstraction of some motivations from practice. We would like to have analysis for more general settings modeling practical problems better. However, there is still a large gap between what we can analyze mathematically and what we observe in practice. In particular, understanding feature learning needs a direct analysis of the optimization dynamics, which is notoriously hard. As far as we know, our setting is already general compared to what has been analyzed in existing work along the line of study on feature learning for the superior performance of neural networks. We hope that our results, albeit in a restricted problem setup, can make a step towards a more thorough analysis.
>
> As a technical example, our setting is more general than that in Daniely and Malach (2020). To see this, let our dictionary be the identity matrix, the set $P$ to be the odd numbers (i.e., the labeling function is a sparse parity). Furthermore, let the distribution of the hidden representation be an equal mixture of the following two:
> 1. Uniform distribution over the hypercube
> 2. Irrelevant patterns $\tilde\phi_j (j \not \in A)$ have appearance probability $p_0 = 1/2$. And the distribution of relevant patterns $\tilde\phi_j(j\in A)$ is: all 0’s with probability 1/2, and all 1’s with probability 1/2.
> Then our problem setting reduces to their setting (up to scaling/translation of $\tilde\phi_j$'s). While our setting allows for more choices for the labeling, the dictionary, and the distributions over $\tilde\phi$.
>
> We would also like to mention that our assumptions can be relaxed. Appendix F presents results for more general settings, e.g., incoherent dictionary, removing assumption (A0) to allow imbalanced classes, adding noise to the input, etc. Furthermore, the assumption (A2) can also be relaxed. The analysis intuition carries over when the set of irrelevant patterns have a large enough variance. For example, we can allow different distributions for different patterns and let $p_0$ denote the minimum appearance probability; we can also allow dependence between these irrelevant patterns.

---

> > ### Author Response · Authors · 2021-11-12
> > **Clarification for the results and model assumptions, Part 2/2**
> >
> > **Upper bound**
> >
> > We note that the interesting actions in the upper bound analysis concentrate in the early stage of the training, since feature learning happens in the early stage, and since feature learning is the key phenomenon that lacks understanding and thus is our focus in this work. This is also consistent with the empirical observations that after a few training steps the networks can already get decent accuracy and the neurons’ weights are already highly structured and significantly different from the random initialization.
> > (A side note: our analysis can also be extended to more than two gradient steps, where the features keep getting improved. We analyze two steps for simplicity. )
> >
> > We do agree that there can exist other methods than the analyzed training algorithm that can learn effective features and thus achieve good accuracy. For example, since the input is generated in the dictionary learning model, by applying factor learning on the input alone, one can recover a lot of information on the dictionary and thus the effective features for the classifier. This is also consistent with recent studies on representation learning via pre-training: the input data distribution allows learning effective features. So this provides some support for our problem setup to capture some interesting aspects of practical data. And the other possible methods are also interesting research directions (such as helping understand unsupervised representation learning or developing alternative training methods), while they are well beyond the scope of this work and left for future study.
> >
> > We are not sure about the meaning of “simple combinatorial methods” in the review. If the reviewer can provide some clarification, we are happy to have further discussions.
> >
> > **Lower bound**
> >
> > Since we are analyzing a more general setting than existing work (so as to provide a more general analysis), our lower bound can be proved by reductions to existing lower bounds. This is expected, since our focus is on feature learning in the upper bound and we would like to analyze that in as general settings as possible.

---

### Official Review · Reviewer_Mhvy · 2021-11-02

**Correctness:** 3
**Technical Novelty And Significance:** 3
**Empirical Novelty And Significance:** 2
**Recommendation:** 8
**Confidence:** 3

**Main Review:**

The primary results of the paper are theoretical. It is well-written and clearly presented. An extensive discussion of the related work together with a clean crisp and highly relevant message on the role of input structure in explaining the learning capabilities of neural networks are some of the strengths of the paper. The experiments are also interesting although not very comprehensive (which is fine as the paper’s primary contributions are theoretical).
The only aspect where the paper lacks is perhaps that the main results and the techniques used seem very similar in spirit to those of Daniely and Malach (2020). Firstly, the lower bound to show that in the absence of any input structure learning can’t happen follows almost directly from a similar lower bound in Daniely and Malach (2020).
Given this, the main contribution of the paper is in showing the power of neural nets over data-independent kernel methods. Here, the paper does go into greater detail (in the Appendix) to contrast the contributions of current work to those of Daniely and Malach (2020) but it is not clear how technically novel the current analysis is. The phrase “Input structure” is a bit vague as one can argue that Daniely and Malach (2020)’s result also shows the importance of “input structure” (since their result shows that parities are learnable only under a certain input distribution). A key technical difference seems to be that the current setup requires a second gradient step to find the right patterns to focus on. This indicates that the current setup is more challenging to learn that the parity problem Daniely and Malach (2020) focus on.

Questions and suggestions:
1.	It would help to explicitly mention that the set A is fixed across inputs when describing the setup.
2.	Condition (A1) seems to not hold for the parity example (Example 2 in page 4)? Can you elaborate on how this condition is satisfied in the case of parities?

Minor typos and comments:
1.	Page 9: “chaning” -> “changing”
2.	In Theorem 1, k is Omega(log^2(m)) and m itself is Omega(k^12). This seems like a circular dependency. Can write this more clearly perhaps by using Theta instead of Omega in the appropriate place?


**Summary Of The Paper:**

The paper delves deep into learning dynamics of a one-layer neural networks and presents a class of functions and input distributions on which one layer neural networks trained with gradient descent learn well while no data-independent embedding map of the input to a polynomial number of dimensions can learn these function/distribution combinations without using weights which are exponentially large.
The main message seems similar in spirit to the paper titled “Learning Parities with Neural Networks” by Daniely and Malach, but as opposed to the artificially created problem of parities, the current paper presents a function/input distribution combination which is closer to distributions seen in practice. An additional result they show is that in the absence of any input structure the learning problem becomes hard under the statistical query model.
Finally the paper concludes by showing some experiments to support their claim that input structure is necessary to explain the good learning capabilities of neural networks.


**Summary Of The Review:**

The paper studies a very central problem in developing an understanding of deep learning today. It builds on a line of literature which theoretically capture settings where neural networks trained with gradient descent are provably more powerful than data-independent kernel methods. Moreover, they authors clearly outline the mechanism by which neural nets derive their additional power. Albeit a limited novelty (due to similarity to prior work), I believe the paper’s contributions are quite significant thereby I believe it clears the bar for acceptance at ICLR.

---

> ### Author Response · Authors · 2021-11-12
> **Thanks for the positive reviews, and discussion on related work**
>
> We thank the reviewer for the detailed positive reviews. Below we elaborate more on comparison with the related work. We would like to clarify the technical novelty of our work, and also point out where our analysis uses reduction to their results.
>
> **Differences and relation to Daniely and Malach (2020)**
>
> Our problem setting is **more general** than that in Daniely and Malach (2020). To see this, let our dictionary be the identity matrix, the set $P$ to be the odd numbers (i.e., the labeling function is a sparse parity). Furthermore, let the distribution of the hidden representation be an equal mixture of the following two:
> 1. D1: Uniform distribution over the hypercube
> 2. D2: Irrelevant patterns $\tilde\phi_j (j \not \in A)$ have appearance probability $p_0 = 1/2$. And the distribution of relevant patterns $\tilde\phi_j(j\in A)$ is: all 0’s with probability 1/2, and all 1’s with probability 1/2.
> Then our problem setting reduces to their setting (up to scaling/translation of $\tilde\phi_j$'s). While our setting allows for more choices for the labeling, the dictionary, and the distributions over $\tilde\phi$.
>
> **Upper bound:** Because of the more general setting, our upper bound proof requires **technical novelty**. Recall that in their work, the input distribution is essentially a mixture of D1 and D2. In D2, the relevant patterns $\tilde\phi_j (j\in A)$ have the specific structure of all 0's or all 1's with probability 1/2. This allows to show that neurons with weight $w$ satisfying $\sum_{j \in A} w_j = 0$ will have good gradients: small components from irrelevant patterns (their Lemma 7) and large components from relevant patterns (their Lemma 8). However, in our setting, the relevant patterns do not have this specific structure, and thus their proof technique is not applicable (or can be applied only when we have an exponentially large number of hidden neurons so that some hit the good positions at random initialization). What we showed is that the gradient has some correlation with the good feature direction. So after the first gradient step, the neuron weights are not good yet but are in a better position for further improvement (in particular, their setting corresponds to $p_0=1/2$ which means large noise in the weights after the first step; see discussion after our Lemma 5.) Then the latter gradient steps are able to improve the weights to better “signal-noise-ratio.” In summary, our proof does not rely on their specific input structure or an exponentially large number of hidden neurons for hitting some good positions. The key is that the good feature will emerge with the help of the input structure, and once in a better position, the neurons’ weights can be improved to the desired quality.
>
> **Lower bound:** On the other hand, our lower bound is proved by a reduction to the lower bound results in Daniely and Malach (2020). They have shown that D1 above can lead to large errors for fixed feature models of polynomial size. Our proof is essentially constructing a mixture of D1 and D2 with weights p_0 and (1-p_0), and applying their lower bound for D1. See our proof in Appendix C.
>
> **Conceptually,**  our work belongs to the same line of research as Daniely and Malach (2020), to analyze how feature learning leads to the superior performance of networks. While their analysis also relies on feature learning from good gradients induced by input structure, their focus is more on separating network learning and fixed feature models and has not explicitly explored the impact of input structures (while we agree that such an explicit study won’t be difficult in their setting). More importantly, their input distribution is specific and atypical in practice, which allows a specific type of feature learning (as explained in the above discussion on upper bounds). Our work thus considers a more general setting that is motivated by practical problems. Our results then bring theoretical insights closer for explaining the feature learning in practice and provide some positive evidence for the importance of analysis under proper models of the input distributions.
>
>
> **Condition (A1) for the parity example**
>
> Whether the parity example satisfies (A1) or not depends on the distribution over $\tilde{\phi}_j$. Consider the following example: $\tilde{\phi}_j (j \in A)$ are all 1’s with probability 1/2, and all 0’s with probability 1/2. Then (A1) is satisfied with $\gamma=1/2$. (This is essentially the example for the lower bound proof.) We will clarify this in the revision.
>
> Other comments:
> We will mention that the set $A$ is fixed and fix the typos in the revision. For parameter ranges in Theorem 1, the current inequalities can be viewed as a system specifying the allowable range of $(k,m)$ jointly. We will replace it by letting $m = poly(D)$ and $k=\Omega(\log^2(D/\gamma\delta))$.

---

### Official Review · Reviewer_zbTh · 2021-11-06

**Correctness:** 3
**Technical Novelty And Significance:** 3
**Empirical Novelty And Significance:** 2
**Recommendation:** 6
**Confidence:** 3

**Main Review:**

Strengths:
- The paper presents an interesting analysis of gradient descent under data-dependent feature distributions and the results appear to be novel.
- The paper is well-written and provides a good summary of related work (esp. in the Appendix).
- The experimental results, albeit limited, are encouraging.

Questions and Concerns:
- Can the authors elaborate on the relationship between network depth and the "emergence" of good features under the model considered herein?
- Can the authors provide explicit details on the improvements / differences with comparison to the results in Daniely and Malach (2020), specifically the lower bound?
- Can the authors provide some formulations for capturing class-dependent feature distributions in real-world problems? The definition assumes such a distribution exists, however one can think of several counterexamples (e.g., occlusion in visual domains) where "class-dependent" nature of features is not apparent.

**Summary Of The Paper:**

The problem of generalization in over-parameterized neural networks is of wide interest to the theoretical machine learning community, and this paper studies this problem from the lens of understanding structures present in feature distributions. The authors present a model for learning neural networks where the decision function can be expressed in terms of a set of correlated features and demonstrate that gradient descent recovers low generalization error under these assumptions. The authors demonstrate that in the absence of structure (i.e., when features are independent of each other), no algorithm can recover low error, which hints at how data-dependent feature learning is necessary for generalization in neural networks.

**Summary Of The Review:**

Overall I like the paper, and I think there are certain interesting results that are demonstrated, however, I believe the authors should elaborate more on the nature of the feature correlations considered: currently their model assumes two types of features, i.e., ones that are correlated with labels, and the ones that aren't. It perhaps would be interesting to examine whether this assumption can be weakened to something that holds stochastically over the data distribution, as such precise models for data generating processes are hardly available in practice (and are usually substituted by generative models).

---

> ### Author Response · Authors · 2021-11-12
> **Response Part 1/2**
>
> We thank the reviewer for the detailed comments, and would like to answer the questions raised.
>
> **Relationship between network depth and the emergence of good features**
>
> The networks our work analyzed have two layers, and the hidden layer allows the emergence of the good features (i.e., after training, the hidden neurons’ output contains good features allowing an accurate linear classifier on top). The depth 2 is necessary for learning the good features: the hidden neurons’ weights get updated by the gradient and the gradient correlates well with the underlying good features; after a few steps (2 steps in our analysis) the weights get dominated by the good features. To see the necessity of the depth, we compare to fixed feature models which can be viewed as having only depth 1, i.e., linear models over fixed features. They are essentially doing feature selection: putting larger weights on good features useful for classification and putting smaller weights on less useful features. The good features do not emerge in this case but just get picked out of the fixed features. This requires a large number of fixed features to cover potentially good features for various possible labeling functions. Or we need a large norm for the linear model’s weights, so that the linear combinations of the fixed features have enough flexibility to form good features. For our problem, it requires an exponentially large number of fixed features or an exponentially large norm of the linear model’s weights.
>
> Our analysis also provides some insights into deeper networks. For example, compare 2-layer and 3-layer networks. The 2-layer network can be viewed as a linear model over the first hidden layer, i.e., feature selection over the features represented by the first hidden layer. In contrast, the 3-layer network can use the second hidden layer to form new features on top of the first hidden layer (feature learning instead of feature selection). Then there are potentially problem settings where 3-layer networks can do polynomial learning while 2-layer networks cannot. A rigorous analysis for deeper networks is a very important and interesting future direction. That requires (we believe) a proper data generation model involving hierarchical hidden representations, and our current work serves as a beginning step and positive evidence towards that direction.
>
> **Feature distributions and correlations**
>
> We acknowledge that the partition of the patterns into class-relevant and class-irrelevant is simplistic, only an abstraction of the real data. This abstraction simplifies our analysis and highlights the intuition. On the other hand, the analysis can be carried out for the relaxed case where the label can have some weak dependence on the “irrelevant” patterns. This will only give an extra term in our analysis of the gradient and thus the neuron weights, which can then be bounded.  So it is possible to weaken this partition.
>
> For class-relevant patterns, note that our assumption (A1) does not make assumptions about the dependence between two class-relevant patterns. That means, we can view a non-occluded pattern and an occluded pattern as two patterns in $A$. And we should allow different correlation strengths for these patterns with the label.
>
> For class-irrelevant patterns, we think of them as background things that can appear in either class. The assumption (A2) for them can be easily relaxed. The analysis intuition carries over when the set of irrelevant patterns have a large enough variance. For example, we can allow different distributions for different patterns and let $p_0$ denote the minimum appearance probability; we can also allow dependence between these irrelevant patterns.

---

> > ### Author Response · Authors · 2021-11-12
> > **Response Part 2/2**
> >
> > **Differences and relation to Daniely and Malach (2020)**
> >
> > Our problem setting is **more general** than that in Daniely and Malach (2020). To see this, let our dictionary be the identity matrix, the set $P$ to be the odd numbers (i.e., the labeling function is a sparse parity). Furthermore, let the distribution of the hidden representation be an equal mixture of the following two:
> > 1. D1: Uniform distribution over the hypercube
> > 2. D2: Irrelevant patterns $\tilde\phi_j (j \not \in A)$ have appearance probability $p_0 = 1/2$. And the distribution of relevant patterns $\tilde\phi_j(j\in A)$ is: all 0’s with probability 1/2, and all 1’s with probability 1/2.
> > Then our problem setting reduces to their setting (up to scaling/translation of $\tilde\phi_j$'s). While our setting allows for more choices for the labeling, the dictionary, and the distributions over $\tilde\phi$.
> >
> > Because of the more general setting, our lower bound is proved by a reduction to the lower bound results in Daniely and Malach (2020). They have shown that D1 above can lead to large errors for fixed feature models of polynomial size. Our proof is essentially constructing a mixture of D1 and D2 with weights $p_0$ and $(1-p_0)$, and applying their lower bound for D1. See our proof in Appendix C.
> >
> > We would like to emphasize **our technical novelty** in the upper bound proof: our more general setting requires new techniques. Recall that in their work, the input distribution is essentially a mixture of D1 and D2. In D2, the relevant patterns $\tilde\phi_j (j\in A)$ have the specific structure of all 0's or all 1's with probability 1/2. This allows to show that neurons with weight $w$ satisfying $\sum_{j \in A} w_j = 0$ will have good gradients: small components from irrelevant patterns (their Lemma 7) and large components from relevant patterns (their Lemma 8). However, in our setting, the relevant patterns do not have this specific structure, and thus their proof technique is not applicable (or can be applied only when we have an exponentially large number of hidden neurons so that some hit the good positions at random initialization). What we showed is that the gradient has some correlation with the good feature direction. So after the first gradient step, the neuron weights are not good yet but are in a better position for further improvement (in particular, their setting corresponds to $p_0=1/2$ which means large noise in the weights after the first step; see discussion after our Lemma 5.) Then the latter gradient steps are able to improve the weights to better “signal-noise-ratio.” In summary, our proof does not rely on their specific input structure or an exponentially large number of hidden neurons for hitting some good positions. The key is that the good feature will emerge with the help of the input structure, and once in a better position, the neurons’ weights can be improved to the desired quality.

---

### Official Review · Reviewer_uM4G · 2021-11-06

**Correctness:** 4
**Technical Novelty And Significance:** 3
**Empirical Novelty And Significance:** 2
**Recommendation:** 6
**Confidence:** 3

**Details Of Ethics Concerns:**

N\A

**Main Review:**

**Strength**:

* A novel analysis of neural learning that connects the learning performance with its ability in adapting to input structure. The analysis is made accessible due to the explicit assumption on data-generation mechanism, which I believe is sufficient given the early-stage of this line of work.
* The theoretical results are empirically verified.

**Weakness**:

This work can greatly benefit from expanding the coverage of the experiments, in terms of data-generation models, sample size and data dimension, etc. Specifically

* Expand the experiment coverage in terms of sample size / data dimension (and optionally other axis such as class imbalance). Since we are stating the main theorems in terms sample size / data-dimension. I wonder if author can expand the first two sections of experiments ( "Simulation: Verification of the Main Results" and "Simulation: Feature Learning in Networks.") to test the robustness of the claim and understand the learning behaviors in a variety of sample size and dimension settings.

* Evaluation in "Simulation: Feature Learning in Networks.": This section seems to be rather crucial in supporting the theoretical claims. However, most of the results reported are qualitative (Figure 2). Please consider reporting quantitative result (e.g., in terms of certain distance measure between model features and the ground-truth $M$), and examine across different data dimensions,sample size, and optionally with respect to class imbalance, etc.

* Please discuss (in problem setup or in conclusion section) the expressiveness of the function space defined in (1). In particular, what type of data generating function is not yet covered by such model? (e.g., there is a nonlinear corruption process involved in the data-generation mechanism). Correspondingly, in the experiment section (e.g., "Simulation: Verification of the Main Results"), it might be also extend the experiment to more flexible data-generation mechanisms (say y=NN(x) where NN is a 1-layer network) to verify if the result still holds and in what extent.


Minor:

* Clarity: "(A1’) The patterns are independent, and ...". Please clarify, do you mean the patterns in A is independent from the labels?

**Summary Of The Paper:**

This work proves the feature learning ability of a 1-hidden-layer network (Theorem 1), under the assumption that the data is generated from a dictionary learning model. Additional result on the limitation of no feature learning and no input structure is also provided (Theorem 2-3). The proof proceeds by closed-form analysis of gradient dynamics, which is possible thanks to the simplicity of the data-generation mechanism. Authors empirically verified their claim on synthetic and semi-realistic experiments.

**Summary Of The Review:**

A novel theoretical analysis that connects neural model's learning behavior to its ability in adapting to input structure. The analysis is done under a simple data-generating model and was relatively accessible. Author empirically verified their claims in simulation experiment. which slightly weakens the quality of this contribution.

---

> ### Author Response · Authors · 2021-11-12
> **Thanks for the suggestions on the experiments**
>
> We thank the reviewer for the positive comments on the analysis and the detailed suggestions on the experiments. We are now running the experiments and will report the results soon.
>
>
> **Experiments on different parameters of the data generation**
>
> We agree that experiments on different data dimensions, class imbalance, and sample size for the first two sections will strengthen the empirical support for the analysis.
>
>
> We note that our analysis is on full gradient descent and does not involve sample size. On the other hand, we also note that the analysis can be carried out with large enough sample sizes to ensure accurate gradients. So we expect that the feature learning will be similar to what we reported if we use large sample sizes. For smaller sizes, we will need the size to be not too small to avoid overfitting. When the size is not too small but not large enough to guarantee an accurate gradient, the sample noise gets into the gradient, and thus more gradient steps may be needed so that the effective feature of the gradients dominates the neuron weights.
>
>
> **Quantitative results for "Simulation: Feature Learning in Networks"**
>
> We have quantitative results in the appendix (Table 1 in Appendix E.1.1), including the cosine similarities between the neuron weights to the effective feature ($\sum_{j\in A} M_j$). In particular, we can see that after one gradient step the cosine similarity is 0.848, and after two steps it’s 0.997. This shows the feature learning phenomena is very significant. We note that we use cosine similarity instead of distance (like Euclidean distance),  since our analysis only shows the convergence of the directions of the neuron weights, but their lengths can vary. We will also report the results for the new experiments when we vary the parameters like data dimension.
>
>
> **Expressiveness of the function space**
>
> The function space in (1) covers quite rich cases. In particular, the labeling function can be any binary function on the pattern count. Therefore, it covers the majority function on the count, and the parity functions. In Appendix F, we present a more general setting, where we allow an incoherent dictionary and also noise in the input $x$. This can also cover the case of a mixture of two Gaussians.
>
>
> On the other hand, there are definitely many interesting data generation processes not covered in (1). Here are two cases we think are very interesting:
>
> 1) Hierarchical representations: first generate hidden representation $h_k$, then generate lower-level hidden representation $h_{k-1}$, ..., finally generate input $x$ from the hidden representation $h_1$. This is because in practice we do observe such hierarchical representations, and it is also believed that the power of deep networks comes from the ability to learn hierarchical representations.
>
> 2) Nonlinear transformations: as suggested by the reviewer, one could consider the case that we generate the input by some nonlinear transformation after the linear mapping from the hidden representation. This should be an integral part of the hierarchical generating process above.
>
> These are important generating processes to model the data. However, it is beyond our current analysis and we leave the study of these settings as future work.
>
>
> Minor: (A1’) means that each pattern is independent of the other patterns. That is, each $\tilde\phi_i$ is independent of the other pattern indicator $\tilde\phi_j (j\neq i)$’s. Then the distribution of the hidden vector $\tilde\phi$ is just uniform over the hypercube. It doesn’t mean that the label is independent of the patterns in $A$.

---

> ### Author Response · Authors · 2021-11-18
> **Updates on Additional Experiments**
>
> We have performed the suggested experiments (varying parameters for the experiments, and more data-generation mechanisms). The results also provide some positive support for our insights.
>
> **Varying parameters**
>
> We performed the simulation on our synthetic data with different data dimensions, class imbalance ratios, and sample sizes. The setup follows that in the old experiments. The results/figures are shown in Appendix G.1 of the revised submission.  (We are not aware of a way of posting figures in the response, so we add the figures in the revised pdf submission and summarize the results in text here.)
>
> 1. Data dimensions.
>
>     We add the experiments with data dimensions D=​​ 100 and 2000. (The old experiment used D=500.) The results, including test accuracy, cosine similarities, and visualization, are very similar to those in our old experiment. One small difference is that after one step, the average cosine similarity between neuron weights is only 0.56 for D=2000 and is 0.97 for D=100  (compared to 0.85 for D=500). This is because in high dimensions it’s harder to get large cosine similarities. But for different D’s, after the second step, the cosine similarity is >0.9, indicating the features have been improved significantly as predicted by our analysis.
>
> 2. Class imbalance
>
>     We add experiments with class ratios 4:1 and 9:1. (The old experiment used 1:1.) Again, the results are very similar to the old experiment, showing the robustness of our claims. A difference is that the NKT method becomes less stable as shown in the test accuracy figures.
>
> 3. Sample size
>
>     We add experiments with dataset sizes 25000 and 10000. (The old experiment used 50000.)  For size 25000, the results are similar to the old experiment. For size 10000, the learning fails with overfitting (the test accuracy is much worse than the training accuracy). To verify this is due to overfitting rather than the failure of our claims, we train a smaller network on data size 10000 (reducing the number of neurons from m=300 to 50). This then leads to successful learning and results similar to the old ones, confirming the robustness of our insights.
>
> **More data generation mechanisms**
>
> Note that our analysis is for the setting where the input distributions have structure revealing some information about the labeling function. (More precisely, the labeling function is specified by $A$ and $P$, while the input distribution also depends on them.) Therefore, our analysis doesn’t apply to the settings where the labeling function is a network with no connections to the input distribution.
> We then consider two other data generation mechanisms where the labeling function has connections to the input distributions. The results and figures are in Appendix G.2.
>
> 1. One typical model in existing studies is a mixture of well-separated Gaussians. We thus first generate $t$ Gaussians, and assign label 1 to some Gaussian components and label -1 for the others. The labeling function is roughly equivalent to a network: $y = \sum_{i=1}^{t} a_i \text{ReLU}(\langle c_i, x \rangle)$ where $c_i$’s are the Gaussian centers, and $a_i \propto 1$ for Gaussian components with label 1 and $a_i \propto -1$ for those with label -1. The results on such data show similar trends as on our old synthetic data: in the early stage of training, the neurons learn the effective features (i.e., the centers $c_i$’s), and later they lead to an accurate classifier.
>
> 2. We also tried another data generation mechanism: we randomly select some binary values for $\tilde{\phi}_j (j \in A)$,  assign label 1 to some and label -1 to the others; uniformly sample irrelevant patterns $\tilde{\phi}_j (j \not\in A)$; the input is generated by $M\tilde{\phi}$. This generalizes a component in the distribution used in the old experiment. The experiments on such data also show successful learning and the feature learning phenomena, providing further support for our analysis.

---

### Official Review · Reviewer_Jr9n · 2021-11-07

**Correctness:** 3
**Technical Novelty And Significance:** 3
**Empirical Novelty And Significance:** 2
**Recommendation:** 6
**Confidence:** 2

**Main Review:**

Strengths:
- The paper is well motivated and studies a question of great interest to the community (the effects and benefits of NN feature learning), and highlight the importance of input structure, which to my knowledge is often neglected in the literature.
- The paper (to the best of my knowledge) is technically sound.
- I appreciated section 5 which helps to provide some intuition for the results (although also see weaknesses).

Weaknesses:
- Theorem 2 seems slightly weak in that it requires a bounded feature map dimension to work to show that fixed feature models fail, so what about infinite dimensional feature maps like say kernel methods? This is not an issue with other settings to show that fixed feature models fail e.g. https://arxiv.org/abs/2012.09816 (which also should be in the related works, as they also look at input structure, a 'multi-view setting' to show the benefits of NN feature learning over feature selection methods like random features).
- The experimental section is not very convincing to show that your theory is relevant for NNs in practice. It would be good to see Figure 3 for a larger setting, perhaps CIFAR10 with a deeper resnet/cnn. You have a nice simplified story for how NNs learn features from your theory: 1st step extracts rough features, 2nd step refines these, then finally the linear classifier is trained. I'm not convinced that this holds in practice with the provided experiments.
- Likewise the networks for MNIST are very narrow, you have m=9? Perhaps MNIST isn't a very good dataset to depict the importance of NN feature learning, as I believe linear models can already achieve a respectable score.
- I appreciate the efforts of the authors, but I'm still somewhat lacking in the intuition for the setup and how it is relevant to practical settings. For example, the first line of section 3 talks about circles and lines, but you make no further mention of this example. How does your 'structured' data look like in terms of circles and lines, particularly your assumption A1) which is the key assumption I believe, compared to A1')?

Minor points:
- Bottom of page 2: close not closed
- Under lemma 5, should be j \in [A] not j \in [D]?
- The setting of having a changing regularisation strength (in classifier learning stage paragraph just before sec 5.2) is quite strange/not standard. Can you provide a bit more justification for this please?

**Summary Of The Paper:**

This is a theoretical paper that studies the feature learning properties of neural networks (NN) and the importance of input structure on the advantages of NN's over fixed feature models like linear or random features models. The authors create a setup motivated by real data where they can analytically show that a single hidden layer NNs can achieve good error (Theorem 1), whereas models with fixed features need to be exponentially large (in the number of relevant features) to match (Thm 2). The authors also show that without input structure, polynomial algorithms in the Statistical Query model achieve random guessing error (Thm 3). Some experimental evidence is provided to support some of the claims.

**Summary Of The Review:**

I'm recommending weak reject, because although I believe the paper is well-motivated and makes some inroads into an important question, I have concerns about some of the theoretical results, the sufficiency of the experimental results to justify the theory, and also some of the intuitive explanations for the theoretical setup.

---

> ### Author Response · Authors · 2021-11-12
> **Clarification about the results and the setup**
>
> We thank the reviewer for the detailed comments. We will clarify that Theorem 2 can be applied to the infinite dimensional feature maps. We are now running the suggested experiments and will report the results soon.
>
> **First bullet**
>
> Theorem 2 can also be applied to infinite dimensional feature maps like kernel methods. This is because they can be approximated by finite dimensional feature maps. For example, Claim 1 in [1] shows that finite dimensional Fourier features can approximate any shift-invariant kernel (e.g., RBF), and the number of features needed is polynomial in the input dimension and the approximation error (and some other related parameters). The study in [2] has more discussions on approximating kernels with finite dimensional maps.
>
> Recall that Theorem 2 shows that models using finite dimensional feature maps need to be exponentially large to get good accuracy. Now if there exist polynomial-size kernel-based models with good accuarcy, then the above approximation gives polynomial-size finite dimensional models with similarly good accuracy, a contradiction to Theorem 2. So models using the kernels that can be approximated should be exponentially large to get good accuracy.
>
> We have only briefly mentioned the approximation before Theorem 2 as this is not our contribution. We will clarify this better in the revision.
>
> We thank the reviewer for pointing out the related work [3] and will add it to our related work section. On the other hand, we note that their focus is on ensembles and is quite different from ours: the analysis is on showing the multi-view input structure allows the ensembles of networks to improve over single ones, while ensembles of fixed feature mappings do not have improvement.
>
> [1] Ali Rahimi and Benjamin Recht. Random features for large-scale kernel machines. NeurIPS'08.
>
> [2] Pritish Kamath, Omar Montasser, and Nathan Srebro. Approximate is good enough: Probabilistic variants of dimensional and margin complexity. COLT'20.
>
> [3] Zeyuan Allen-Zhu, Yuanzhi Li. Towards Understanding Ensemble, Knowledge Distillation and Self-Distillation in Deep Learning.
>
> **Second and third bullets**
>
> 1. We will add the experiments in Figure 3 for a larger setting.
> 2. Our analysis can be carried out for more gradient steps following similar intuition, while we analyze two steps for simplicity. The gradient steps can be repeated multiple times (with step sizes properly scaled down), and the analysis for each such step will be similar to the current analysis.
> We acknowledge that our problem setup is only a theoretical model and the practical scenario can be more complicated. However, we do believe that in practice, similar feature learning phenomena happen: in the early stage, gradient descent helps effective features emerge and improve (though not in two steps but in many more steps); the classifier on top of the effective features gradually learns to good accuracy.
> 3. For “Real Data: Feature Learning in Networks” on MNIST, m=50. For “Real Data: The Effect of Input Structure”, m=9. We will run experiments with larger networks for the latter, and we expect similar results there.
>
> **Last bullet**
>
> We will clarify the intuition better in the revision. Here is a simple illustrative example. Suppose the pattern vectors $M_j (j \in A)$ correspond to circles, lines, rectangles etc. There are two classes: indoor room images and outdoor flower images. If there are patterns like straight lines and rectangles appearing then it is likely to be an indoor room image, ie, the occurrence of such patterns correlates positively with the class label as stated in (A1).
>
> Minor: We will correct the typos in the revision. For regularisation, we agree that a changing strength is not standard. Our purpose of setting a large regularization strength in the first two gradient steps is for the learned features to replace the old neuron weights (see the remark before the classifier learning stage paragraph). More precisely, since
>           new weight = old weight * (1- 2 * learning_rate * regularisation) - learning_rate * gradient,
> a regularisation strength 1/(2* learning_rate) makes sure the old weight is forgotten and the features learned by gradient dominate the new weight.
>
> It is possible to do the proof using the more standard way of fixed regularisation but with a more complicated analysis. The first two gradient steps can be replaced with many more gradient steps so that eventually the neurons' weights are dominated by the effective features in the gradient. The analysis is then repeating that for our current second gradient step multiple times. We choose our current presentation for better illustration, following the practice as in related work like [4,5].
>
> [4] Amit Daniely and Eran Malach. Learning parities with neural networks. NeurIPS'2020.
>
> [5] Eran Malach, Pritish Kamath, Emmanuel Abbe, and Nathan Srebro. Quantifying the benefit of using differentiable learning over tangent kernels.

---

> > ### Comment · Reviewer_Jr9n · 2021-11-19
> > **Thank you, more clarifications please**
> >
> > Thanks for responding to my review and for the extra results. These will help me make a more informed decision of the work. I still have a few confusions after your reply which I would appreciate some extra clarifications please?
> >
> > 1. *Theorem 2* I'm still not convinced by the explanation you provided for Theorem 2 re kernel methods. As I understand it, you argued that RFF allows shift-invariant kernels to be polynomially (in input dimension $d$) approximated, whereas Theorem 2 requires exponential (in $k$) feature dimension. It is not clear to me on the relation between sizes of $k$ and $d$, which is necessary for your argument to go through right?
> >
> > > Recall that Theorem 2 shows that models using finite dimensional feature maps need to be exponentially large to get good accuracy. Now if there exist polynomial-size kernel-based models with good accuarcy, then the above approximation gives polynomial-size finite dimensional models with similarly good accuracy, a contradiction to Theorem 2. So models using the kernels that can be approximated should be exponentially large to get good accuracy.
> >
> > It feels like there is some circular logic going on above^ and it is not clear wrt what one is 'polynomial' or 'exponential'. If we do not consider finite approximations to kernels and instead purely think about infinite dimensional kernel methods, then I am still struggling to see how Theorem 2 tells us anything about the inability of fixed infinite-dimensional features to perform in your setting?
> >
> > 2. *Experiments* Can I ask what do the green and red clusters in Figure 26 symbolise, and what about the blue weights? Are they clusters that have been hand picked from step 20? Also sec G.3 is on page 75 not 73 of the revised work.

---

> > > ### Author Response · Authors · 2021-11-19
> > > **Thanks for the response and further clarification**
> > >
> > > **Theorem 2**
> > >
> > > Our reasoning is as follows. In general, for any function $f$ using a kernel, we can first apply the approximation to show the existence of a function $\langle \Psi(x), w\rangle \approx f(x)$ where the dimension of $\Psi$ and the norm of $w$ are polynomial in the related parameters of $f$. Then we apply our Theorem 2 on $\langle \Psi(x), w\rangle$ to show that its loss is at least about $p_0 (1- \text{(polynomial of related parameters)}/2^k)$, and thus the loss of $f$ is at least about this amount. Then at least one of the related parameters of $f$ needs to be exponential in $k$ to get nontrivial loss.
> > >
> > > To be concrete, let’s consider a shift-invariant kernel $K$ and apply the random feature approximation.
> > >
> > > 1. Consider any function $f$ using a shift-invariant kernel with RKHS norm bounded by $L$. Or as typical in practice, a function $f(x) = \sum_i \alpha_i K(z_i, x)$ for some $\alpha_i$’s and $z_i$’s, where $||\alpha||_2 \le L$. Let $d$ denote the input dimension.
> > >
> > > 2. Now apply the random feature approximation in [1] to this classifier. We then get that for any $\epsilon>0$, there exists $N$ Fourier features $\Psi_j$ and a weight vector $w$ such that $\langle \Psi(x), w\rangle$ approximates $f(x)$ upto error $\epsilon$, and we have $N= \text{polynomial}(d, L, 1/\epsilon)$ and the norm of $w$ bounded by $B=\text{polynomial}(d, L, 1/\epsilon)$.
> > >
> > >     More technical details: consider $f(x) = \sum_i \alpha_i K(z_i, x)$. We get that for any $\nu>0$, there exists $N=\text{polynomial}(d, 1/\nu)$ Fourier features $\Psi_j$ that can approximate the kernel up to error $\nu$. Then $f$ can be approximated by $\sum_i \alpha_i \langle \Psi(z_i), \Phi(x) \rangle = \langle \sum_i \alpha_i \Psi(z_i), \Psi(x) \rangle$. Let $w = \sum_i \alpha_i \Psi(z_i)$ and let $\nu$ sufficiently small, then $\langle \Psi(x), w\rangle$ approximates $f(x)$ upto error $\epsilon$ and $N, B$ are polynomials in $d, L, 1/\epsilon$. The reasoning is the same for $f$ in the RKHS form, replacing sum with integral.
> > >
> > > 3. Next apply our Theorem 2 on the above $\langle \Psi(x), w\rangle$. Then it will have hinge-loss at least $p_0 (1 - \sqrt{2N}B/2^k)$. So the function $f$ has loss at least $p_0 (1 - \text{polynomial}(d, L, 1/\epsilon)/2^k) - \epsilon$. We can choose $\epsilon=1/\text{polynomial}(d, L)$. Then this means to get nontrivial loss, at least one of $d, L$ needs to be exponential in $k$.
> > >
> > > [1] Ali Rahimi and Benjamin Recht. Random features for large-scale kernel machines. NeurIPS'08.
> > >
> > > **Update:** in the revised draft, we have updated the second paragraph after Theorem 2 to incorporate the above clarification.
> > >
> > > **Experiments**
> > >
> > > In Figure 26, the red and green points denote some selected neuron weights that have high cosine similarities, and the blue points are the other neuron weights.
> > >
> > > The points are selected at step 20. We first visualize the weights at step 20, and then hand pick the points that roughly form two clusters (i.e., the points in the same cluster are close to each other while those in different clusters are far away). We assign red and green colors to the two clusters at step 20, and then assign these weights with the same color in step 0 and 3. Finally, we compute the cosine similarities and show that the hand picked points are indeed roughly clusters in the high-dimension.
> > >
> > > **Update:** we have updated Appendix G.3 with the above clarification about the visualization. We also added the experimental results for a larger network ResNet(256) that has 256 filters in the first residual block, which show more significant feature learning and clustering effect. This agrees with our explanation that more overparameterized networks can have more filters/neurons learn the same effective feature and thus form clusters.

---

> > > > ### Comment · Reviewer_Jr9n · 2021-11-28
> > > > **Thanks**
> > > >
> > > > Thanks for the clarification and sorry for delay. Your argument for theorem 2 applying to infinite dimensional feature maps makes sense. Can I ask why you have only mentioned this corollary in passing in the updated manuscript? To me it seems more profound than theorem 2 as my concern with theorem 2 is that it is simply a problem of underparameterisation?

---

> > > > > ### Author Response · Authors · 2021-11-28
> > > > > **Thanks for the feedback**
> > > > >
> > > > > **Why only mention this corollary**
> > > > >
> > > > > The main reason why we didn't have a theorem for infinite-dimensional feature maps is that we don't have a unified formal statement for different kernels. As mentioned above, in general, an $f$ using a kernel will have a loss at least about $p_0 (1 - (\text{polynomial of related parameters of } f)/2^k)$. The exact form of the polynomial of related parameters depends on the kernel used. To have a unified formal statement, existing work (e.g., [1]) has adopted the approach of presenting a formal statement for finite-dimensional feature maps and referring to approximation results using finite feature maps to approximate kernels (e.g., [2]). We thus followed this convention.
> > > > >
> > > > > We will add the corollary about the RBF kernel into our future version. In this way, readers can see a concrete example of applying Theorem 2 to infinite-dimensional feature maps. Hopefully,  this example combined with our discussion will let the reader get an intuition about the general case.
> > > > >
> > > > > [1] Amit Daniely and Eran Malach. Learning parities with neural networks. NeurIPS'2020.
> > > > >
> > > > > [2] Pritish Kamath, Omar Montasser, and Nathan Srebro. Approximate is good enough: Probabilistic variants of dimensional and margin complexity. COLT'20.
> > > > >
> > > > >
> > > > > **Theorem 2 is simply a problem of underparameterisation?**
> > > > >
> > > > > The main reason for the lower bound in Theorem 2 is not underparameterisation, but because the feature map $\Psi$ is independent of the data.
> > > > >
> > > > > 1. There exists a small linear model on a small dimensional feature map allowing 0 loss for each data distribution in our problem set $\mathcal{F}_\Xi$.
> > > > >
> > > > >     Formally, Lemma 4 shows that there exists a two-layer network with loss 0. Each hidden neuron $\sigma(\langle w_i^*, x \rangle + b_i^*) $ can be viewed as a feature, and the hidden layer of that network can be viewed as a feature map. We can see that the dimension is just the number of neurons $3(k+1)$, and the weights $a^*_i$ on the feature map have small values bounded by $32k$.
> > > > >
> > > > > 2. However, the small dimensional feature map is different for different data distribution in our problem family $\mathcal{F}_\Xi$, i.e., depends on the data.
> > > > >
> > > > >     The feature $\sigma(\langle w_i^*, x \rangle + b_i^*) $ depends on $w_i^*, b_i^*$, which are different for different data distributions in $\mathcal{F}_\Xi$.
> > > > >
> > > > > 3. On the other hand, the feature map $\Psi$ is data-independent, i.e., it's fixed before seeing the data. For $\Psi$ to work simultaneously for all the data distributions in our problem family $\mathcal{F}_\Xi$, it needs to have exponential dimensions.
> > > > >
> > > > >     Intuitively, it needs a large number of features, so that for each $\sigma(\langle w_i^*, x \rangle + b_i^*) $ there are some features to approximate $\sigma(\langle w_i^*, x \rangle + b_i^*) $. There are exponentially many data distributions in our problem family $\mathcal{F}_\Xi$, and thus exponentially many data-dependent features $\sigma(\langle w_i^*, x \rangle + b_i^*) $, which requires $\Psi$ to have an exponentially large dimension. Infinite-dimensional features maps like kernels are similar: though they are in infinite dimensions, the ''effective dimension'' of a model using them is finite (more precisely, the model can be approximated by a linear model on a finite-dimensional feature map where the linear weight norm and the feature map dimension are polynomial in the related parameters).
> > > > >
> > > > >     We can let $\Psi$ have dimensions polynomial in $k$, for example, $3(k+1)^3$. This is larger than the dimension of the ground-truth in Lemma 4, so it's overparameterized, not underparametrized. But still, it cannot achieve a good loss, due to that $\Psi$ is fixed rather than depending on the data.
> > > > >
> > > > >     On the other hand, network learning updates the hidden neuron (i.e., the features) using the data, so it can move the features to the right positions to approximate the ground-truth data-dependent features $\sigma(\langle w_i^*, x \rangle + b_i^*) $. It then doesn't need an exponentially large dimension feature map.

---

> > > > > > ### Comment · Reviewer_Jr9n · 2021-12-02
> > > > > > **Thank you, raising my score**
> > > > > >
> > > > > > Thanks for these clarifications. I am raising my score to 6. I hope that you can convey both these intuitions (regarding kernel approximations and also it is not underparameterisation but rather data-independentness of features that leads to theorem 2) more clearly in a revised version.

---

> > > > > > > ### Author Response · Authors · 2021-12-02
> > > > > > > **Thank you**
> > > > > > >
> > > > > > > Thanks for the discussion! We will incorporate these intuitions in the future revision.

---

> ### Author Response · Authors · 2021-11-18
> **Updates on Additional Experiments**
>
>  **Larger models on CIFAR10**
>
> We have performed experiments using deep convolutional neural networks on two classes of CIFAR10. The results align with our theoretical insights that feature learning happens in the early stage of the training where the gradients guide the neurons’ weights (filters for CNNs) towards directions of effective features.
>
> We used a ResNet-18 convolutional neural network with 128 filters in the first residual block on the Automobile and Airplane classes from CIFAR10. The training follows typical practice, with a mini-batch size of 100. The training runs for 20 epochs and the final test accuracy is 95.75%.
>
> The results and visualization of the filter weights in different layers at Epochs 0, 3, 20 are shown in Appendix G.3 on Page 73 of the revised submission.  (We are not aware of a way of posting figures in the response, so we add the figures in the revised pdf submission and summarize the results in text here.) Take the figure for the last layer as an example.
>
> 1. We can see that the filter weights change significantly during the early stage of the training, indicating feature learning happens in the early stage: the change between Epoch 0 and Epoch 3 is much more significant than that between Epoch 3 and Epoch 20.
>
> 2. We also verify that the feature learning is guided by the gradients: the gradients of a filter in the early gradient steps point to similar directions (and thus the updated filter will learn this direction). More precisely,  for a filter, we average the gradients every 10 gradient steps (so to reduce the variance due to mini-batch), and get v1 v2 and v3 for the first 30 steps and compute the average pairwise cosine similarities between them. The average cosine similarities are about 0.6 to 0.65, consistently for different neurons. This is significant in the high-dimension (the filters are in 9216 dimensions).
>
> 3. We also observe some clustering effect of the filter weights, though not as significant as in our simulations. In the figure, we show two clusters in red and green. The average cosine similarity for filter weights in the red cluster is about 0.62 and that for the green is about 0.7, while the cosine similarity between the two clusters’ centers is about -0.72. This shows significant similarities within the cluster while difference between clusters.
>
>     Note that the clustering is less significant than our simulation experiments. This is because practical data have more patterns (i.e., effective feature directions) to be learned than our synthetic data, and also the practical network is not that overparameterized as in our simulation. Then filters are likely to learn different patterns (or their mixtures) without forming significant clusters. The results of ResNet-18 with fewer filters (64 filters) on all classes of CIFAR10 show less significant clustering which supports our explanation. On the other hand, we emphasize that the key insight of our analysis is that the gradient guides the learning of effective features in the early stage of training, which is verified as discussed above.
>
>
> **Large network on MNIST to check the effect of input structure**
>
> We added the experiment using networks with $m=50$ hidden neurons for MNIST (the old experiments used $m=9$). The experiment followed the old setup, only increasing $m$.
> The results/figures are included in Appendix G.4. We can see that they are similar to those for $m=9$. The separation between the learning on the MNIST part and the Gaussian part is actually more significant than that for $m=9$, providing stronger support for our insights.

---

### Author Response · Authors · 2021-11-18
**Revision Uploaded**

We have uploaded a revised submission, incorporating the various suggestions by the reviewers. We would like to thank the reviewer again for these valuable suggestions, and are happy to hear any further feedback and provide more discussion.


**Added Clarifications**

1. We clarified that the lower bound for fixed features (Theorem 2) can also be applied to the infinite-dimensional feature maps. See the second paragraph after Theorem 2. (Suggested by Reviewer Jr9n and bp8c)

2. We added some intuition behind the problem setup in the first paragraph of the section. (Suggested by Reviewer Jr9n)

3. We added a brief discussion of its limitations after Example 2. (Suggested by Reviewer uM4G)

4. We clarified that set A is fixed in the distribution.  (Suggested by Reviewer Mhvy)


**Added Suggested Experiments**

The added experiments are summarized below. Please see the response to individual reviewers and see the revised submission for more details.

1. We added simulation experiments for varying parameters: different input data dimensions, class imbalance ratios, and sample sizes. The results are similar to our old simulation results, demonstrating the robustness of our results. See Appendix G.1. (Suggested by Reviewer uM4G)

2. We added experiments on two more data generation models in Appendix G.2. The results show similar feature learning phenomena in the early stage of training.  (Suggested by Reviewer uM4G)

3. We added experiments of ResNet on two classes of CIFAR10 in Appendix G.3. We verified that the filter weights are guided by the gradients to learn effective features during the early stage of the training. We observed less significant clustering of the filter weights and gave some explanations with evidence. (Suggested by Reviewer ​​Jr9n)

4. We added experiments using a larger network on MNIST to check the effect of the input structure in Appendix G.4. The results are similar to those in our old experiments using a smaller network. Actually, the results even better align with our analysis. (Suggested by Reviewer ​​Jr9n)


**Minor**

We have also corrected the typos pointed out by the reviewers.

---

> ### Author Response · Authors · 2021-11-20
> **Further Revision Uploaded**
>
> We have uploaded a new revision with the following updates:
>
> **Clarifications**
>
> 1. We updated the clarification that the lower bound for fixed features (Theorem 2) can also be applied to the infinite-dimensional feature maps, in the second paragraph after the theorem. (Suggested by Reviewer Jr9n)
>
> 2. We updated Appendix G.3 with clarification about the visualization. We also added the experimental results for a larger network, which provide stronger support for our explanation. (Suggested by Reviewer Jr9n)
>
> **More general problem setup**
>
> We have updated the analysis in Appendix F for a more general problem setting with a relaxed assumption (A2). The old Appendix F already showed that network learning can succeed under a general problem setting allowing incoherent dictionary, noise, class imbalance etc (and thus containing the result in the main text as a special case). The revision considers an even more general setting, in particular, relaxes the assumption (A2) on class irrelevant patterns. Now it only requires the class irrelevant patterns are independent of the class relevant patterns, but it allows class irrelevant patterns to be dependent on each other and can have different distributions. (Related to reviews by Reviewer zbTh and bp8c)

---

### Decision · Program_Chairs · 2022-01-20

**Decision:**

Accept (Poster)

**Comment:**

The authors theoretically analyze learning of two-layer neural
networks by gradient descent with respect to a data distribution that
exposes how useful features are learned during training.

Overall, the reviewers felt that the analysis yielded useful insight,
and was original.

During the discussion period, a reviewer recommended that the authors
look at papers providing lower bounds on statistical query learning of two-layer networks,
and consider comparing the lower-bound technique of this paper with that earlier work.